# Learning to Rank from Incomplete Rankings

**Cristiano Migali** [1]   **Gianmarco Genalti** [1]   **Alberto Maria Metelli** [1]   **Marco Mussi** [1]

## Abstract

In domains such as recommender systems and information retrieval, learning from human-generated feedback is especially challenging because the information provided is often sparse and incomplete. In this work, we address the problem of learning the top-$k$ items from incomplete rankings. Most existing models for incomplete rankings rely on rigid assumptions regarding both the ranking model that generates the latent ranking and the censoring mechanism that determines which comparisons remain unobserved. On the one hand, the ranking model is often assumed to follow a Plackett-Luce (PL) or Mallows distribution. On the other hand, the censoring mechanism is typically assumed to be Missing Completely At Random (MCAR) or to exhibit well-behaved dependencies on the latent ranking, such as winner feedback or top-$h$ feedback. We introduce a new, general framework for learning from incomplete rankings that unifies and strictly generalizes the established frameworks in the literature. We consider the broad class of ranking models that satisfy the complete consensus property, which comprehends all widely adopted models, including PL and Mallows. Furthermore, we present a new preference-based feedback model, named positional censoring, which generalizes winner and top-$h$ feedback. We show that it is possible to learn in this general setting by presenting the PIRATE algorithm and providing a near-optimal instance-dependent bound to the sample complexity. Finally, we show that, under the PL ranking, PIRATE matches the sample complexity of state-of-the-art algorithms in the relevant scenarios of winner and top-$h$ feedback.

[1]Politecnico di Milano, Milan, Italy. Correspondence to: C. Migali <cristiano.migali@polimi.it>.

*Proceedings of the $43^{rd}$ International Conference on Machine Learning*, Seoul, South Korea. PMLR 306, 2026. Copyright 2026 by the author(s).

## 1. Introduction

*Preference* is the most natural form of human-generated feedback. Given two alternatives, a human is asked to compare them and tell which one is preferred. When the alternatives are more than two, it is unrealistic to expect a human to provide a comparison between each pair, or, equivalently, a full ranking of the alternatives. It is more likely that the human will provide a few of the comparisons only, i.e., an incomplete ranking. Datasets consisting of collections of incomplete rankings are ubiquitous in the real world, spanning domains from recommender systems (Rendle et al., 2009), where the human interacts with just a few of the items in the catalog, to large language model training, where human annotators rank only a small set of model-generated responses (Ouyang et al., 2022). To exploit the vast amount of information hidden in these datasets, it is crucial to develop robust methods that aggregate noisy, incomplete rankings to learn the underlying preferences among the alternatives. Previous works partially addressed the problem of learning to rank from incomplete rankings. However, none of them provides a unified view on the problem under the lens of both the ranking model, which describes the distribution of the latent ranking among the alternatives, and the feedback type, which determines the comparisons that are observed. Most of the existing works rely on the assumption that the ranking model follows a Plackett-Luce (PL) distribution (Plackett, 1975; Luce, 1959), e.g., (Negahban et al., 2017; Fahandar et al., 2017; Hajek et al., 2014; Khetan & Oh, 2016; Saha & Gopalan, 2019a), or a Mallows distribution (Mallows, 1957), e.g., (Busa-Fekete et al., 2014; Chierichetti et al., 2018; Lu & Boutilier, 2014). On the feedback model side, we have that previous works either assume that the comparisons are Missing Completely At Random (MCAR), i.e., the missing comparisons are independent from the positions of the corresponding alternatives in the latent ranking (e.g., Negahban et al., 2017), or that they have well-behaved dependencies on the latent positions. Examples of this second class of feedback models are *winner feedback*, where the learner observes only the best alternative, and *top-$h$ feedback* (Saha & Gopalan, 2019a), in which the learner observes the ordered collection of the top-$h$ alternatives. Fahandar et al. (2017) introduce the *rank-dependent coarsening* feedback model, where the learner observes an incomplete ranking involving only the alternatives that occupy a certain subset

of the positions in the latent ranking. This feedback model generalizes winner and top-$h$ feedback, allowing more complex dependencies on the latent positions, but their analysis considers only pairwise comparisons under the PL model.

**Research Goal.** The goal of this work is to answer the following research question.

> *Is it possible to design an algorithm which learns to rank the alternatives from the observation of incomplete rankings when the customary assumptions on the ranking model (PL or Mallows), and the feedback type (MCAR, winner, or top-$h$ feedback) are relaxed?*

In particular, we study the problem of identifying the collection of top-$k$ alternatives and their order from incomplete rankings when the latent ranking is generated by the broad class of ranking models which satisfy the mild assumption of *complete consensus* (Critchlow et al., 1991)—which include PL and Mallows models with customary metrics—and the comparisons are Missing Not At Random (MNAR), i.e., missing comparisons depend on the positions of the corresponding alternatives in the latent ranking. In particular, we introduce a new feedback type, which generalizes winner feedback, top-$h$ feedback, and rank-dependent coarsening, allowing to model complex dependencies between the observed comparisons and the position of the corresponding alternatives in the latent ranking.

**Summary of Contributions.** The contributions of this paper are organized as follows.

- In Section 3, we introduce *positional censoring*, a novel feedback type for MNAR comparisons, and show how it generalizes existing models. Then, we formalize the *top-$k$ item learning under positional censoring* problem.
- In Section 4, we present the PIRATE algorithm and characterize its sample complexity when the ranking model satisfies complete consensus.
- In Section 5, we show that PIRATE enjoys near-optimal sample complexity under the established Plackett-Luce ranking model—with both winner and top-$h$ feedback—matching specialized algorithms from the literature.

The rest of the manuscript is structured as follows. Section 2 introduces the notation used throughout the paper and provides the fundamental notions on ranking models and feedback types for incomplete rankings. In Section 6, we compare the behavior of the PIRATE algorithm on synthetic data against well-established baselines in the field of Learning to Rank (LTR, Liu, 2009). The related works are in Appendix A and provide an extensive overview of the assumptions on both the ranking model and the feedback type which are customary in the literature. Appendix B supplements Section 2 with additional notation and preliminary

technical details omitted from the main text for brevity. In Appendix C, we show that many established methods in the incomplete rankings literature fail to identify the correct ranking among the items in some instances of the general setting we tackle in this work. Appendices D and E provide the proofs of the results stated in the paper.

## 2. Preliminaries

In this section, we introduce the notation that will be used throughout the paper, together with fundamental notions on ranking models and observation models in this context.

### 2.1. Notation

Let $\Delta(\mathcal{X}, \mathcal{F})$ denote the set of probability measures over the measurable space $(\mathcal{X}, \mathcal{F})$. We write $\Delta(\mathcal{X})$ when the $\sigma$-algebra $\mathcal{F}$ is clear from context. For any $l, n \in \mathbb{N}_{\geq 0}$ with $l \leq n$, let $[\![l, n]\!] \coloneqq \{l, \dots, n\}$ and $[\![n]\!] \coloneqq [\![1, n]\!]$. We denote by $S_n$ the set of *permutations* over $[\![n]\!]$, i.e., the set of bijective functions $\pi : [\![n]\!] \to [\![n]\!]$. We use $e \in S_n$ for the *identity permutation* $e(l) = l$ for every $l \in [\![n]\!]$ and $\tau_{l,m} \in S_n$ for the permutation that *swaps* $l, m \in [\![n]\!]$, $l \neq m$: $\tau_{l,m}(v) = v\mathbb{I}[v \neq l, m] + m\mathbb{I}[v = l] + l\mathbb{I}[v = m]$ for every $v \in [\![n]\!]$. We denote by $\omega \in S_n$ the *reverse permutation*: $\omega(l) = n+1-l$ for every $l \in [\![n]\!]$. Given a set $\mathcal{X}$, let $\mathrm{SPos}(\mathcal{X})$ denote the set of *strict partial orders* $\prec \subseteq \mathcal{X}^2$ over $\mathcal{X}$, i.e., binary relations satisfying the following properties for every $x, y, z \in \mathcal{X}$: (a) *irreflexivity*, $x \nprec x$; (b) *asymmetry*, if $x \prec y$, then $y \nprec x$; and (c) *transitivity*, if $x \prec y$ and $y \prec z$. When $\mathcal{X} = [\![n]\!]$, we write $\mathrm{SPos}_<([\![n]\!])$ to denote the set of strict partial orders over $[\![n]\!]$, consistent with the standard order over integers $<$. Formally: $\mathrm{SPos}_<([\![n]\!]) \coloneqq \{\prec \in \mathrm{SPos}([\![n]\!])$ s.t. $x \prec y$ implies $x < y$ for every $x, y \in [\![n]\!]\}$. We use $I_{n,<} \in \mathbb{R}^{n \times n}$ for the upper triangular matrix with ones above the main diagonal. Given $A, B \in \mathbb{R}^{n \times n}$, we denote by $\langle A, B \rangle_F \coloneqq \sum_{l,m \in [\![n]\!]} a_{l,m} b_{l,m}$ their Frobenius product and by $\|A\|_F$ the Frobenius norm. For every $A \in \mathbb{R}^{n \times n}$, we denote by $\mathrm{triu}(A) \coloneqq (a_{l,m}\mathbb{I}[l < m])_{l,m \in [\![n]\!]}$.

### 2.2. Ranking Models

A *ranking model* $\rho \in \Delta(S_n)$ is a probability distribution over the set of permutations $S_n$ (Critchlow et al., 1991). Ranking models allow us to describe the ranking of a set of $n$ items from best (1) to worst ($n$) according to some non-deterministic process. In particular, let $\mathcal{I}$ be a finite set of items with cardinality $n$ and $\ell : \mathcal{I} \to [\![n]\!]$ be a bijective labeling function that associates a unique integer label $\ell(i)$ to each item $i \in \mathcal{I}$. A permutation $\pi \in S_n$, sampled from the ranking model $\rho$, describes the position $\pi(\ell(i))$ of item $i \in \mathcal{I}$ in the observed ranking. When the ranking model admits a unique *modal ranking* $\pi_0 = \arg\max_{\pi \in S_n} \rho(\pi)$, it induces an underlying ranking of the items: $i \prec j$ if and

only if $\pi_0(\ell(i)) < \pi_0(\ell(j))$ for every $i, j \in \mathcal{I}$.

Notable examples of ranking models include Random Utility Models (RUMs, Thurstone, 1927), the Plackett-Luce model (PL, Luce, 1959; Plackett, 1975) and Mallows $\phi$-model (Mallows, 1957).

**Random Utility Models.** A RUM is described by an ordered collection of $n$ distributions $\boldsymbol{\nu} = (\nu_l)_{l \in [\![n]\!]}$ with $\nu_l \in \Delta(\mathbb{R})$. $\nu_l$ is assumed to be absolutely continuous w.r.t. the Lebesgue measure, i.e., $\nu_l \ll \lambda$ for $l \in [\![n]\!]$. The ranking model $\rho$ is induced by an ordered collection of independent random variables $(U_l)_{l=1}^n$ with $U_l \sim \nu_l$, that model the perceived utility of each item. In particular:

$$\rho(\pi) = \mathbb{P}_{\boldsymbol{\nu}}[U_{\pi^{-1}(1)} > \cdots > U_{\pi^{-1}(n)}] \text{ for every } \pi \in S_n,$$

where $\mathbb{P}_{\boldsymbol{\nu}} = \otimes_{l \in [\![n]\!]} \nu_l$. The meaning is that the observed ranking depends on the realizations of the perceived utilities of the items: the item with the highest perceived utility is ranked first, and so on. Note that, being the distributions absolutely continuous, the probability of ties $U_l = U_m$ for $l, m \in [\![n]\!]$, $l \neq m$ is 0, so that $\sum_{\pi \in S_n} \rho(\pi) = 1$. In the literature, it is standard to study RUMs where the utility distributions $(\nu_l)_{l=1}^n$ are a *location family*, i.e., their CDFs $(F_l)_{l=1}^n$ satisfy $F_l(x) = F(x - u_l)$, where $u_l$ is a shift parameter representing the true utility of the item $\ell^{-1}(l)$. Such models are referred to as *Thurstone models* (Yellott, 1977) and are of particular interest because of their tractability.

**Plackett-Luce Model.** The PL model is a special case of the RUM where the utility distributions are assumed to be $\mathrm{Gumbel}(u_l, 1)$ for every $l \in [\![n]\!]$, making PL a Thurstone model. This particular choice provides a closed form for $\rho$ (Yellott, 1977), that is:

$$\rho(\pi) = \prod_{r \in [\![n-1]\!]} \frac{\exp\left(u_{\pi^{-1}(r)}\right)}{\sum_{s \in [\![r,n]\!]} \exp\left(u_{\pi^{-1}(s)}\right)} \text{ for } \pi \in S_n. \quad (1)$$

Due to Equation (1), it is usual to parametrize the PL model by $\theta_l = \exp(u_l)$ for $l \in [\![n]\!]$. PL is the unique RUM that satisfies the Independence of Irrelevant Alternatives (IIA) assumption (Luce, 1959): the relative likelihood of choosing one item over another is unaffected by the alternatives.

**Mallows $\phi$-Models.** A Mallows $\phi$-model is described by a collection $(\phi, d, \pi_0)$ where $\phi \in (0, 1]$, $d$ is a metric on the set of permutations and $\pi_0$ is the modal ranking. The probability distribution over permutations is $\rho(\pi) = C(\phi, d, \pi_0)\phi^{d(\pi, \pi_0)}$, with $C(\phi, d, \pi_0)^{-1} = \sum_{\pi \in S_n} \phi^{d(\pi, \pi_0)}$. In a Mallows $\phi$-model, $\rho(\pi)$ decreases as $\pi$ gets further from $\pi_0$ according to $d$. The parameter $\phi$ controls the spread of $\rho$ from the modal ranking. The most popular metric for this class of models is the Kendall metric:

$$T(\pi, \sigma) = \sum_{\substack{l, m \in [\![n]\!] \\ l < m}} \mathbb{I}[((\pi(l) - \pi(m))(\sigma(l) - \sigma(m)) < 0].$$

**Complete Consensus of a Ranking Model.** Critchlow et al. (1991) identify several desirable properties of ranking models. Here, we focus on the *complete consensus* property.

**Definition 2.1** (Complete Consensus, Critchlow et al. 1991)**.** A ranking model $\rho \in \Delta(S_n)$ satisfies *complete consensus* with respect to a *consensus ordering* $\pi_0 \in S_n$ if, for every $l, m \in [\![n]\!]$ such that $\pi_0(l) < \pi_0(m)$, and any permutation $\pi$ where $\pi(l) < \pi(m)$, it holds that $\rho(\pi) \geq \rho(\pi \circ \tau_{l,m})$, with strict inequality if $\pi = \pi_0$.[1] If the inequality is strict for every such $\pi$, we say $\rho$ satisfies *strict complete consensus*.

Intuitively, complete consensus implies that if item $i$ precedes item $j$ in the consensus ordering (i.e., $\pi_0(\ell(i)) < \pi_0(\ell(j))$), then, holding the positions of all other items unmodified, any ranking where $i$ precedes $j$ is more likely than the one where $j$ precedes $i$. Complete consensus is a common property, enjoyed by many of the ranking model families studied in the literature.[2]

### 2.3. Observation Models in Ranking

In a typical ranking problem, we assume that a latent permutation $\pi$ is sampled from a ranking model $\rho$. However, the learner may not observe the full permutation $\pi$, but rather an incomplete version of it. We report relevant observation models from the literature.

**Full-Ranking Feedback.** The most informative setting, where the realization of the permutation $\pi \sim \rho$ is observed.

**Top-$h$ Feedback.** This setting often arises in the context of subset-wise preferences (Saha & Gopalan, 2019a), where the learner observes the ordering of the items occupying the first $h$ positions, but receives no information regarding the ordering of the other items.

**Winner Feedback.** This is a specific instance of top-$h$ feedback with $h = 1$ (Saha & Gopalan, 2019a). Here, the learner observes only the item ranked in the first position.

**Pairwise Feedback.** This observation model is typical in dueling bandits (Yue et al., 2012). This model assumes the observation is limited to the relative preference in the latent ranking between a pair of items $i, j \in \mathcal{I}$, $i \neq j$, often chosen by the learner.

**Rank-Dependent Coarsening.** This observation model selects an ordered collection of positions $\boldsymbol{r} = (r_1, \ldots, r_h) \in [\![n]\!]^h$ with $h \in [\![2, n]\!]$ and $r_u < r_{u+1}$ for every $u \in [\![h-1]\!]$. The observation consists in the ordered list of items occupying the specified positions in the latent ranking.

---

[1] Definition 2.1 implies that the consensus ordering is the unique modal ranking of the ranking model.

[2] For example, Henery (1981) shows that if the shift parameters $(u_l)_{l \in [\![n]\!]}$ of a Thurstone model $\rho$ are distinct and the likelihood ratio $F'(x - u_l)/F'(x - u_m)$ is a non-increasing function of $x$ for $u_l < u_m$, $l, m \in [\![n]\!]$, then $\rho$ enjoys complete consensus.

*Table 1.* Definition of strict partial orders which allow to implement established observation models through positional censoring.

| Model | Order Definition |
|---|---|
| Full-Rank. | $\prec_{\text{pos}}^{\text{full}} := \{(l, m) \text{ s.t. } l < m\}$ |
| Top-$h$ | $\prec_{\text{pos}}^{\text{top-}h} := \{(l, m) \text{ s.t. } l \in [\![h]\!], l < m\}$ |
| Winner | $\prec_{\text{pos}}^{\text{win}} := \{(l, m) \text{ s.t. } l = 1, l < m\}$ |
| Rank-Dep. Coarsening | $\prec_{\text{pos}}^{r} := \{(r_u, r_v) \text{ s.t. } u, v \in [\![h]\!], u < v\}$ |

## 3. Problem Formulation

In this section, we introduce the concept of *positional censoring* and provide its mathematical formulation. We then formally define the learning problem addressed in this work.

### 3.1. Positional Censoring

*Positional censoring* is a novel observation model for incomplete ranking data presented in this work. The model posits that in incomplete ranking data—represented as strict partial orders over the set of items $\prec_{\text{obs}} \in \text{SPos}(\mathcal{I})$—the absence of a comparison between a pair of items $(i, j)$ is significantly influenced by their unobserved positions in the latent permutation $\pi$, i.e., comparisons are MNAR. Our model relies on the concept of *positional censorship mechanism*, defined as a strict partial order over the ranking positions: $\prec_{\text{pos}} \in \text{SPos}_{<}([\![n]\!])$ such that $\prec_{\text{pos}} \neq \{\}$. Given the permutation $\pi$ sampled from the ranking model $\rho$, the observed incomplete ranking data is defined as the pullback relation of $\prec_{\text{pos}}$ under the bijection $\pi \circ \ell$: $i \prec_{\text{obs}} j$ iff $\pi(\ell(i)) \prec_{\text{pos}} \pi(\ell(j))$. Let $\mathcal{M} := \{\prec_{\text{pos}} \in \text{SPos}_{<}([\![n]\!]) \text{ s.t. } \prec_{\text{pos}} \neq \{\}\}$ be the set of positional censorship mechanisms. The model has the following interpretation: $\prec_{\text{pos}}$ determines the pairs of positions in the latent ranking which correspond to pairs of items that are compared by the human in the observed incomplete ranking $\prec_{\text{obs}}$. Observe that, since $\prec_{\text{pos}} \in \text{SPos}_{<}([\![n]\!])$, then $i \prec_{\text{obs}} j$ implies $\pi(\ell(i)) < \pi(\ell(j))$. This means that $\prec_{\text{pos}}$ can censor part of the ranking but it cannot alter the relative position of a pair of items. In Table 1, we show how positional censoring can implement most of the established observation models in the literature.

### 3.2. Setting

In this work, we address the problem of learning the ordered collection of top-$k$ items from a set of $n$ items $\mathcal{I}$ ($k \leq n$) with confidence $1 - \delta$ ($\delta \in (0, 1]$), under the positional censoring observation model described in Section 3.1.

An instance of the learning problem is described by a collection $\xi = (\mathcal{I}, \rho, \psi, \ell^*, k)$, where:

• $\mathcal{I}$ is a set of $n \in \mathbb{N}_{\geq 2}$ items.

• $\rho \in \Delta(S_n)$ is a ranking model with unique modal ranking $\pi_0$ corresponding to the identity permutation: $\pi_0 = e$.[3]

• $\psi \in \Delta(\mathcal{M})$ is a distribution over the set of positional censorship mechanisms $\mathcal{M}$.

• $\ell^* : \mathcal{I} \to [\![n]\!]$ is a bijective labeling function which associates a unique integer to each item. Being the modal ranking of $\rho$ equal to the identity, $\ell^*$ encodes the underlying ranking among items induced by the ranking model (see Section 2.2): $i \prec j$ if and only if $\ell^*(i) < \ell^*(j)$ for every $i, j \in \mathcal{I}$. We denote by $i_l^* = \ell^{*-1}(l)$ for every $l \in [\![n]\!]$, so that: $i_1^* \prec \cdots \prec i_n^*$.

• $k \in [\![n]\!]$ is the number of top items (i.e., items occupying the first positions in the underlying ranking) that the learner must identify. In particular, the learner is expected to identify the ordered collection: $(i_l^*)_{l \in [\![k]\!]}$.

In each round $t \in \mathbb{N}_{\geq 1}$, the learner observes an incomplete ranking $\prec_{\text{obs}}^{(t)} \in \text{SPos}(\mathcal{I})$, obtained as follows. A permutation $\pi^{(t)} \sim \rho$ is sampled from the ranking model. A positional censorship mechanism $\prec_{\text{pos}}^{(t)} \sim \psi$ is sampled from the corresponding distribution, *independently* from $\pi^{(t)}$. The observed incomplete ranking $\prec_{\text{obs}}^{(t)}$ is the pullback relation of $\prec_{\text{pos}}^{(t)}$ under the bijection $\pi^{(t)} \circ \ell^*$:

$$i \prec_{\text{obs}}^{(t)} j \text{ iff } \pi^{(t)}(\ell^*(i)) \prec_{\text{pos}}^{(t)} \pi^{(t)}(\ell^*(j)) \text{ for } i, j \in \mathcal{I}.$$

The learner is *passive*: in each round, it can either ask for one more sample or return an ordered collection of $k$ distinct items which it estimates as the top-$k$. Formally, the learner is modeled as a function Alg which takes three arguments. The first argument is $k \in [\![n]\!]$, i.e., the number of top items that the learner must identify. The second argument is the confidence parameter $\delta \in (0, 1]$ which determines the required confidence $1 - \delta$ for the estimation. The third argument is an ordered collection of size $T \in \mathbb{N}_{\geq 1}$ of observations: $(\prec_{\text{obs}}^{(t)})_{t \in [\![T]\!]}$. The learner returns $\perp$ when asking for an additional sample. Otherwise it returns the ordered collection of distinct items $(\hat{i}_1^*, \ldots, \hat{i}_k^*)$, $\hat{i}_l^* \neq \hat{i}_m^*$ for every $l, m \in [\![k]\!]$, $l \neq m$, estimated as the top-$k$.

We work under the following mild assumptions on the ranking model $\rho$.

**Assumption 3.1** (Strict Complete Consensus)**.** The ranking model $\rho$ satisfies *strict complete consensus* (Definition 2.1) with *consensus ordering* corresponding to the identity $e$.

The fact that $\rho$ satisfies strict complete consensus prevents the construction of pathological instances: in Theorem D.1

---

[3]The existence of a unique modal ranking is required to define a learning objective. Indeed our learning objective is specified in terms of the underlying ranking among items $i \prec j$ induced by the ranking model (see Section 2.2). The fact that the modal ranking corresponds to the identity is w.l.o.g., since, if it were $\pi_0 \neq e$, we could relabel the items changing the labeling function from $\ell$ to $\pi_0^{-1} \circ \ell$ obtaining $i \prec j$ if and only if $\ell(i) < \ell(j)$.

we show that, relaxing Assumption 3.1, we can find instances of the learning problem which induce the same probability distribution over observations despite having a different underlying ranking between items, making the identification of the top-$k$ items not feasible.

**Assumption 3.2** (Full Support). The ranking model $\rho$ has full support over the set of permutations $S_n$.

Analogously to Assumption 3.1, in Theorem D.2 we show that relaxing Assumption 3.2, we can find instances where the top-$k$ items identification is not feasible. Furthermore, Theorem F.1 proves that, under Assumption 3.1, Assumption 3.2 is equivalent to $\rho(\omega) > 0$ where $\omega$ is the reverse permutation. This property is satisfied by most of the ranking model families established in the literature, including RUMs, PL, and Mallows $\phi$-models. We denote by $\mathcal{E}$ the set of problem instances $\xi$ satisfying Assumptions 3.1 and 3.2.

We now formally define the *learning objective*. We adopt the *fixed confidence* setting (often referred to as $\delta$-correct or PExact), a standard framework in pure exploration problems (Even-Dar et al., 2006; Kalyanakrishnan et al., 2012). For an instance $\xi \in \mathcal{E}$, we denote by $\mathbb{P}_\xi$ the probability distribution over the sequences of observations and by $\mathbb{E}_\xi$ the corresponding expectation. Let $\mathbb{F} = (\mathcal{F}^{(T)})_{T \in \mathbb{N}_{\geq 1}}$ be the natural filtration induced by the observations. Let $T_{\text{Alg}}^{(\delta)}$ be a stopping time w.r.t. the filtration $\mathbb{F}$, defined as the first round $T$ where the algorithm returns a non-$\perp$ value. With the necessary notation established, we formally define the properties of an admissible learner.

**Definition 3.3** ($\delta$-Correct Learner). The learner Alg is said to be $\delta$-*correct* (or *PExact*) if, for every confidence parameter $\delta \in (0, 1]$ and every problem instance $\xi \in \mathcal{E}$, it satisfies the following two properties:

- *Almost Sure Termination:* the algorithm terminates in finite time with probability 1, i.e.:

$$\mathbb{P}_\xi\left[ T_{\text{Alg}}^{(\delta)} < +\infty \right] = 1.$$

- *Correctness:* upon termination, the returned ordered collection coincides with the true ordered top-$k$ items with probability greater than $1 - \delta$. Formally:

$$\mathbb{P}_\xi\left[ \text{Alg}\left( k, \delta, (\prec_{\text{obs}}^{(t)})_{t \in \llbracket T_{\text{Alg}}^{(\delta)} \rrbracket} \right) = (i_l^*)_{l \in \llbracket k \rrbracket} \right] > 1 - \delta.$$

The quality of a $\delta$-correct learner is evaluated through its sample complexity (the lower, the better).

**Definition 3.4** (Sample Complexity). The *sample complexity* of a $\delta$-correct ($\delta \in (0, 1]$) learner Alg on a problem instance $\xi \in \mathcal{E}$ is defined as the expected number of samples required for termination. We denote this quantity by:

$$S_\xi^{(\delta)}(\text{Alg}) := \mathbb{E}_\xi\left[ T_{\text{Alg}}^{(\delta)} \right].$$

---

**Algorithm 1** PIRATE Algorithm.

1: **Input:** $k \in \llbracket n \rrbracket$, $\delta \in (0, 1]$, $(\prec_{\text{obs}}^{(t)})_{t \in \llbracket T \rrbracket} \in \text{SPos}(\mathcal{I})^T$
2: **for** $i, j \in \mathcal{I}, i \neq j$ **do**
3:     Let $w_{i,j} \leftarrow \sum_{t=1}^{T} \mathbb{I}[i \prec_{\text{obs}}^{(t)} j]$
4:     Let $N_{i,j} \leftarrow \sum_{t=1}^{T}(\mathbb{I}[i \prec_{\text{obs}}^{(t)} j] + \mathbb{I}[j \prec_{\text{obs}}^{(t)} i])$
5:     Let $\hat{\mu}_{i,j} \leftarrow w_{i,j}/N_{i,j}$ if $N_{i,j} > 0$, $1/2$ otherwise
6:     Let $b_{i,j}^{(\delta)} \leftarrow \sqrt{\frac{\ln(3Tn/\delta)}{N_{i,j}}}$ if $N_{i,j} > 0$, $+\infty$ otherwise
7: **end for**
8: Let $E^{(\delta)} \leftarrow \{(i, j) \in \mathcal{I}^2 \text{ s.t. } \hat{\mu}_{i,j} - b_{i,j}^{(\delta)} > \hat{\mu}_{j,i} + b_{j,i}^{(\delta)}\}$
9: Let $\hat{\prec}^{(\delta)} \leftarrow$ transitive closure of $E_\delta$
10: Let $d_{\text{out}}^{(\delta)}(i) \leftarrow$ out-degree of $i$ in $(\mathcal{I}, \hat{\prec}^{(\delta)})$
11: Let $(d_{\text{out}}^{(\delta)}(\ell^{-1}(l)))_{l \in \llbracket n \rrbracket} \leftarrow$ sorting of $(d_{\text{out}}^{(\delta)}(i))_{i \in \mathcal{I}}$
12: **if** $(d_{\text{out}}^{(\delta)}(\ell^{-1}(l)))_{l \in \llbracket n-k+1, n \rrbracket} = (n-k, \ldots, n-1)$ **then**
13:     **return** $(\ell^{-1}(n - l + 1))_{l \in \llbracket k \rrbracket}$
14: **end if**
15: **return** $\perp$

---

## 4. The PIRATE Algorithm: $\delta$-Correctness and Near-Optimality in the General Setting

In this section, we introduce the Patching Incomplete RAnkings through Transitive Edges algorithm (a.k.a. PIRATE, Algorithm 1) for the top-$k$ item learning under positional censoring problem which we prove is a $\delta$-correct learner. Then, we develop an upper bound to its sample complexity, and provide a matching lower bound up to logarithmic dependencies, proving that the algorithm is near-optimal.

### 4.1. Algorithm

The PIRATE algorithm (Algorithm 1) constructs a dominance graph $(\mathcal{I}, E^{(\delta)})$ on the set of items $\mathcal{I}$ by adding an edge from item $i$ to item $j$ only when sufficient statistical evidence has accumulated to infer, with confidence $1 - \delta/n^2$, that item $i$ precedes item $j$ in the underlying ranking $\prec$. To this end, it builds an estimate $\hat{\mu}_{i,j}$ of the probability $\mu_{i,j}$ that item $i$ precedes item $j \neq i$ in the observed incomplete ranking, conditioned on the event that a comparison between $i$ and $j$ is observed. Formally $\mu_{i,j} := p_{i,j}/(p_{i,j} + p_{j,i})$, where $p_{i,j} = \mathbb{P}_\xi\left[ i \prec_{\text{obs}}^{(1)} j \right]$ for every $i, j \in \mathcal{I}$, $i \neq j$. The algorithm adds the edge $(i, j)$ to the dominance graph when the lower confidence bound of $\mu_{i,j}$ is greater than the upper confidence bound of $\mu_{j,i}$ (Line 8). This implies that with high probability: $\mu_{i,j} > \mu_{j,i}$, which, under Assumption 3.1 on the ranking model $\rho$, in turn implies that $i \prec j$. At the cost of propagating the uncertainty, we can state that item $i$ precedes $j$ in the underlying ranking when there is a path from $i$ to $j$ in the dominance graph, being the relation $\prec$ transitive. When the path has length greater than 1, we say that the inference is *transitive*, in contrast to the *direct* inference represented by $(i, j) \in E^{(\delta)}$. The transitive inference

that an item precedes another in the ranking is correct as long as all the involved direct inferences $(i, j) \in E^{(\delta)}$ are correct. Since the probability that a direct inference is correct is at least $1 - \delta/n^2$, they are all correct with probability at least $1 - \delta$, being $n^2$ greater than the maximum number of direct inferences in the graph, which is $n(n-1)/2$. The algorithm builds an estimate of the underlying ranking $\hat{\prec}^{(\delta)}$ by taking the transitive closure of the dominance graph $(\mathcal{I}, E^{(\delta)})$ (Line 9). The out-degree $d_{\text{out}}^{(\delta)}$ (Line 10) of a node $i$ in the transitive closure $(\mathcal{I}, \hat{\prec}^{(\delta)})$ is the number of items $j$ which follow $i$ in the estimated ranking. The item $\hat{i}_1^*$ with out-degree $d_{\text{out}}^{(\delta)}(\hat{i}_1^*) = n - 1$ precedes $n - 1$ items, being thus the first in the ranking. By iterating this reasoning, the items $(\hat{i}_l^*)_{l \in [\![k]\!]}$ with $d_{\text{out}}^{(\delta)}(\hat{i}_l^*) = n - l$ for $l \in [\![k]\!]$ are the first $k$ items in the ranking. Hence, when there is a collection of $k$ items satisfying such condition (Line 12), the algorithm can terminate.

The formalization of these arguments, which is deferred to Appendix E, allows to prove that PIRATE is $\delta$-correct.

**Theorem 4.1** ($\delta$-Correctness of PIRATE). *For every confidence parameter $\delta \in (0, 1]$ and problem instance $\xi \in \mathcal{E}$:*

- PIRATE *terminates in finite time with probability* 1.
- *Upon termination, the ordered collection returned by* PIRATE *coincides with the true ordered top-k items with probability greater than $1 - \delta$.*

### 4.2. Sample Complexity Upper Bound

PIRATE terminates when the following two complementary identification problems are resolved:

(P1) Identification of the order among the top-$k$ items $i_1^*, \ldots, i_k^*$: $i_1^* \hat{\prec}^{(\delta)} \cdots \hat{\prec}^{(\delta)} i_k^*$.

(P2) Identification of the item $i_k^*$ which separates the top-$k$ items: $i_l^* \hat{\prec}^{(\delta)} i_k^*$, $l \in [\![k-1]\!]$, from the remaining $n - k$: $i_k^* \hat{\prec}^{(\delta)} i_l^*$, $l \in [\![k+1, n]\!]$.

We can reframe these problems as equivalent conditions on the dominance graph. Specifically, (P1) is solved when all the edges $(i_l^*, i_{l+1}^*)$ with $l \in [\![k-1]\!]$ are added to $E^{(\delta)}$. (P2) instead, as a direct consequence of the definition of the transitive closure $\hat{\prec}^{(\delta)}$, demands the existence of a path from $i_k^*$ to every $i_l^*$, $l \in [\![k+1, n]\!]$ in the dominance graph. For item labels $l < m$, let $S_{l,m}$ be the *expected number of samples* required to add the edge $(i_l^*, i_m^*)$ to the dominance graph. The expected number of samples to solve (P1) is simply $\max_{l \in [\![k-1]\!]} S_{l,l+1}$. For (P2), the matter is more delicate. In particular, we want to identify the subset of the edges $E_k^* \subseteq \{(i_l^*, i_m^*) \text{ s.t. } l, m \in [\![k, n]\!], l < m\}$, which connects $i_k^*$ to $i_l^*$, $l > k$, and minimizes the bottleneck cost $\max_{(i,j) \in E_k^*} S_{\ell^*(i), \ell^*(j)}$. This is an instance of the *minimum bottleneck spanning arborescence* problem (Camerini,

1978). When the graph is directed and acyclic (as in our case), Lemma F.5 shows that the minimum bottleneck cost admits a closed form:

$$\max_{(i,j) \in E_k^*} S_{\ell^*(i), \ell^*(j)} = \max_{m \in [\![k+1, n]\!]} \min_{l \in [\![k, m-1]\!]} S_{l,m}.$$

In Appendix E, we show that $S_{l,m}$ scales as a strictly increasing function of $1/c_{l,m}$, being $c_{l,m}$ the *identification rate* of the pair $i_l^*$ and $i_m^*$, formally defined as:

$$c_{l,m} := f_{i_l^*, i_m^*}(\mu_{i_l^*, i_m^*} - \mu_{i_m^*, i_l^*})^2,$$

for every $l, m \in [\![n]\!]$, $l \neq m$, where $f_{i,j} := p_{i,j} + p_{j,i}$ is the *frequency* with which we observe a comparison between $i$ and $j$. The identification rate $c_{l,m}$ measures how easy it is to determine by direct inference the relative underlying ranking of items $i_l^*$ and $i_m^*$ from the observations. We can upper bound the sample complexity as the maximum of the expected samples to solve (P1) and (P2). Because of what we mentioned before, this is a strictly increasing function of the reciprocal of the *bottleneck identification rate*:

$$c_k^* := \min\{\underbrace{\min_{l \in [\![k-1]\!]} c_{l,l+1}}_{\text{Bottleneck for (P1)}}, \underbrace{\min_{m \in [\![k+1, n]\!]} \max_{l \in [\![k, m-1]\!]} c_{l,m}}_{\text{Bottleneck for (P2)}}\}.$$

The formalization of these arguments, which is deferred to Appendix E, allows to derive an upper bound to the sample complexity of the PIRATE algorithm.

**Theorem 4.2** (Sample Complexity Upper Bound). *For every confidence parameter $\delta \in (0, 1]$ and problem instance $\xi \in \mathcal{E}$:*

$$S_\xi^{(\delta)}(\text{PIRATE}) \leq \frac{128}{c_k^*} \ln\left(\frac{96n}{c_k^* \delta}\right).$$

As we show in Appendix E, Assumptions 3.1 and 3.2 guarantee $c_k^* > 0$, providing a finite upper bound to the sample complexity of the PIRATE algorithm.

### 4.3. Sample Complexity Lower Bound

In this part, we study the statistical complexity of the top-$k$ item learning under positional censoring problem by providing an instance-dependent lower bound to the sample complexity of every $\delta$-correct learner. In particular, we consider the worst-case sample complexity of the learner on the class of problem instances $\mathcal{E}(c_k^*)$ which share the same value $c_k^*$ for the *bottleneck identification rate*. The proof of the result is deferred to Appendix E.

**Theorem 4.3** (Sample Complexity Lower Bound). *For every $\delta$-correct learner Alg, number of items $n \in \mathbb{N}_{\geq 2}$, number of top items $k \in [\![n]\!]$, bottleneck identification rate $c_k^* \in \left(0, \frac{1}{6n^3}\right]$ and confidence parameter $\delta \in (0, 1/2)$, we have that:*

$$\sup_{\substack{\xi \in \mathcal{E}(c_k^*) \\ \text{s.t. } |\mathcal{I}| = n}} S_\xi^{(\delta)}(\text{Alg}) \geq \frac{1}{16 c_k^*} \ln\left(\frac{1}{4\delta}\right).$$

The sample complexity lower bound derived in Theorem 4.3 matches the sample complexity upper bound of PIRATE up to logarithmic dependencies, making PIRATE near-optimal.

### 4.4. Factorization of the Identification Rate

Despite its ability to capture the statistical complexity of the learning problem, the identification rate $c_{l,m}$ fails to distinguish the instance hardness caused by noise in the ranking model from that caused by the censorship mechanism.

In this part, we provide a lower bound on the identification rate $c_{l,m}$, expressed as a Frobenius product of an $n \times n$ matrix that depends solely on the ranking model $\rho$ and an $n \times n$ matrix that depends solely on the distribution of positional censorship mechanisms $\psi$. This yields a further upper bound on the sample complexity of the PIRATE algorithm, as it is non-increasing in $c_{l,m}$, showing how the sample complexity scales when we fix the observation model and we change the ranking model or vice-versa.

Now, we define the matrices involved in the factorization. Let $M_{\rho\,l,m} \coloneqq (\mathbb{P}_{\pi\sim\rho}\left[\pi(l) = r, \pi(m) = s\right])_{r,s\in[\![n]\!]}$ be the matrix of *second order marginals* for the item labels $l, m \in [\![n]\!]$. We define the *advantage gap* of item labels $l, m \in [\![n]\!]$ as: $\gamma_{\rho\,l,m} \coloneqq \mathbb{P}_{\pi\sim\rho}\left[\pi(l) < \pi(m)\right] - \mathbb{P}_{\pi\sim\rho}\left[\pi(m) < \pi(l)\right]$ and the *ideal censorship mechanism* for the pair $(i_l^*, i_m^*)$ as:

$$H_{\rho\,l,m} \coloneqq \gamma_{\rho\,l,m}[(2 - \gamma_{\rho\,l,m})\operatorname{triu}(M_{\rho\,l,m}) - (2 + \gamma_{\rho\,l,m})\operatorname{triu}(M_{\rho\,m,l})].$$

Let $P_\psi \coloneqq (\mathbb{P}_{\prec_{\mathrm{pos}}\sim\psi}\left[r \prec_{\mathrm{pos}} s\right])_{r,s\in[\![n]\!]}$ be the expected adjacency matrix of the positional censorship mechanism $\prec_{\mathrm{pos}}$ sampled from the distribution $\psi$. We have the required machinery for the factorization.

**Corollary 4.4** (Factorized Sample Complexity Upper Bound). *For every confidence parameter $\delta \in (0,1]$ and problem instance $\xi \in \mathcal{E}$:*

$$S_\xi^{(\delta)}(\text{PIRATE}) \leq \frac{128}{\tilde{c}_k^*} \ln\left(\frac{96n}{\tilde{c}_k^* \delta}\right),$$

*with:*

$$\tilde{c}_k^* \coloneqq \min\left\{\min_{l\in[\![k-1]\!]}\tilde{c}_{l,l+1}, \min_{m\in[\![k+1,n]\!]}\max_{l\in[\![k,m-1]\!]}\tilde{c}_{l,m}\right\},$$
$$\tilde{c}_{l,m} \coloneqq \max\left\{\langle H_{\rho\,l,m}, P_\psi\rangle_F, 0\right\}, \text{ for } l, m \in [\![n]\!], l \neq m.$$

We use the term *factorized identification rate* for $\tilde{c}_{l,m}$. The expression of $\tilde{c}_{l,m}$ in Corollary 4.4 justifies the name "ideal censorship mechanism" for $H_{\rho\,l,m}$: $\tilde{c}_{l,m}$ is maximized when $P_\psi$ is collinear with $H_{\rho\,l,m}$.

The following example provides an intuitive interpretation of the factorized identification rate as a scalar product, when $n = 3$ and the latent rankings are sampled from a Mallows $\phi$-model. This can be extended to the general case.

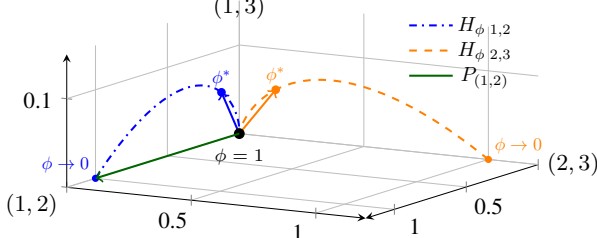

*(a)* Trajectories of the Ideal Censorship Mechanisms.

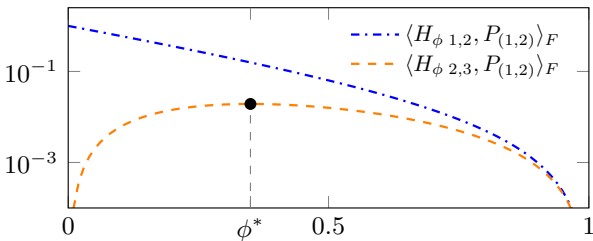

*(b)* Factorized Identification Rates.

*Figure 1.* Panel a displays the evolution of the above diagonal entries of the matrix $H_{\phi\,1,2}$ and $H_{\phi\,2,3}$ as a function of $\phi \in (0,1]$. Panel b depicts the evolution of the Frobenius products of $P_{(1,2)}$ with $H_{\phi\,1,2}$ and $H_{\phi\,2,3}$ w.r.t. $\phi \in (0,1]$, in logarithmic scale.

**Example 1.** Let $\mathcal{I} = \{A, B, C\}$, $\ell^*(A) = 1$, $\ell^*(B) = 2$, $\ell^*(C) = 3$. Assume that the ranking model is generated by a Mallows $\phi$-model with the Kendall metric: $\mathrm{M}_\phi[\pi] = C(\phi, T, e)\phi^{T(\pi,e)}$. We consider a fixed positional censorship mechanism: $\prec_{\mathrm{pos}}^{(1,2)} \coloneqq \{(1,2)\}$, i.e., each observation is a pairwise comparison which tells that the first item in the latent ranking precedes the second. In what follows, we use $\phi$ instead of $\rho$ and $(1,2)$ instead of $\psi$ in the subscripts, highlighting the chosen ranking and observation models. Consider the case $k = n = 3$, where the objective is to learn the full ranking. In Appendix E.3.1, we report the computations of the closed forms of all the quantities involved in this example: $P_{(1,2)}$, $H_{\phi\,1,2}$, and $H_{\phi\,2,3}$. Figure 1 allows to understand the evolution of the sample complexity as a function of the parameter $\phi$, which represents the dispersion from the modal ranking in the ranking model. In particular, in Figure 1a we represent the evolution of the above diagonal entries of $H_{\phi\,1,2}$ and $H_{\phi\,2,3}$ as a function of $\phi \in (0,1]$ in addition to $P_{(1,2)}$ which is constant w.r.t. $\phi$, while in Figure 1b, we depict the evolution of the factorized identification rates in logarithmic scale. Observe that, by construction, $P_{(1,2)}$, $H_{\phi\,1,2}$ and $H_{\phi\,2,3}$ are strictly upper triangular, so that Figure 1a represent the values of all their non-null entries. Recall that, when the objective is to learn the full ranking, the sample complexity scales as $1/\min_{l\in[\![n-1]\!]}\tilde{c}_{l,l+1}$. When $\phi \to 0$, i.e., in the deterministic case, $P_{(1,2)}$ is perfectly aligned with $H_{\phi\,(1,2)}$ and orthogonal to $H_{\phi\,(2,3)}$. This matches the intuition: when the sampled permutation is the identity, each item keeps its

*Table 2.* We provide the minimax sample complexity of the corresponding learning setting, specified by the learning objective (*rows*) and the observation model (*columns*). In particular, we reference a sample complexity Lower Bound (LB) and show that the PIRATE algorithm attains such a rate by casting the general Upper Bound (UB) provided in Corollary 4.4 to the specific setting. We use $\widetilde{\Theta}(\cdot)$ to represent both UBs ($\widetilde{\mathcal{O}}(\cdot)$) and LBs ($\Omega(\cdot)$) suppressing logarithmic dependencies other than $\log(1/\delta)$.

| Learning Objective | | Full-Ranking | Winner | Top-$h$ |
|---|---|:---:|:---:|:---:|
| **Learning the Full Ranking** | Rate | $\widetilde{\Theta}\left(\dfrac{1}{\min\limits_{l\in[\![n-1]\!]}\Delta_{l,l+1}^2}\log\left(\dfrac{1}{\delta}\right)\right)$ | $\widetilde{\Theta}\left(\dfrac{n}{\min\limits_{l\in[\![n-1]\!]}\Delta_{l,l+1}^2}\log\left(\dfrac{1}{\delta}\right)\right)$ | $\widetilde{\Theta}\left(\dfrac{n}{h\min\limits_{l\in[\![n-1]\!]}\Delta_{l,l+1}^2}\log\left(\dfrac{1}{\delta}\right)\right)$ |
| | UB | PIRATE, Cor. 4.4 ($k=n$) + Lem. E.10 | PIRATE, Cor. 4.4 ($k=n$) + Lem. E.9 | PIRATE, Cor. 4.4 ($k=n$) + Lem. E.8 |
| | LB | Saha & Gopalan, 2019a, Thm. 15 ($m=n$) | Saha & Gopalan, 2019a, Thm. 12 | Saha & Gopalan, 2019a, Thm. 15 |
| **Learning the Best Item** | Rate | $\widetilde{\Theta}\left(\dfrac{1}{\Delta_{1,2}^2}\log\left(\dfrac{1}{\delta}\right)\right)$ | $\widetilde{\Theta}\left(\dfrac{n}{\Delta_{1,2}^2}\log\left(\dfrac{1}{\delta}\right)\right)$ | $\widetilde{\Theta}\left(\dfrac{n}{h\Delta_{1,2}^2}\log\left(\dfrac{1}{\delta}\right)\right)$ |
| | UB | PIRATE, Cor. 4.4 ($k=1$) + Lem. E.10 | PIRATE, Cor. 4.4 ($k=1$) + Lem. E.9 | PIRATE, Cor. 4.4 ($k=1$) + Lem. E.8 |
| | LB | Saha & Gopalan, 2019b, Thm. 10 ($m=n$) | Saha & Gopalan, 2019b, Thm. 2 | Saha & Gopalan, 2019b, Thm. 10 |

place, thus, $\prec_{\mathrm{pos}}^{(1,2)}$ is excellent to infer that $A \prec B$, but does not provide any comparison of $A$ or $B$ with item $C$, making it impossible to learn the full ranking. As $\phi$ increases, both $H_{\phi\,1,2}$ and $H_{\phi\,2,3}$ tend towards $I_{3,<}$, but their norm goes to $0$. From the learning point of view, the alignment towards $I_{3,<}$ is beneficial, as the angle between $P_{(1,2)}$ and $H_{\phi\,2,3}$ decreases, while the fact that the norm shrinks is detrimental as it is proportional to the factorized identification rate. Initially, the effect of alignment towards $I_{3,<}$ outweighs the norm reductions, decreasing the sample complexity until the value $\phi^*$; conversely, beyond this point, the norm reduction becomes dominant up to the limit $\phi = 1$. When $\phi = 1$, the distribution over latent rankings is uniform, rendering learning impossible. This is captured by the fact that both the factorized identification rates are equal to $0$. It is interesting to note that the optimal amount of noise to facilitate learning is $\phi^* > 0$, instead of the deterministic case $\phi \to 0$.

## 5. Near-Optimality of the PIRATE Algorithm in Restricted Settings

In this section, we show that, by casting the factorized sample complexity upper bound (Corollary 4.4), PIRATE enjoys a near-optimal sample complexity even in restricted settings which are well-established in the literature. In particular, we consider instances where the latent ranking is generated according to the PL model and the observation model is one among full-ranking feedback, winner feedback, or top-$h$ feedback (Saha & Gopalan, 2019a). Let $\boldsymbol{\theta} := (\theta_l)_{l\in[\![n]\!]}$ be the parameters of the PL ranking model

with $1 = \theta_1 > \cdots > \theta_n \geq 1/2$. Assuming that the parameters are bounded away from $0$ is equivalent to assume that the underlying utilities $u_l = \ln(\theta_l)$ associated to the PL model (see Section 2.2) are bounded below. This is analogous to the *ratios bounded by a constant* assumption (Ren et al., 2018). For ease of notation, we choose the value $1/2$ as the parameters lower bound. We denote by $\Delta_{l,m} := \theta_l - \theta_m$ for every $l, m \in [\![n]\!]$, $l < m$ the parameter differences of the PL model. Table 2 reports the minimax sample complexity rates for each of the settings, referencing a Lower Bound (LB) in the literature and the corresponding near-optimal Upper Bound (UB) to the sample complexity of the PIRATE algorithm.

**Remark 5.1.** The lower bounds referenced in Table 2 are established for the $(\epsilon, \delta)$-PAC setting. In that framework, the goal is to return a solution that is $\epsilon$-close to the optimal one. Our PExact setting can be viewed as a specific instance of the PAC setting where the approximation parameter $\epsilon$ is sufficiently small to distinguish the optimal solution from the second-best candidate. Specifically, for the full-ranking objective, Saha & Gopalan (2019a) define an $\epsilon$-best ranking as a permutation $\sigma$ such that for any pair of item labels $l, m$ with $\sigma(l) < \sigma(m)$, the parameters satisfy $\theta_l \geq \theta_m - \epsilon$ (Saha & Gopalan, 2019a, Definition 2). To ensure the recovered ranking is exactly correct with high probability, we require $\epsilon < \min_{l\in[\![n-1]\!]} \Delta_{l,l+1}$. For the best-item objective, Saha & Gopalan (2019b) define an $\epsilon$-best item as any item with label $l$ such that $\theta_l \geq \theta_1 - \epsilon$. To identify the unique best item with high probability, the tolerance must be smaller than the gap between the best and the second-best item, i.e.,

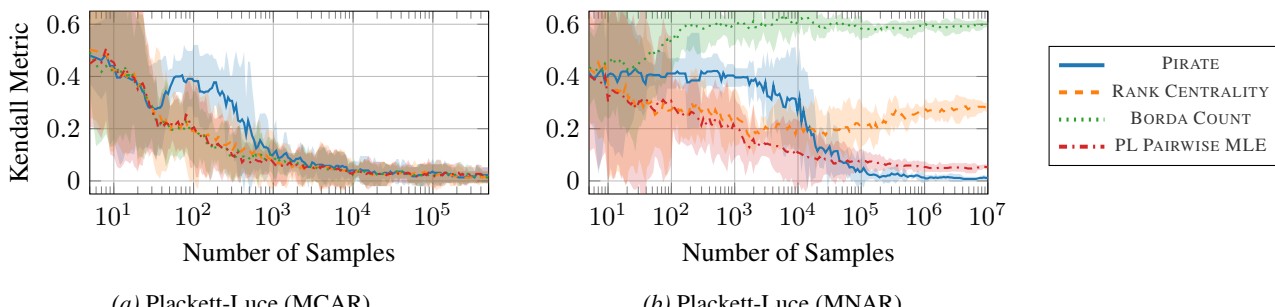

*(a)* Plackett-Luce (MCAR).  *(b)* Plackett-Luce (MNAR).

*Figure 2.* Comparison of PIRATE against the baselines under MCAR and MNAR missing comparisons respectively, when the latent ranking is generated according to a Plackett-Luce distribution. Shaded regions represent the 90% confidence intervals over 5 runs.

$\epsilon < \Delta_{1,2}$. By substituting these instance-dependent gaps for $\epsilon$ in the PAC lower bounds, we obtain the correct rates for the exact setting. Finally, the framework considered in (Saha & Gopalan, 2019a;b) is *active*: the learner can ask a comparison among a subset of the items $\mathcal{S} \subseteq \mathcal{I}$ with fixed cardinality $|\mathcal{S}|$. The passive setting can be viewed as a special case of the active setting where $|\mathcal{S}| = n$. In Appendix E.4, we show how it is possible to formally derive the PExact lower bounds with the stated rates.

## 6. Numerical Simulations

In this section, we present the results of numerical simulations of PIRATE against state-of-the-art LTR algorithms.[4]

**Baselines.** We consider the following baseline algorithms: RANK CENTRALITY (Negahban et al., 2017), a non-parametric LTR algorithm which creates a Markov chain on the set of items and estimates the ranking from the corresponding stationary distribution; BORDA COUNT (Ammar & Shah, 2011), which ranks each item according to the ratio between the number of times it precedes any other item and the total number of comparisons in which it is involved (see Appendix C), and, finally, PL PAIRWISE MLE which relies on the maximum likelihood estimate of the PL parameters from a dataset of pairwise comparisons—under the MCAR assumption—and ranks the items accordingly.

**Setting.** For the sake of fairness in the comparison of PIRATE against the baselines, we consider the *fixed budget* (Kaufmann et al., 2016) version of the algorithm in which we modify Line 8 so that $\hat{\mu}_{i,j} > \hat{\mu}_{j,i}$ is sufficient to add an edge to $E^{(\delta)}$, and we remove the `if` statement at Line 12. In Figure 2, we compare the baselines in terms of average normalized Kendall Metric (see Section 2) between the estimated ranking and the true one. In both simulations, the latent ranking is sampled from a PL distribution with 10 items for Figure 2a and 15 items for Figure 2b. The

---

[4]Additional experiments are reported in Appendix G. The code to reproduce the results is available at https://github.com/m1gwings/pirate-experiments.

parameters are sampled uniformly in $(0, 1)$. In Figure 2a, the algorithms observe a comparison among a pair of items chosen uniformly at random (MCAR). In Figure 2b, instead, the observation consists in a strict partial order among items obtained through the following positional censoring: the first item in the latent ranking precedes the second which precedes the third and the third last item precedes the second last which precedes the last. This filter arises naturally in real-world scenarios, as users are more prone to provide feedback over items that they like very much or that they dislike.

**Results.** In Figure 2a, PIRATE obtains a comparable performance w.r.t. the baselines which are designed for this restricted setting (i.e., PL latent ranking with MCAR comparisons), showing that the great level of generality at which PIRATE operates does not affect its performance on easier instances. In Figure 2b, PIRATE is the only algorithm which learns the true underlying ranking, showing that, even when the latent ranking is generated according to a well-behaved distribution like PL, the MNAR bias is sufficient to make the baselines fail, showing that these assumptions are needed for the algorithms in order to converge.

## 7. Conclusions

We studied the problem of LTR alternatives from the observation of incomplete rankings when the comparisons are MNAR. We defined the positional censoring feedback model for MNAR incomplete rankings, which generalizes many of the established models. Then, we formulated the problem of top-$k$ item learning under positional censoring. We designed the PIRATE algorithm and provided a near-optimal sample complexity bound, proving that it is possible to learn from incomplete rankings even when the rigid customary assumptions on the ranking and feedback model are relaxed. Finally, we showed that PIRATE enjoys a near-optimal sample complexity in the restricted setting with PL ranking model and winner or top-$h$ feedback. These results positively answer our research question.

## Acknowledgements

This publication was funded with the contribution of Ministero dell'Università e della Ricerca pursuant to D.D. n. 7206 of 17 April 2025 – BANDO FIS 2. Project FIS-2023-02598 (Starting Grant), title: "Unified Learning from Diverse Human Feedback" (HUmLrn). CUP: D53C25000710001.

## Impact Statement

This paper presents work whose goal is to advance the field of machine learning. There are many potential societal consequences of our work, none of which we feel must be specifically highlighted here.

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

# A. Related Works

In this appendix, we describe previous works which address the problem of learning to rank from incomplete rankings, with a particular focus on the assumptions regarding the ranking model which generates the latent ranking and the feedback type which specifies the comparisons that remain unobserved. A taxonomy of previous works according to these properties is reported in Table 3.

*Table 3.* Taxonomy of existing works according to the ranking model (*rows*) and the feedback model (*columns*). The table is split into two parts for better readability.

| Ranking Model | Pairwise Comparison | | | Winner Feedback |
|---|---|---|---|---|
| | Active | MCAR | MNAR | |
| Plackett-Luce | (Szörényi et al., 2015) | (Negahban et al., 2017) | (Soufiani et al., 2014) | (Saha & Gopalan, 2019a;b; 2020) |
| Mallows | (Busa-Fekete et al., 2014) | (Lu & Boutilier, 2014) | — | (Vitelli et al., 2018) |

| Ranking Model | Top-$h$ Feedback | Rank-dep. Coarsening | Subset Ranking |
|---|---|---|---|
| Plackett-Luce | (Saha & Gopalan, 2019a;b; 2020) | (Fahandar et al., 2017) | (Khetan & Oh, 2016; Jang et al., 2017) |
| Mallows | (Chierichetti et al., 2018) | — | (Vitelli et al., 2018) |

**Pairwise Comparisons (Active).** Szörényi et al. (2015) consider the PAC (Probably Approximately Correct) problem of identifying a near-best item under the PL ranking model by querying pairwise comparisons. They develop a budgeted version of the *Quick Sort* algorithm which they apply to the setting of noisy comparisons. Busa-Fekete et al. (2014) work with the same type of feedback, but with a different ranking model: a Mallows $\phi$-model. Their goal is to learn some of the properties of the ranking model, like the modal ranking or the most preferred item. In (Braverman & Mossel, 2008), the outcome of $\binom{n}{2}$ noisy comparisons among items is sampled independently once. The learner can query each of these outcomes (which is not resampled) and needs to aggregate them into a ranking.

**Pairwise Comparisons (Passive).** Negahban et al. (2017) design the RANK CENTRALITY algorithm which aggregates a dataset of pairwise comparisons among items, which need not be chosen by the learner. Their analysis relies on the fact that the learner observes a comparison for a pair of alternatives which does not depend on their positions in the latent ranking, which is generated according to the PL model. In this setting, the outcomes of pairwise comparisons follow the well-known Bradley-Terry-Luce model (BTL, Bradley & Terry, 1952; Luce, 1959). Rank Centrality provides a non-parametric approach to rank-aggregation in the PL setting, by computing the stationary distribution of a Markov chain on the set of alternatives to rank, where transition probabilities are proportional to the probability that an alternative is preferred to the other. Lu & Boutilier (2014) provide efficient methods to sample the outcomes of pairwise comparisons when the latent ranking is generated by a Mallows $\phi$-model. They exploit such sampling techniques in a *monte carlo expectation maximization* algorithm which enables learning mixtures of Mallows $\phi$-models by observing MCAR pairwise comparisons among the alternatives.

**Winner and Top-$h$ Feedback.** Saha & Gopalan (2019a;b) study the PAC problem of identifying a near-best ranking of the items and a near-best item. They consider the active setting where the learner can query the subset of alternatives that the human has to compare. In particular, they assume that a latent ranking of the subset of alternatives is generated according to the PL model. They design different learners for different types of feedback: winner feedback, where the learner observes the best alternative in the latent ranking among those in the subset, and the more general top-$h$ feedback, where the learner observes the ordered collection of the top-$h$ alternatives. They provide an upper bound to the sample complexity of their learner and a matching lower bound. In particular, they show that top-$h$ feedback allows reducing the sample complexity by a factor $1/h$. Chierichetti et al. (2018) introduce a generalization of the Mallows model specifically for top-$h$ feedback. They define a tractable distance measure for top-$h$ lists and provide efficient algorithms for sampling and learning (reconstructing) the model parameters from such incomplete data.

**Rank-dependent Coarsening.** Fahandar et al. (2017) present the *rank-dependent coarsening* feedback model, which

allows for MNAR comparisons among the alternatives. In particular, they assume that a latent ranking among the alternatives is generated according to the PL model, but then only an incomplete ranking is observed, involving the alternatives which occupy a certain subset of the positions in the latent ranking. They study the problem when the incomplete rankings have length 2 (i.e., pairwise comparisons), showing that even if the coarsening mechanism depends on ranks (and not items), established algorithms are still able to learn the correct ranking.

**Subset Ranking and Rank Breaking.** Khetan & Oh (2016) study the problem of estimating the parameters of the PL model from incomplete rankings provided by different individuals, through a *rank-breaking* estimator. The idea of rank breaking consists of the disaggregation of incomplete rankings into a collection of pairwise comparisons which are processed independently. They provide a high-probability bound on the estimation error of the rank-breaking estimator which depends on the topology of the collected data, when represented in a comparison graph built by aggregating the pairwise comparisons. Soufiani et al. (2014) consider a dataset of MNAR pairwise comparisons generated from the *rank-breaking* of incomplete rankings obtained by censoring some of the comparisons among the alternatives according to their position in the latent ranking. The goal is to show that, even if the pairwise comparisons are correlated, since some are obtained from the same incomplete ranking, it is still possible to process them as independent and maintain consistency under some assumptions on the MNAR censorship mechanisms. This allows to reduce the computational load of the estimation, as working with pairwise comparisons requires lighter methods. Jang et al. (2017) adapt the RANK CENTRALITY algorithm from pairwise comparisons to $h$-wise rankings, generated according to the PL model. In particular, they break $h$-wise rankings into pairwise comparisons after a circular permutation of the data which is sampled uniformly at random. They show that $h$-wise ranking data improves the convergence of the estimator by a factor $1/\sqrt{h}$. Finally, Vitelli et al. (2018) propose a comprehensive Bayesian framework for the Mallows model that handles various forms of incomplete data, including pairwise comparisons and subset rankings. By employing MCMC methods, they estimate the posterior distribution of the consensus ranking and the spread parameter, allowing for the aggregation of partial preferences even in the presence of noise or user heterogeneity.

# B. Supplement to Preliminaries

## B.1. Additional Notation

In this appendix, we introduce additional notation, required for the analysis.

For a set $\mathcal{X}$, we denote by $\mathcal{X}^{\mathbb{N}}$ the set of *sequences* with values in $\mathcal{X}$. We use $\mathcal{P}(\mathcal{X})$ for the *power set* of $\mathcal{X}$.

We denote by:

- $\mathrm{Gumbel}(u, 1) \in \Delta(\mathbb{R})$ the probability measure with CDF $F(x) = \exp(-\exp(-(x - u)))$.
- $\mathrm{Bin}(N, p) \in \Delta(\llbracket 0, N \rrbracket)$ with $N \in \mathbb{N}_{\geq 1}$ and $p \in [0, 1]$ the Binomial distribution.
- $\mathrm{Be}(p) \in \Delta(\{0, 1\})$ with $p \in [0, 1]$ the Bernoulli distribution.
- $\lambda$ the standard Lebesgue measure on $\mathbb{R}$.

Given two measures $\nu$ and $\mu$, we write $\nu \ll \mu$ to denote that $\nu$ is absolutely continuous w.r.t. $\mu$, and we use $\nu \otimes \mu$ to denote the product of measures. $\nu^{\otimes \mathbb{N}}$ denotes the infinite product of a measure. We adopt an analogous notation for the product and infinite product of $\sigma$-algebras.

Given two measures $\nu \ll \mu$, we denote by $\frac{d\nu}{d\mu}$ the *Radon-Nikodym derivative* of $\nu$ w.r.t. $\mu$ and by $D_{\mathrm{KL}}(\nu \| \mu) = \int \ln\left(\frac{d\nu}{d\mu}\right) d\nu$ their *Kullback-Leibler divergence* (Kullback & Leibler, 1951).

We denote by $I_n$ the identity matrix of $\mathbb{R}^{n \times n}$. Given a matrix $A \in \mathbb{R}^{n \times m}$, we denote by $\mathrm{vec}(A)$ its vectorization. We denote by $\mathrm{mat}_{n,m}$ the inverse operation such that $\mathrm{mat}_{n,m}(\mathrm{vec}(A)) = A$. We use the shorthand $\mathrm{mat}$ to denote $\mathrm{mat}_{n,n}$. We denote by $A \otimes B$ the Kronecker product of $A$ and a matrix $B$. We denote by $T_{n,m}$ the matrix such that $T_{n,m} \mathrm{vec}(A) = \mathrm{vec}(A^\top)$. We use $\odot$ to denote the element-wise product between matrices. For matrices $A, B$ of the same dimension, we denote by $\frac{A}{B}$ their element-wise ratio. Finally, we denote by $A^{\odot 2} = A \odot A$ the element-wise square of a matrix $A$.

Given a collection $\boldsymbol{\theta} = (\theta_l)_{l \in \llbracket n \rrbracket}$, $\theta_1 > \cdots > \theta_n > 0$, we denote by $\mathrm{PL}_{\boldsymbol{\theta}} \in \Delta(S_n)$ the PL model with parameters $\boldsymbol{\theta}$. That is, in accordance with Equation (1):

$$\mathrm{PL}_{\boldsymbol{\theta}}(\pi) = \prod_{r \in \llbracket n-1 \rrbracket} \frac{\theta_{\pi^{-1}(r)}}{\sum_{s \in \llbracket r, n \rrbracket} \theta_{\pi^{-1}(s)}} \text{ for every } \pi \in S_n.$$

We denote by $\Theta = \sum_{l \in [\![n]\!]} \theta_l$.

### B.2. Supplement to Ranking Models

**Mallows $\phi$-Models.** In addition to Kendall distance, several other metrics are commonly employed within Mallows $\phi$-models:

- *Spearman metric*:

$$R(\pi, \sigma) = \sqrt{\sum_{l \in [\![n]\!]} (\pi(l) - \sigma(l))^2}.$$

- *Spearman footrule metric*:

$$F(\pi, \sigma) = \sum_{l \in [\![n]\!]} |\pi(l) - \sigma(l)|.$$

- *Hamming metric*:

$$H(\pi, \sigma) = \sum_{l=1}^{n} \mathbb{I}[\pi(l) \neq \sigma(l)].$$

The availability of different metrics makes Mallows $\phi$-models very flexible to describe ranking data. These models are specifically tailored for the identification of an underlying ranking among items from noisy observations.

**Babington-Smith Models.** The *Babington-Smith* model proposes a method for inducing a probability distribution over the set of permutations $S_n$ from a set of independent paired comparison probabilities (Critchlow et al., 1991). Let $p_{l,m} \in [0, 1]$ denote the probability that item $\ell^{-1}(l)$ is preferred to item $\ell^{-1}(m)$ in a pairwise comparison, such that $p_{m,l} = 1 - p_{l,m}$ for every $l, m \in [\![n]\!]$. The model conceptually assumes a tournament where all pairs are compared independently; if the outcomes form a strict ordering (containing no circular triads), that ranking is observed. The resulting probability mass function is defined as:

$$\rho(\pi) \propto \prod_{\substack{l,m \in [\![n]\!] \\ \pi(l) < \pi(m)}} p_{l,m} \text{ for every } \pi \in S_n,$$

where the proportionality constant ensures the probabilities sum to 1. This framework connects the properties of the paired comparison probabilities $\{p_{l,m}\}$ to the resulting ranking model; for instance, if the probabilities are strongly stochastically transitive, $\rho$ satisfies the complete consensus property (Critchlow et al., 1991).

## C. Failure Cases of Standard Approaches

In this appendix, we construct instances of the top-$k$ item learning under positional censoring problem, in which some of the established approaches in the incomplete rankings literature fail to identify the correct ranking among the items.

### C.1. Pivoting

Some algorithms in the incomplete rankings literature (e.g. Saha & Gopalan, 2019a, BEAT-THE-PIVOT) rely on the concept of *pivoting*, i.e., they choose a pivot item $b \in \mathcal{I}$ and sort the other item $i \in \mathcal{I} \setminus \{b\}$ according to $\mu_{i,b}$. Remember that $\mu_{i,b}$, defined in Section 4.1, is the probability that $i$ precedes $b$ in the observed incomplete ranking, conditioned on the fact that a comparison among the two items is observed. Intuitively, $\mu_{i,b}$ tells how strongly item $i$ is preferred to $b$, so that one could think that if item $i$ is preferred to $b$ more strongly than item $j$ is preferred to $b$, then item $i$ is preferred to item $j$. This line of reasoning can fail catastrophically outside of the PL ranking model, when positional censoring is applied. For simplicity let $[\![3]\!]$ be the set of items and the identity permutation $e$ be the labeling function. Consider the following Babington-Smith ranking model (Critchlow et al., 1991), defined in terms of the pairwise comparison probabilities $p_{1,3} = 0.6$, $p_{1,2} = p_{2,3} = 0.7$. It is straightforward to check by direct inspection that such a model satisfies both Assumption 3.1 and 3.2. Let the distribution over positional censorship mechanisms be a Dirac delta distribution over $\prec_{\text{pos}}^{(1,2)} = \{(1,2)\}$: $\delta(\prec_{\text{pos}}^{(1,2)})$. In this instance, we have:

$$p_{2,1} \propto (1 - p_{1,2}) p_{2,3} p_{1,3},$$
$$p_{1,2} \propto p_{1,2} p_{1,3} p_{2,3},$$

$$p_{3,1} \propto (1 - p_{1,3})(1 - p_{2,3})p_{1,2},$$
$$p_{1,3} \propto p_{1,3}p_{1,2}(1 - p_{2,3}),$$

where $\propto$ hides the normalization constant of the Babington-Smith model. Then:

$$\mu_{2,1} = \frac{(1 - p_{1,2})p_{2,3}p_{1,3}}{(1 - p_{1,2} + p_{1,2})p_{1,3}p_{2,3}} = 1 - p_{1,2},$$

$$\mu_{3,1} = \frac{(1 - p_{1,3})(1 - p_{2,3})p_{1,2}}{(1 - p_{1,3} + p_{1,3})(1 - p_{2,3})p_{1,2}} = 1 - p_{1,3}.$$

Thus, $\mu_{2,1} < \mu_{3,1}$, implying that, by pivoting on item 1, we would learn that $3 \prec 2$, which is incorrect.

## C.2. RANK CENTRALITY

The RANK CENTRALITY algorithm (Negahban et al., 2017) creates a Markov chain on the set of items $\mathcal{I}$, where the transition probability from item $i$ to item $j$ is proportional to the probability that $j$ beats $i$ in a pairwise comparison. The final ranking is determined by the stationary distribution $\eta$ of this Markov chain, based on the intuition that stronger items will have higher stationary probabilities.

Formally, the transition matrix $P$ is defined such that $P_{i,j} \propto \mu_{j,i}$ for $i \neq j$. We consider an instance where the underlying ranking is $1 \prec 2 \prec 3$, and item 1 is strongly preferred over both items 2 and 3. Consequently, the stationary mass accumulates significantly on item 1, i.e., $\eta_1 \approx 1$.

We can inspect the relative ordering of items 2 and 3 by analyzing the flow of probability mass from the dominant item 1. The stationary condition implies that for any item $i \in \{2, 3\}$, the probability mass $\eta_i$ is roughly proportional to the incoming flow from the dominant node:

$$\eta_i \propto \eta_1 P_{1,i} + \sum_{j \in \{2,3\}} \eta_j P_{j,i} \approx \eta_1 P_{1,i}.$$

Since the transition probability $P_{1,i}$ is proportional to $\mu_{i,1}$, we have:

$$\frac{\eta_3}{\eta_2} \approx \frac{P_{1,3}}{P_{1,2}} = \frac{\mu_{3,1}}{\mu_{2,1}}.$$

Thus, if $\mu_{2,1} < \mu_{3,1}$ as in Section C.1, we obtain $\eta_3 > \eta_2$. This indicates that RANK CENTRALITY would return the ranking $3 \succ 2$, which contradicts the ground truth preference.

## C.3. Expected Borda Count

The *expected Borda count*, which is estimated in many algorithms in the learning to rank literature (e.g., Saha & Gopalan, 2019a, SCORE-AND-RANK), is defined as follows:

$$\text{ExpBorda}(i) := \frac{\mathbb{P}_\xi \left[ i \prec_{\text{obs}}^{(1)} j \text{ for some } j \in \mathcal{I} \setminus \{i\} \right]}{\mathbb{P}_\xi \left[ i \prec_{\text{obs}}^{(1)} j \text{ or } j \prec_{\text{obs}}^{(1)} i \text{ for some } j \in \mathcal{I} \setminus \{i\} \right]}.$$

Let us work under the PL model with $\theta_1 = 1/2$, $\theta_2 = 1/2 - \epsilon$, $\theta_3 = \epsilon(1 - \epsilon)$, $\theta_4 = \epsilon^2$. Let $\mathcal{I} = [\![4]\!]$ and consider the identity permutation $e$ as the labeling function. Consider a fixed positional censorship mechanism: $\prec_{\text{pos}} = \{(1, 2), (3, 4)\}$. First of all observe that: $\mathbb{P}_\xi \left[ i \prec_{\text{obs}}^{(1)} j \text{ or } j \prec_{\text{obs}}^{(1)} i \text{ for some } j \in \mathcal{I} \setminus \{i\} \right] = 1$ and $\mathbb{P}_\xi \left[ i \prec_{\text{obs}}^{(1)} j \text{ for some } j \in \mathcal{I} \setminus \{i\} \right] = \text{PL}_{\boldsymbol{\theta}} \left[ \pi(i) = 1 \right] + \text{PL}_{\boldsymbol{\theta}} \left[ \pi(i) = 3 \right]$ for every $i \in \mathcal{I}$. Thanks to Equation (1), it is possible to show that:

$$\text{ExpBorda}(1) \to \frac{1}{2},$$
$$\text{ExpBorda}(3) \to 1,$$

as $\epsilon \to 0$. Thus, for $\epsilon$ sufficiently small, the expected Borda count estimator misranks 1 and 3.

## D. Non-Learnability Results

In this appendix we prove the non-learnability results stated in Section 3.

The next result shows that there exists a ranking model with modal ranking equal to the identity permutation, that does not satisfy the complete consensus property (Definition 2.1), making two instances with different underlying rankings not distinguishable one from the other.

**Theorem D.1** (Non-Learnability without Complete Consensus of the Ranking Model). *There exist two instances of the top-$k$ item learning under positional censoring problem $\xi_1^{no\text{-}cc} = (\mathcal{I}, \rho_{no\text{-}cc}, \psi, \ell_1^*, k)$, $\xi_2^{no\text{-}cc} = (\mathcal{I}, \rho_{no\text{-}cc}, \psi, \ell_2^*, k)$ with the same item set $\mathcal{I}$ and ranking model $\rho_{no\text{-}cc}$ with modal ranking equal to the identity permutation $e$, that does not satisfy the complete consensus property (Definition 2.1) and produces the same distribution over the observations induced by the instances, despite having $\ell_1^{*-1}(2) \neq \ell_2^{*-1}(2)$.*

*Proof.* Let $\rho_{no\text{-}cc} = C(\lambda) \exp(-\lambda H(\pi, e))$ with $C(\lambda)^{-1} = \sum_{\pi \in S_n} \exp(-\lambda H(\pi, e))$ and $\lambda = -\log \phi$ be a Mallows $\phi$-model (see Section 2.2) with the Hamming metric. Critchlow et al. (1991) show that a Mallows $\phi$-model satisfies complete consensus if and only if its metric $d$ possesses the *transposition property*, which is not satisfied by the Hamming metric. Let the set of items be $\mathcal{I} = [\![3]\!]$, the distribution over positional censorship mechanisms be $\psi = \delta(\prec_{\text{pos}}^{\text{win}})$, i.e. a Dirac delta distribution over winner feedback. Finally, let $\ell_1^* = e$ be the identity labeling function and $\ell_2^* = \tau_{2,3}$ be the labeling function which assigns label 2 to item 3 and vice-versa. Now, since we are dealing with winner feedback, there is a bijective correspondence between the observations and the set of items $\mathcal{I}$: each observation is univocally identified by the winner item. Thus, to compute the distribution over observations in both instances, we can simply consider the probability that each item wins. Let us start with $\xi_1^{no\text{-}cc}$:

$$
\begin{aligned}
\mathbb{P}_{\xi_1^{no\text{-}cc}}\left[1 \prec_{\text{obs}}^{(1)} 2, 1 \prec_{\text{obs}}^{(1)} 3\right] &= \rho_{no\text{-}cc}\left[\pi(1)=1, \pi(2)=2, \pi(3)=3\right] + \rho_{no\text{-}cc}\left[\pi(1)=1, \pi(2)=3, \pi(3)=2\right] \\
&= C(\lambda)\left(1 + \exp(-2\lambda)\right), \\
\mathbb{P}_{\xi_1^{no\text{-}cc}}\left[2 \prec_{\text{obs}}^{(1)} 1, 2 \prec_{\text{obs}}^{(1)} 3\right] &= \rho_{no\text{-}cc}\left[\pi(1)=2, \pi(2)=1, \pi(3)=3\right] + \rho_{no\text{-}cc}\left[\pi(1)=3, \pi(2)=1, \pi(3)=2\right] \\
&= C(\lambda)\left(\exp(-2\lambda) + \exp(-3\lambda)\right), \\
\mathbb{P}_{\xi_1^{no\text{-}cc}}\left[3 \prec_{\text{obs}}^{(1)} 1, 3 \prec_{\text{obs}}^{(1)} 2\right] &= \rho_{no\text{-}cc}\left[\pi(1)=2, \pi(2)=3, \pi(3)=1\right] + \rho_{no\text{-}cc}\left[\pi(1)=3, \pi(2)=2, \pi(3)=1\right] \\
&= C(\lambda)\left(\exp(-3\lambda) + \exp(-2\lambda)\right).
\end{aligned}
$$

Analogously:

$$
\begin{aligned}
\mathbb{P}_{\xi_2^{no\text{-}cc}}\left[1 \prec_{\text{obs}}^{(1)} 2, 1 \prec_{\text{obs}}^{(1)} 3\right] &= \rho_{no\text{-}cc}\left[\pi(1)=1, \pi(2)=2, \pi(3)=3\right] + \rho_{no\text{-}cc}\left[\pi(1)=1, \pi(2)=3, \pi(3)=2\right] \\
&= C(\lambda)\left(1 + \exp(-2\lambda)\right), \\
\mathbb{P}_{\xi_2^{no\text{-}cc}}\left[2 \prec_{\text{obs}}^{(1)} 1, 2 \prec_{\text{obs}}^{(1)} 3\right] &= \rho_{no\text{-}cc}\left[\pi(1)=2, \pi(2)=3, \pi(3)=1\right] + \rho_{no\text{-}cc}\left[\pi(1)=3, \pi(2)=2, \pi(3)=1\right] \\
&= C(\lambda)\left(\exp(-3\lambda) + \exp(-2\lambda)\right), \\
\mathbb{P}_{\xi_2^{no\text{-}cc}}\left[3 \prec_{\text{obs}}^{(1)} 1, 3 \prec_{\text{obs}}^{(1)} 2\right] &= \rho_{no\text{-}cc}\left[\pi(1)=2, \pi(2)=1, \pi(3)=3\right] + \rho_{no\text{-}cc}\left[\pi(1)=3, \pi(2)=1, \pi(3)=2\right] \\
&= C(\lambda)\left(\exp(-2\lambda) + \exp(-3\lambda)\right).
\end{aligned}
$$

It is evident by direct inspection that the two probability distributions over observations are identical. $\square$

The next result shows that there exists a ranking model with modal ranking equal to the identity permutation that does not satisfy the full support property, making two instances with different underlying rankings not distinguishable one from the other.

**Theorem D.2** (Non-Learnability without Full Support of the Ranking Model). *There exist two instances of the top-$k$ item learning under positional censoring problem $\xi_1^{no\text{-}fs} = (\mathcal{I}, \rho_{no\text{-}fs}, \psi, \ell_1^*, k)$, $\xi_2^{no\text{-}fs} = (\mathcal{I}, \rho_{no\text{-}fs}, \psi, \ell_2^*, k)$ with the same item set $\mathcal{I}$ and ranking model $\rho_{no\text{-}fs}$ with modal ranking equal to the identity permutation $e$ that does not have full support and produces the same distributions over the observations induced by the instances, despite having $\ell_1^{*-1}(2) \neq \ell_2^{*-1}(2)$.*

*Proof.* Let $\rho_{\text{no-fs}} = \delta(e)$ be a Dirac delta distribution over the identity permutation $e$. Let the set of items be $\mathcal{I} = [\![3]\!]$, the distribution over positional censorship mechanisms be $\delta(\prec_{\text{pos}}^{\text{win}})$. Finally, let $\ell_1^* = e$ be the identity labeling function and $\ell_2^* = \tau_{2,3}$ be the labeling function which assigns label 2 to item 3 and vice-versa. Observe that in both instances we always observe item 1 as the winner, making the relative underlying ranking of items 2 and 3 impossible to learn. $\qquad\square$

# E. Analysis of the $\delta$-Correctness and Near-Optimality of the PIRATE Algorithm

In this appendix, we provide the proofs of the results stated in Section 4.

## E.1. $\delta$-Correctness and Sample Complexity Upper Bound

In this part, we provide the proofs of the results stated in Sections 4.1 and 4.2. We rely on the additional notation introduced in Appendix B.1.

### E.1.1. CONCENTRATIONS

For every $i, j \in \mathcal{I}$, $i \neq j$, $T \in \mathbb{N}_{\geq 1}$, let $w_{i,j}^{(T)}$, $N_{i,j}^{(T)}$, $\hat{\mu}_{i,j}^{(T)}$, $b_{i,j}^{(\delta,T)}$, $E^{(\delta,T)}$, $\hat{\prec}^{(\delta,T)}$, $d_{\text{out}}^{(\delta,T)}$, $\ell^{(T)}$ be the random elements computed from the observations $(\prec_{\text{obs}}^{(t)})_{t \in [\![T]\!]}$ according to the PIRATE algorithm (Algorithm 1). First of all, observe that, due to Assumption 3.2 and the fact that any positional censoring mechanism $\prec_{\text{pos}} \in \mathcal{M}$ satisfies $\prec_{\text{pos}} \neq \{\}$, then $p_{i,j} > 0$, so that $\mu_{i,j}$ is well-defined. We can start the analysis with concentration results for $\hat{\mu}_{i,j}^{(T)}$ and $N_{i,j}^{(T)}$.

**Lemma E.1** (Concentration of $\hat{\mu}_{i,j}^{(T)}$). *For every confidence parameter $\delta \in (0,1]$, number of observations $T \in \mathbb{N}_{\geq 1}$, we define the good event:*

$$\mathcal{G}^{(\delta,T)} = \left\{ \left| \hat{\mu}_{i,j}^{(T)} - \mu_{i,j} \right| < b_{i,j}^{(\delta,T)} \text{ for every } i, j \in \mathcal{I}, i \neq j \right\}.$$

*Then:*

$$\mathbb{P}\left[ \overline{\mathcal{G}^{(\delta,T)}} \right] < \frac{\delta}{4T^2}.$$

*Proof.* We start by proving that:

$$w_{i,j}^{(T)} \mid N_{i,j}^{(T)} = N \sim \text{Bin}(N, \mu_{i,j}).$$

Observe that, for each observation $t \in \mathbb{N}_{\geq 1}$, one and only one of the following three event happens: $i \prec_{\text{obs}}^{(t)} j$, $j \prec_{\text{obs}}^{(t)} i$, neither $i \prec_{\text{obs}}^{(t)} j$ nor $j \prec_{\text{obs}}^{(t)} i$. Then for $w, N \in \mathbb{N}_{\geq 1}$, $w \leq N \leq T$, being the observations i.i.d.:

$$\mathbb{P}_\xi\left[ w_{i,j}^{(T)} = w, N_{i,j}^{(T)} = N \right] = \binom{T}{N}\binom{N}{w} p_{i,j}^w p_{j,i}^{N-w}(1 - f_{i,j})^{T-N}.$$

For what regards $N_{i,j}^{(T)}$ instead, we have $N_{i,j}^{(T)} \sim \text{Bin}(T, f_{i,j})$. Thus:

$$\mathbb{P}_\xi\left[ w_{i,j}^{(T)} = w \mid N_{i,j}^{(T)} = N \right] = \frac{\mathbb{P}_\xi\left[ w_{i,j}^{(T)} = w, N_{i,j}^{(T)} = N \right]}{\mathbb{P}_\xi\left[ N_{i,j}^{(T)} = N \right]} = \binom{N}{w} \mu_{i,j}^w \mu_{j,i}^{N-w}.$$

Hence:

$$\begin{aligned}
&\mathbb{P}_\xi\left[ |\hat{\mu}_{i,j}^{(T)} - \mu_{i,j}| \geq b_{i,j}^{(\delta,T)} \right] \\
&= \sum_{N \in [\![0,T]\!]} \mathbb{P}_\xi\left[ \left| \hat{\mu}_{i,j}^{(T)} - \mu_{i,j} \right| \geq b_{i,j}^{(\delta,T)} \mid N_{i,j}^{(T)} = N \right] \mathbb{P}_\xi\left[ N_{i,j}^{(T)} = N \right] \\
&= \sum_{N \in [\![T]\!]} \mathbb{P}_\xi\left[ \left| \hat{\mu}_{i,j}^{(T)} - \mu_{i,j} \right| \geq b_{i,j}^{(\delta,T)} \mid N_{i,j}^{(T)} = N \right] \mathbb{P}_\xi\left[ N_{i,j}^{(T)} = N \right] \qquad (2) \\
&\leq \sum_{N \in [\![T]\!]} \left( \mathbb{P}_\xi\left[ \hat{\mu}_{i,j}^{(T)} - \mu_{i,j} \geq b_{i,j}^{(\delta,T)} \mid N_{i,j}^{(T)} = N \right] \mathbb{P}_\xi\left[ N_{i,j}^{(T)} = N \right] \right)
\end{aligned}$$

$$+ \mathbb{P}_\xi \left[ \hat{\mu}_{i,j}^{(T)} - \mu_{i,j} \leq -b_{i,j}^{(\delta,T)} \mid N_{i,j}^{(T)} = N \right] \mathbb{P}_\xi \left[ N_{i,j}^{(T)} = N \right] \Big)$$

$$= \sum_{N \in [\![T]\!]} \left( \mathbb{P}_\xi \left[ w_{i,j}^{(T)} - N\mu_{i,j} \geq N b_{i,j}^{(\delta,T)} \mid N_{i,j}^{(T)} = N \right] \mathbb{P}_\xi \left[ N_{i,j}^{(T)} = N \right] \right.$$

$$\left. + \mathbb{P}_\xi \left[ w_{i,j}^{(T)} - N\mu_{i,j} \leq -N b_{i,j}^{(\delta,T)} \mid N_{i,j}^{(T)} = N \right] \mathbb{P}_\xi \left[ N_{i,j}^{(T)} = N \right] \right)$$

$$\leq 2 \sum_{N \in [\![T]\!]} \exp \left( -\frac{2}{N} N^2 \frac{\ln(3Tn/\delta)}{N} \right) \mathbb{P}_\xi \left[ N_{i,j}^{(T)} = N \right] \tag{3}$$

$$= \frac{2\delta^2}{9T^2 n^2} \sum_{N \in [\![T]\!]} \mathbb{P}_\xi \left[ N_{i,j}^{(T)} = N \right] < \frac{\delta^2}{4T^2 n^2} \leq \frac{\delta}{4T^2 n^2},$$

where Line (2) follows from $b_{i,j}^{(\delta,T)} = +\infty$ when $N_{i,j}^{(T)} = 0$, and Line (3) follows from (Lattimore & Szepesvári, 2020, Theorem 5.3), being the Binomial distribution $\sigma$-subgaussian with $\sigma^2 = N/4$. Then, thanks to a union bound:

$$\mathbb{P}_\xi \left[ \overline{\mathcal{G}^{(\delta,T)}} \right] = \mathbb{P}_\xi \left[ \left| \hat{\mu}_{i,j}^{(T)} - \mu_{i,j} \right| \geq b_{i,j}^{(\delta,T)} \text{ for some } i,j \in \mathcal{I}, i \neq j \right]$$

$$\leq \sum_{\substack{i,j \in \mathcal{I} \\ i \neq j}} \mathbb{P}_\xi \left[ \left| \hat{\mu}_{i,j}^{(T)} - \mu_{i,j} \right| \geq b_{i,j}^{(\delta,T)} \right] < \frac{\delta}{4T^2 n^2} \left| \mathcal{I}^2 \right| = \frac{\delta}{4T^2}.$$

$\square$

**Lemma E.2** (Concentration of $N_{i,j}^{(T)}$). *For every pair of items $i,j \in \mathcal{I}$, $i \neq j$ and number of observations $T \in \mathbb{N}_{\geq 1}$:*

$$\mathbb{P}_\xi \left[ N_{i,j}^{(T)} \leq \frac{1}{2} T f_{i,j} \right] \leq \exp \left( -\frac{1}{8} T f_{i,j} \right).$$

*Proof.* Let $X_{i,j}^{(t)} = \mathbb{I}[i \prec_{\text{obs}}^{(t)} j] + \mathbb{I}[j \prec_{\text{obs}}^{(t)} i] = \mathbb{I}[i \prec_{\text{obs}}^{(t)} j \text{ or } j \prec_{\text{obs}}^{(t)} i]$. Then $(X_{i,j}^{(t)})_{t \in \mathbb{N}_{\geq 1}}$ form a collection of i.i.d. random variables, with $X_{i,j}^{(t)} \sim \text{Be}(f_{i,j})$ for every $t \in \mathbb{N}_{\geq 1}$. Observe that $N_{i,j}^{(T)} = \sum_{t \in [\![T]\!]} X_{i,j}^{(t)}$. Thus:

$$\mathbb{P}_\xi \left[ N_{i,j}^{(T)} \leq \frac{1}{2} T f_{i,j} \right] = \mathbb{P}_\xi \left[ \frac{N_{i,j}^{(T)}}{T} \leq f_{i,j} - \frac{1}{2} f_{i,j} \right]$$

$$\leq \exp \left( -T D_{\text{KL}} \left( \text{Be} \left( 1/2 f_{i,j} \right) \| \text{Be} \left( f_{i,j} \right) \right) \right) \tag{4}$$

$$\leq \exp \left( -T \frac{(1/2 f_{i,j})^2}{2 f_{i,j}} \right) = \exp \left( -\frac{1}{8} T f_{i,j} \right), \tag{5}$$

where Line (4) follows from (Lattimore & Szepesvári, 2020, Lemma 10.3) and Line (5) follows from Lemma F.3. $\square$

### E.1.2. PROOF OF THEOREM 4.1

The goal of this section is to prove Theorem 4.1, which shows that PIRATE is $\delta$-correct (Definition 3.3). We prove the almost sure termination and the correctness of the PIRATE algorithm in two distinct lemmas.

**Lemma E.3** (Almost Sure Termination of PIRATE). *For every confidence parameter $\delta \in (0,1]$ and every problem instance $\xi \in \mathcal{E}$, PIRATE terminates in finite time with probability 1, i.e.:*

$$\mathbb{P}_\xi \left[ T_{\text{PIRATE}}^{(\delta)} < +\infty \right] = 1.$$

*Proof.* Because of Assumptions 3.1 and 3.2, if $\ell^*(i) < \ell^*(j)$ for $i,j \in \mathcal{I}$, then:

$$p_{i,j} = \mathbb{E}_{\prec_{\text{pos}} \sim \psi} \left[ \sum_{r \prec_{\text{pos}} s} \mathbb{P}_{\pi \sim \rho} \left[ \pi(\ell^*(i)) = r, \pi(\ell^*(j)) = s \right] \right]$$

$$> \mathbb{E}_{\prec_{\text{pos}} \sim \psi} \left[ \sum_{r \prec_{\text{pos}} s} \mathbb{P}_{\pi \sim \rho} \left[ \pi(\ell^*(j)) = r, \pi(\ell^*(i)) = s \right] \right]$$

$$= p_{j,i}.$$

This implies $\mu_{i,j} > \mu_{j,i}$. By what we remarked in the proof of Lemma E.1 and, thanks to the strong law of large numbers:

$$\hat{\mu}_{i,j}^{(T)} \to \mu_{i,j} \text{ a.s. and}$$
$$b_{i,j}^{(\delta,T)} \to 0 \text{ a.s.,}$$

for every $i, j \in \mathcal{I}$, $i \neq j$, when $T \to +\infty$. This fact, combined with $\mu_{i,j} > \mu_{j,i}$ if and only if $\ell^*(i) < \ell^*(j)$ implies that termination is triggered at finite time with probability 1. Indeed, at some point the graph $(\mathcal{I}, E^{(\delta,T)})$ will become an acyclic tournament graph, which implies that its transitive closure, which is equal to the graph itself, will satisfy the termination condition. □

**Lemma E.4** (Correctness of PIRATE). *For every confidence parameter $\delta \in (0, 1]$ and every problem instance $\xi \in \mathcal{E}$, upon termination, the ordered collection returned by* PIRATE *coincides with the true ordered top-$k$ items with probability greater than $1 - \delta$:*

$$\mathbb{P}_{\xi} \left[ \text{PIRATE} \left( k, \delta, (\prec_{obs}^{(t)})_{t \in [\![T_{\text{PIRATE}}^{(\delta)}]\!]} \right) = (i_l^*)_{l \in [\![k]\!]} \right] > 1 - \delta.$$

*Proof.* Let $\mathcal{G}^{(\delta)} := \cap_{T \in \mathbb{N}_{\geq 1}} \mathcal{G}^{(\delta,T)}$. Under $\mathcal{G}^{(\delta)}$, we have that, given $i, j \in \mathcal{I}$, $i \neq j$, if $\hat{\mu}_{i,j}^{(T)} - b_{i,j}^{(\delta,T)} > \hat{\mu}_{j,i}^{(T)} + b_{i,j}^{(\delta,T)}$ for some $T \in \mathbb{N}_{\geq 1}$, then:

$$\mu_{i,j} > \hat{\mu}_{i,j}^{(T)} - b_{i,j}^{(T)} > \hat{\mu}_{j,i}^{(T)} + b_{i,j}^{(T)} > \mu_{j,i}.$$

By what we have remarked in the proof of Lemma E.3, $\mu_{i,j} > \mu_{j,i}$ if and only if $\ell^*(i) < \ell^*(j)$. By taking the transitive closure of $E^{(\delta,T)}$, we have that $i \hat{\prec}^{(\delta)} j$ if and only if there exist distinct $i_1, \ldots, i_h \in \mathcal{I}$ ($h \in [\![0, n-2]\!]$) such that $(i, i_1), (i_1, i_2), \ldots, (i_{h-1}, i_h), (i_h, j) \in E^{(\delta,T)}$, which in turn implies $\ell^*(i) < \ell^*(i_1) < \cdots < \ell^*(i_h) < \ell^*(j)$. Observe that if the out degree of an item $\hat{i}_1^* \in \mathcal{I}$ w.r.t. $\hat{\prec}^{(\delta)}$ is $n - 1$, it means that $\ell^*(\hat{i}_1^*) < \ell^*(j)$ for every $j \in \mathcal{I} \setminus \{\hat{i}_1^*\}$, making it the first item $i_1^*$. Analogously, if the out-degree of $\hat{i}_2^*$ is $n - 2$, then $\hat{i}_2^*$ is either the first or the second item ($\hat{i}_2^* \in \{i_1^*, i_2^*\}$), but if we already know $i_1^*$ and we have $\hat{i}_2^* \neq i_1^*$, then it must be $\hat{i}_2^* = i_2^*$. We can iterate this reasoning for the first $k$ items, proving that the termination condition of PIRATE guarantees that, under the good event $\mathcal{G}^{(\delta)}$, the returned ordered collection is exactly $(i_1^*, \ldots, i_k^*)$. Finally:

$$\mathbb{P}_{\xi} \left[ \overline{\mathcal{G}^{(\delta)}} \right] = \mathbb{P}_{\xi} \left[ \cup_{T \in \mathbb{N}_{\geq 1}} \overline{\mathcal{G}^{(\delta,T)}} \right] \leq \sum_{T \in \mathbb{N}_{\geq 1}} \mathbb{P}_{\xi} \left[ \overline{\mathcal{G}^{(\delta,T)}} \right]$$

$$< \delta \sum_{T \in \mathbb{N}_{\geq 1}} \frac{1}{4T^2} < \frac{\delta}{2}, \tag{6}$$

where at Line (6) we use Lemma E.1 and the fact that $\sum_{T \in \mathbb{N}_{\geq 1}} \frac{1}{T^2} = \frac{\pi^2}{6} < 2$. This implies that $\mathbb{P}_{\xi} \left[ \mathcal{G}^{(\delta)} \right] > 1 - \delta$. □

**Theorem 4.1** ($\delta$-Correctness of PIRATE). *For every confidence parameter $\delta \in (0, 1]$ and problem instance $\xi \in \mathcal{E}$:*

- PIRATE *terminates in finite time with probability 1.*
- *Upon termination, the ordered collection returned by* PIRATE *coincides with the true ordered top-$k$ items with probability greater than $1 - \delta$.*

*Proof.* It is a direct consequence of Lemmas E.3 and E.4. □

### E.1.3. PROOF OF THEOREM 4.2

The goal of this section is to prove Theorem 4.2, which provides an instance dependent upper bound to the sample complexity of the PIRATE algorithm.

**Theorem 4.2** (Sample Complexity Upper Bound). *For every confidence parameter $\delta \in (0, 1]$ and problem instance $\xi \in \mathcal{E}$:*

$$S_\xi^{(\delta)}(\text{PIRATE}) \leq \frac{128}{c_k^*} \ln\left(\frac{96n}{c_k^*\delta}\right).$$

*Proof.* Let $T \in \mathbb{N}_{\geq 1}$. Let us assume to be under the good event $\mathcal{G}^{(\delta,T)}$. Given $l, m \in [\![n]\!]$, $l < m$, we define:

$$S_{l,m} := \frac{64}{c_{l,m}} \ln\left(\frac{96n}{c_{l,m}} \cdot \frac{1}{\delta}\right),$$

where $c_{l,m} := f_{i_l^*, i_m^*}(\mu_{i_l^*, i_m^*} - \mu_{i_m^*, i_l^*})^2$ is the *identification rate* for the pair $(i_l^*, i_m^*)$, defined in Section 4. We now show that $S_{l,m}$ corresponds to the number of samples sufficient to have $(i_l^*, i_m^*) \in E^{(\delta,T)}$, under the condition $N_{i_l^*, i_m^*}^{(T)} > \frac{1}{2}f_{i_l^*, i_m^*}T$. Indeed, because of Lemma F.8:

$$T \geq S_{l,m} \text{ implies } T \geq 32\frac{\ln(3Tn/\delta)}{c_{l,m}}.$$

The last inequality, together with $N_{i_l^*, i_m^*}^{(T)} > \frac{1}{2}f_{i_l^*, i_m^*}T$, implies:

$$N_{i_l^*, i_m^*}^{(T)} > 16\frac{\ln(3Tn/\delta)}{(\mu_{i_l^*, i_m^*} - \mu_{i_m^*, i_l^*})^2},$$

which can be rearranged into:

$$\mu_{i_l^*, i_m^*} - \mu_{i_m^*, i_l^*} > 4\sqrt{\frac{\ln(3Tn/\delta)}{N_{i_l^*, i_m^*}^{(T)}}} = 4b_{i_l^*, i_m^*}^{(T)}.$$

Since we are under the good event, exploiting $\mu_{i,j} - b_{i,j}^{(\delta,T)} < \hat{\mu}_{i,j}^{(T)} < \mu_{i,j} + b_{i,j}^{(\delta,T)}$, the inequality above implies:

$$\hat{\mu}_{i_l^*, i_m^*}^{(T)} - b_{i_l^*, i_m^*}^{(T)} > \hat{\mu}_{i_m^*, i_l^*}^{(T)} + b_{i_l^*, i_m^*}^{(T)},$$

which is equivalent to $(i_l^*, i_m^*) \in E^{(\delta,T)}$. Lemmas F.6 and F.7, together with the $\delta$-correctness of the algorithm (Lemma E.4), collectively prove that PIRATE terminates if and only if $E^{(\delta,T)}$ contains the edges $(i_1^*, i_2^*), \ldots, (i_{k-1}^*, i_k^*)$ and there is a path in $(\mathcal{I}, E^{(\delta,T)})$ which connects $i_k^*$ and $i_l^*$ for every $l \in [\![k+1, n]\!]$. We can express this condition as:

$$\{(i_l^*, i_m^*) \text{ s.t. } (l, m) \in E_{\text{top}} \cup E_k\} \subseteq E^{(\delta,T)}, \tag{7}$$

where $E_{\text{top}} := \cup_{l \in [\![k-1]\!]}(l, l+1)$ and $E_k$ is a set of edges between the nodes $l, m \in [\![k, n]\!]$, $l < m$ s.t. there is a path from $k$ to every other node $l > k$ in the graph $([\![k, n]\!], E_k)$. Because of what we showed at the beginning of the proof, Equation (7) (and thus the termination of the algorithm) is implied by:

$$T \geq \max_{(l,m) \in E_{\text{top}} \cup E_k} S_{l,m},$$

and:

$$N_{i_l^*, i_m^*} \geq \frac{1}{2}Tf_{i_l^*, i_m^*} \text{ for every } (l, m) \in E_{\text{top}} \cup E_k,$$

for any valid choice of $E_k$. Let $E_k = E_k^*$ be the set of edges that minimize the bottleneck cost $\max_{(l,m) \in E_k} S_{l,m}$. By Lemma F.5, we have that $\max_{(l,m) \in E_k^*} S_{l,m} = \max_{m \in [\![k+1, n]\!]} \min_{l \in [\![k, m-1]\!]} S_{l,m}$. Then $S_k^* := \max_{(l,m) \in E_{\text{top}} \cup E_k^*} S_{l,m} = \max\{\max_{l \in [\![k-1]\!]} S_{l,l+1}, \max_{m \in [\![k+1, n]\!]} \min_{l \in [\![k, m-1]\!]} S_{l,m}\}$. Let $c_k^* = \min\{\min_{l \in [\![k-1]\!]} c_{l,l+1}, \min_{m \in [\![k+1, n]\!]} \max_{l \in [\![k, m-1]\!]} c_{l,m}\}$ be the bottleneck identification rate defined in Section 4. Observe that $S_k^* = 64\ln(96n/(c_k^*\delta))/c_k^*$, being $S_{l,m}$ strictly decreasing in $c_{l,m}$. Now, because of the layer cake representation:

$$\mathbb{E}_\xi\left[T_{\text{PIRATE}}^{(\delta)}\right] = \sum_{T \in \mathbb{N}_{\geq 0}} \mathbb{P}_\xi\left[T_{\text{PIRATE}}^{(\delta)} > T\right] = \sum_{T \in [\![0, \lceil S_k^* \rceil - 1]\!]} \mathbb{P}_\xi\left[T_{\text{PIRATE}}^{(\delta)} > T\right] + \sum_{T \in \mathbb{N}_{\geq \lceil S_k^* \rceil}} \mathbb{P}_\xi\left[T_{\text{PIRATE}}^{(\delta)} > T\right]$$

$$\leq S_k^* + \sum_{T \in \mathbb{N}_{\geq \lceil S_k^* \rceil}} \mathbb{P}_\xi\left[\overline{\mathcal{G}^{(\delta,T)}} \text{ or } N_{i_l^*, i_m^*}^{(T)} \leq 1/2f_{i_l^*, i_m^*}T \text{ for some } (l, m) \in E_{\text{top}} \cup E_k^*\right]$$

$$\leq S_k^* + \sum_{T \in \mathbb{N}_{\geq \lceil S_k^* \rceil}} \frac{\delta}{4T^2} + \sum_{T \in \mathbb{N}_{\geq \lceil S_k^* \rceil}} |E_{\text{top}} \cup E_k^*| \max_{(l,m) \in E_{\text{top}} \cup E_k^*} \exp\left(-\frac{1}{8} T f_{i_l^*, i_m^*}\right) \tag{8}$$

$$\leq S_k^* + \delta + n^2 \sum_{T \in \mathbb{N}_{\geq \lceil S_k^* \rceil}} \exp\left(-\frac{1}{8} T c_k^*\right) \tag{9}$$

$$\leq S_k^* + 1 + n^2 \frac{\exp(-1/8 \lceil S_k^* \rceil c_k^*)}{1 - \exp(-c_k^*/8)}$$

$$\leq S_k^* + 1 + 16 n^2 \frac{\exp(-1/8 S_k^* c_k^*)}{c_k^*} = S_k^* + 1 + 16 \frac{(c_k^*)^7 \delta^8}{96^8 n^6} \leq 2 S_k^*, \tag{10}$$

where Line (8) follows from a union bound, Lemma E.1 and Lemma E.2, Line (9) follows from the fact that $f_{i_l^*, i_m^*} \geq f_{i_l^*, i_m^*} (\mu_{i_l^*, i_m^*} - \mu_{i_m^*, i_l^*})^2 = c_{l,m}$ and $c_k^* = \min_{(l,m) \in E_{\text{top}} \cup E_k^*} c_{l,m}$ being $S_k^* = \max_{(l,m) \in E_{\text{top}} \cup E_k^*} S_{l,m}$ and due to the relation between the two quantities, finally Line (10) follows from Lemma F.9. The final expression in the bound follows from the definition of $S_k^*$ and its relationship with $c_k^*$. $\qquad \square$

## E.2. Sample Complexity Lower Bound

In this section we prove the result stated in Section 4.3.

We start with an auxiliary result that is adapted from (Kaufmann et al., 2016, Lemma 1) to the i.i.d. observations setting.

**Lemma E.5.** *Let $(\mathcal{X}, \mathcal{F})$ be a measurable space equipped with a reference measure $\lambda$. Let $\nu, \mu \in \Delta(\mathcal{X}, \mathcal{F})$ such that $\nu \ll \mu$, $\nu, \mu \ll \lambda$. We define the sequence of random variables $(X^{(t)})_{t \in \mathbb{N}_{\geq 1}}$ on the infinite product space $(\mathcal{X}^{\mathbb{N}}, \mathcal{F}^{\otimes \mathbb{N}})$ with $X^{(t)} : \mathcal{X}^{\mathbb{N}} \to \mathcal{X}$ which project $\omega = (\omega_1, \omega_2, \dots) \in \mathcal{X}^{\mathbb{N}}$ onto its coordinate $X^{(t)}(w) = \omega_t \in \mathcal{X}$ for every $t \in \mathbb{N}_{\geq 1}$. Let $\mathbb{P}_\nu = \nu^{\otimes \mathbb{N}}$, $\mathbb{P}_\mu = \mu^{\otimes \mathbb{N}}$ and $\mathbb{E}_\nu, \mathbb{E}_\mu$ the respective expectations. Under $\mathbb{P}_\nu$ (respectively $\mathbb{P}_\mu$) the sequence $(X^{(t)})_{t \in \mathbb{N}_{\geq 1}}$ is i.i.d. with marginal distribution $\nu$ (respectively $\mu$). Let $\mathbb{F} = (\mathcal{F}^{(t)})_{t \in \mathbb{N}_{\geq 1}}$ be the natural filtration where $\mathcal{F}^{(t)} = \sigma(X^{(1)}, \dots, X^{(t)})$. Let $T_{stop}$ be a stopping time w.r.t. the filtration $\mathbb{F}$ such that $\mathbb{P}_\nu[T_{stop} < +\infty] = 1$. Then, for any event $\mathcal{A} \in \mathcal{F}^{(T_{stop})}$ (where $\mathcal{F}^{(T_{stop})}$ is the $\sigma$-algebra of $T_{stop}$-past), the following inequality holds:*

$$\mathbb{E}_\nu[T_{stop}] D_{\text{KL}}(\nu \| \mu) \geq D_{\text{KL}} \left( \text{Be}\left(\mathbb{P}_\nu(\mathcal{A})\right) \| \text{Be}\left(\mathbb{P}_\mu(\mathcal{A})\right) \right),$$

*where $D_{\text{KL}}$ is the Kullback-Leibler divergence of the two distributions, formally defined in Section B.1.*

*Proof.* Let $f = \frac{d\nu}{d\lambda}$ and $g = \frac{d\mu}{d\lambda}$ be respectively the Radon-Nikodym derivative of $\nu$ and $\mu$ w.r.t. $\lambda$. Let us define the *cumulative log-likelihood process* $S^{(T)}$ for $T \in \mathbb{N}_{\geq 1}$ as:

$$S^{(T)} = \sum_{t \in \llbracket T \rrbracket} \ln \frac{f(X^{(t)})}{g(X^{(t)})}.$$

Let $\mathcal{B} \in \mathcal{F}^{(T)}$ for some $T \in \mathbb{N}_{\geq 1}$. Then, because of the definition of the $\sigma$-algebra $\sigma(X)$ induced by the random variable $X$, $\mathcal{B} = \mathcal{C} \times \mathcal{X}^{\mathbb{N}_{\geq T+1}}$ with $\mathcal{C} \in \otimes_{t \in \llbracket T \rrbracket} \mathcal{F}^{(t)}$. Then:

$$\mathbb{P}_\mu [\mathcal{B}] = \mu^{\otimes \mathbb{N}}(\mathcal{B}) = \left(\otimes_{t \in \llbracket T \rrbracket} \mu\right)(\mathcal{C})$$

$$= \int \mathbb{I}_{\mathcal{C}}(\omega_1, \dots, \omega_T) \prod_{t \in \llbracket T \rrbracket} g(\omega_t) d\left(\otimes_{t \in \llbracket T \rrbracket} \lambda\right)(\omega_1, \dots, \omega_T) \tag{11}$$

$$\geq \int \mathbb{I}_{\mathcal{C}}(\omega_1, \dots, \omega_T) \mathbb{I}\left[f(\omega_t) > 0 \text{ for every } t \in \llbracket T \rrbracket\right] \prod_{t \in \llbracket T \rrbracket} g(\omega_t) d\left(\otimes_{t \in \llbracket T \rrbracket} \lambda\right)(\omega_1, \dots, \omega_T) \tag{12}$$

$$= \int \mathbb{I}_{\mathcal{C}}(\omega_1, \dots, \omega_T) \mathbb{I}\left[f(\omega_t) > 0 \text{ for every } t \in \llbracket T \rrbracket\right] \prod_{t \in \llbracket T \rrbracket} \frac{g}{f}(\omega_t) \prod_{t \in \llbracket T \rrbracket} f(\omega_t) d\left(\otimes_{t \in \llbracket T \rrbracket} \lambda\right)(\omega_1, \dots, \omega_T) \tag{13}$$

$$= \int \mathbb{I}_{\mathcal{C}}(\omega_1, \dots, \omega_T) \prod_{t \in \llbracket T \rrbracket} \frac{g}{f}(\omega_t) d\left(\otimes_{t \in \llbracket T \rrbracket} \nu\right)(\omega_1, \dots, \omega_T) \tag{14}$$

$$= \int \mathbb{I}_{\mathcal{B}} \prod_{t \in [\![T]\!]} \frac{g}{f}\left(X^{(t)}\right) d\mathbb{P}_\nu = \mathbb{E}_\nu \left[ \mathbb{I}_{\mathcal{B}} \exp\left(-S^{(T)}\right) \right],$$

where Line (11) expresses the probability using the density $g$ of the product measure w.r.t. the reference measure $\lambda$, Line (12) establishes an inequality by restricting the integration domain to the set where $f > 0$ (discarding any mass $\mu$ places where $f = 0$), Line (13) is obtained by multiplying and dividing by $f(\omega_t)$, and Line (14) changes the measure back to $\nu$ (since $\nu$ assigns probability 1 to the set where $f > 0$).

Since $\mathbb{P}_\nu \left[ T_{\text{stop}} < +\infty \right] = 1$, we have:

$$\mathbb{P}_\mu[\mathcal{A}] = \sum_{T \in \mathbb{N}_{\geq 1}} \mathbb{P}_\mu \left[ \mathcal{A} \cap \{ T_{\text{stop}} = T \} \right]$$

$$\geq \sum_{T \in \mathbb{N}_{\geq 1}} \mathbb{E}_\nu \left[ \mathbb{I}_{\mathcal{A} \cap \{ T_{\text{stop}} = T \}} \exp\left(-S^{(T)}\right) \right] \tag{15}$$

$$= \sum_{T \in \mathbb{N}_{\geq 1}} \mathbb{E}_\nu \left[ \mathbb{I}_{\mathcal{A}} \mathbb{I}[T_{\text{stop}} = T] \exp\left(-S^{(T_{\text{stop}})}\right) \right]$$

$$= \mathbb{E}_\nu \left[ \mathbb{I}_{\mathcal{A}} \exp\left(-S^{(T_{\text{stop}})}\right) \right], \tag{16}$$

where Line (15) follows from the fact that $\mathcal{A} \in \mathcal{F}^{(T_{\text{stop}})}$, which implies $\mathcal{A} \cap \{ T_{\text{stop}} = T \} \in \mathcal{F}^{(T)}$ and allows us to use the inequality we derived above, and Line (16) is a direct consequence of the monotone convergence theorem.

Let $p = \mathbb{P}_\nu [\mathcal{A}]$ and $q = \mathbb{P}_\mu [\mathcal{A}]$. We analyze the term $p \ln \frac{p}{q}$ which is one of the summands of the Kullback-Leibler divergence of two Bernoulli distributions, assuming $q > 0$. Observe that:

$$\ln \frac{q}{p} \geq \ln \frac{\mathbb{E}_\nu \left[ \mathbb{I}_{\mathcal{A}} \exp\left(-S^{(T_{\text{stop}})}\right) \right]}{\mathbb{P}_\nu [\mathcal{A}]} \tag{17}$$

$$= \ln \mathbb{E}_\nu \left[ \exp\left(-S^{(T_{\text{stop}})}\right) \mid \mathcal{A} \right]$$

$$\geq \mathbb{E}_\nu \left[ -S^{(T_{\text{stop}})} \mid \mathcal{A} \right], \tag{18}$$

where Line (17) follows from the inequality derived in (16) combined with the monotonicity of the logarithm, and Line (18) follows from Jensen's inequality, which we can apply because $\mathbb{E}_\nu \left[ \mathbb{I}_{\mathcal{A}} \exp\left(-S^{(T_{\text{stop}})}\right) \right] \leq q \leq 1$ and is thus integrable. Then, if $p > 0$:

$$p \ln \frac{p}{q} = p\left(-\ln \frac{q}{p}\right) \leq p\mathbb{E}_\nu \left[ S^{(T_{\text{stop}})} \mid \mathcal{A} \right]$$

$$= \mathbb{E}_\nu \left[ \mathbb{I}_{\mathcal{A}} S^{(T_{\text{stop}})} \right].$$

When $p = 0$, the inequality above holds trivially since the last term consists in the integration of a function which is 0 outside of a set of measure 0. Applying the same logic to $\overline{\mathcal{A}}$ (obtaining the probabilities $1 - p$ and $1 - q$), we have:

$$(1 - p) \ln \frac{1 - p}{1 - q} \leq \mathbb{E}_\nu \left[ \mathbb{I}_{\overline{\mathcal{A}}} S^{(T_{\text{stop}})} \right].$$

Summing the inequalities, we get:

$$D_{\text{KL}} \left( \text{Be}(p) \| \text{Be}(q) \right) \leq \mathbb{E}_\nu \left[ S^{(T_{\text{stop}})} \right].$$

Since $\nu \ll \mu$, $\mathbb{P}_\mu [\mathcal{A}] = 0$ implies $\mathbb{P}_\nu [\mathcal{A}] = 0$, making everything to hold even when $q = 0$ under the usual convention: $0 \ln(0/0) = 0$.

Under $\mathbb{P}_\nu$, the *log-likelihood increments* $\ln(f(X^{(t)})/g(X^{(t)}))$ are i.i.d., being functions of i.i.d. random variables $\left(X^{(t)}\right)_{t \in \mathbb{N}_{\geq 1}}$ with distribution $\nu$. The expected value of a single increment is:

$$\mathbb{E}_\nu \left[ \ln \frac{f\left(X^{(t)}\right)}{g\left(X^{(t)}\right)} \right] = \int_{\mathcal{X}} \ln \frac{f(x)}{g(x)} f(x) d\lambda(x) = D_{\text{KL}}(\nu \| \mu).$$

Since $T_{\text{stop}}$ is a stopping time w.r.t. $\mathbb{F}$ and has finite expectation (otherwise the desired result is trivial), Wald's identity applies:

$$D_{\text{KL}}(\text{Be}(p)\|\text{Be}(q)) \leq \mathbb{E}_\nu\left[S^{(T_{\text{stop}})}\right] = \mathbb{E}_\nu\left[T_{\text{stop}}\right]D_{\text{KL}}(\nu\|\mu),$$

concluding the proof. $\qquad\square$

In order to derive a lower bound to the sample complexity suffered by any $\delta$-correct learner, we need to construct a collection of problem instances that produce similar observations, making them hard to distinguish. To this end, we need to choose a ranking model and a distribution over positional censorship mechanisms. Analogously to what is done by Saha & Gopalan (2019a), we choose PL (Section 2.2) as the ranking model and the winner feedback $\prec^{\text{win}}_{\text{pos}}$ as the positional censorship mechanism: the distribution over positional censorship mechanisms that characterizes our instance is a Dirac delta measure over the value $\prec^{\text{win}}_{\text{pos}}$, that we denote by $\delta(\prec^{\text{win}}_{\text{pos}})$. Given a parameter collection $\boldsymbol{\theta} = (\theta_l)_{l\in[\![n]\!]}$ with $1 = \theta_1 > \cdots > \theta_n \geq 1/2$, let $\xi^{\text{win}}_{\boldsymbol{\theta}\,k} = ([\![n]\!], \text{PL}_{\boldsymbol{\theta}}, \delta(\prec^{\text{win}}_{\text{pos}}), e, k)$ (where the labeling function is the identity permutation $e$) be a problem instance with the chosen ranking model $\text{PL}_{\boldsymbol{\theta}}$, which is defined in Section B.1, and distribution over positional censorship mechanisms. Under this choice of ranking model and positional censorship mechanism, the identification rate $c_{\boldsymbol{\theta}\,l,m}$ of the pair $(i^*_l, i^*_m)$ in the instance $\xi^{\text{win}}_{\boldsymbol{\theta}\,k}$ admits a closed form. Indeed:

$$\mathbb{P}_{\xi^{\text{win}}_{\boldsymbol{\theta}\,k}}\left[i^*_l \prec^{(1)}_{\text{obs}} i^*_m\right] = \text{PL}_{\boldsymbol{\theta}}\left[\pi(l) = 1\right] = \frac{\theta_l}{\Theta},$$

where the last equality follows from a known property of the PL model (Yellott, 1977), and thus:

$$c_{\boldsymbol{\theta}\,l,m} = \frac{1}{\Theta}\frac{(\theta_l - \theta_m)^2}{\theta_l + \theta_m}.$$

Furthermore, under the previous choices, the observations $\prec^{(t)}_{\text{obs}}$ have a one-to-one correspondence with the set of items $[\![n]\!]$. In particular $\prec^{(t)}_{\text{obs}}$ is determined by the winner item in the ranking sampled in round $t$, which we denote by $i^{(t)} \in [\![n]\!]$. Thus, to describe the stochastic process of interest, we can simply consider the following probability measure:

$$\mathbb{P}_{\boldsymbol{\theta}}[i] = \text{PL}_{\boldsymbol{\theta}}[\pi(i) = 1] = \frac{\theta_i}{\Theta} \text{ for every } i \in [\![n]\!].$$

Now, for every $u < v$, $u, v \in [\![n]\!]$, denote by $\boldsymbol{\theta}^{(u,v)} = (\theta^{(u,v)}_l)_{l\in[\![n]\!]}$ the parameter collection such that $\theta^{(u,v)}_l = \theta_l$ for every $l \in [\![n]\!] \setminus \{u,v\}$, $\theta^{(u,v)}_u = \theta_v$, $\theta^{(u,v)}_v = \theta_u$, i.e., the parameter collection obtained by swapping parameters $\theta_u$ and $\theta_v$ in $\boldsymbol{\theta}$. Observe that $\Theta^{(u,v)} = \sum_{l\in[\![n]\!]} \theta^{(u,v)}_l = \Theta$. Let $\xi^{(u,v)\,\text{win}}_{\boldsymbol{\theta}\,k} = ([\![n]\!], \text{PL}_{\boldsymbol{\theta}}, \delta(\prec^{\text{win}}_{\text{pos}}), \tau_{u,v}, k)$ be the instance where the labels of the items $u, v \in [\![n]\!]$, $u < v$, are swapped. Observe that the probability measure that describes the observations in $\xi^{(u,v)\,\text{win}}_{\boldsymbol{\theta}\,k}$ is $\mathbb{P}_{\boldsymbol{\theta}^{(u,v)}}$. Furthermore, because of the properties of the PL model and because of how we defined $\boldsymbol{\theta}$: $\xi^{\text{win}}_{\boldsymbol{\theta}\,k}, \xi^{(u,v)\,\text{win}}_{\boldsymbol{\theta}\,k} \in \mathcal{E}$ for every $u, v \in [\![n]\!]$, $u < v$, i.e., they satisfy both Assumptions 3.1 and 3.2.

We have defined all the machinery that is needed to prove Theorem 4.3.

**Theorem 4.3** (Sample Complexity Lower Bound). *For every $\delta$-correct learner* Alg, *number of items $n \in \mathbb{N}_{\geq 2}$, number of top items $k \in [\![n]\!]$, bottleneck identification rate $c^*_k \in \left(0, \frac{1}{6n^3}\right]$ and confidence parameter $\delta \in (0, 1/2)$, we have that:*

$$\sup_{\substack{\xi\in\mathcal{E}(c^*_k) \\ s.t. \, |\mathcal{I}|=n}} S^{(\delta)}_\xi(\text{Alg}) \geq \frac{1}{16c^*_k}\ln\left(\frac{1}{4\delta}\right).$$

*Proof.* The theorem follows by an application of Lemma E.5 to $T^{(\delta)}_{\text{Alg}}$. To this end, let us compute an upper bound to the Kullback-Leibler divergence of the probability measures that describe the observations of the instances $\xi^{\text{win}}_{\boldsymbol{\theta}\,k}$ and $\xi^{(u,v)\,\text{win}}_{\boldsymbol{\theta}\,k}$ with $u, v \in [\![n]\!]$, $u < v$.

$$D_{\text{KL}}(\mathbb{P}_{\boldsymbol{\theta}}\|\mathbb{P}_{\boldsymbol{\theta}^{(u,v)}}) \leq \sum_{i\in[\![n]\!]}\frac{\mathbb{P}_{\boldsymbol{\theta}}[i]^2}{\mathbb{P}_{\boldsymbol{\theta}^{(u,v)}}[i]} - 1 = \frac{1}{\Theta}\sum_{i\in[\![n]\!]}\frac{\theta_i^2}{\theta_i^{(u,v)}} - 1 \qquad (19)$$

$$= \frac{1}{\Theta} \left( \sum_{\substack{i \in [\![n]\!] \\ i \neq u,v}} \theta_i + \frac{\theta_u^2}{\theta_v} + \frac{\theta_v^2}{\theta_u} \right) - 1$$

$$= \frac{1}{\Theta} \left( \Theta - \theta_u - \theta_v + \frac{\theta_u^2}{\theta_v} + \frac{\theta_v^2}{\theta_u} \right) - 1$$

$$= \frac{1}{\Theta} \left( \frac{\theta_u^2 - \theta_v^2}{\theta_v} + \frac{\theta_v^2 - \theta_u^2}{\theta_u} \right) = \frac{(\theta_u + \theta_v)(\theta_u - \theta_v)}{\Theta} \left( \frac{1}{\theta_v} - \frac{1}{\theta_u} \right)$$

$$= \frac{\theta_u + \theta_v}{\Theta \theta_u \theta_v} (\theta_u - \theta_v)^2 = c_{\boldsymbol{\theta}\ u,v} \frac{(\theta_u + \theta_v)^2}{\theta_u \theta_v} \leq 16 c_{\boldsymbol{\theta}\ u,v}, \tag{20}$$

where Line (19) follows from (Popescu et al., 2016, Theorem 1.4), and Line (20) follows from $1 \geq \theta_l \geq 1/2$ for every $l \in [\![n]\!]$. Consider the event $\mathcal{A}_k = \left\{ \mathrm{Alg}\left( k, \delta, (\prec_{\mathrm{obs}}^{(t)})_{t=1}^{T_{\mathrm{Alg}}^{(\delta)}} \right) = [\![k]\!] \right\}$. Because of the $\delta$-correctness of Alg, if the observations are generated by $\xi_{\boldsymbol{\theta}\ k}^{\mathrm{win}}$, then $\mathbb{P}_{\xi_{\boldsymbol{\theta}\ k}^{\mathrm{win}}}[\mathcal{A}_k] > 1 - \delta$. Conversely, if they are generated by $\xi_{\boldsymbol{\theta}\ k}^{(u,v)\ \mathrm{win}}$ with $u \in [\![k]\!]$, $v \in [\![u+1, n]\!]$, $\mathbb{P}_{\xi_{\boldsymbol{\theta}\ k}^{(u,v)\ \mathrm{win}}}[\mathcal{A}_k] < \delta$, since, in such a case, the correct top-$k$ items collection would have $v$ at the place of $u$. In virtue of Lemma F.4, we have that:

$$D_{\mathrm{KL}} \left( \mathrm{Be}\left( \mathbb{P}_{\xi_{\boldsymbol{\theta}\ k}^{\mathrm{win}}}[\mathcal{A}_k] \right) \| \mathrm{Be}\left( \mathbb{P}_{\xi_{\boldsymbol{\theta}\ k}^{(u,v)\ \mathrm{win}}}[\mathcal{A}_k] \right) \right) \geq \ln\left( \frac{1}{4\delta} \right). \tag{21}$$

Thus, Lemma E.5 together with Equations (20) and (21), implies:

$$S_{\xi_{\boldsymbol{\theta}\ k}^{\mathrm{win}}}^{(\delta)}(\mathrm{Alg}) \geq \frac{1}{16 c_{\boldsymbol{\theta}\ u,v}} \ln\left( \frac{1}{4\delta} \right) \quad \text{for every } u \in [\![k]\!], v \in [\![u+1, n]\!].$$

This implies:

$$S_{\xi_{\boldsymbol{\theta}\ k}^{\mathrm{win}}}^{(\delta)}(\mathrm{Alg}) \geq \frac{1}{16 \min\left\{ \min_{l \in [\![k-1]\!]} c_{\boldsymbol{\theta}\ l,l+1}, \min_{m \in [\![k+1,n]\!]} c_{\boldsymbol{\theta}\ k,m} \right\}} \ln\left( \frac{1}{4\delta} \right).$$

Because of the expression of $c_{\boldsymbol{\theta}\ l,m}$, the fact that $\theta_1 > \cdots > \theta_n$, being $f_a(x) = (x-a)^2/(x+a)$ increasing for $x > a > 0$, for every $m \in [\![k+1, n]\!]$, $l \in [\![k, m-1]\!]$ we have:

$$c_{\boldsymbol{\theta}\ k,m} = \frac{1}{\Theta} \frac{(\theta_k - \theta_m)^2}{\theta_k + \theta_m} = \frac{1}{\Theta} f_{\theta_m}(\theta_k) \geq \frac{1}{\Theta} f_{\theta_m}(\theta_l) = \frac{1}{\Theta} \frac{(\theta_l - \theta_m)^2}{\theta_l + \theta_m} = c_{\boldsymbol{\theta}\ l,m},$$

implying: $c_{\boldsymbol{\theta}\ k,m} = \max_{l \in [\![k,m-1]\!]} c_{\boldsymbol{\theta}\ l,m}$. Hence:

$$S_{\xi_{\boldsymbol{\theta}\ k}^{\mathrm{win}}}^{(\delta)}(\mathrm{Alg}) \geq \frac{1}{16 \min\left\{ \min_{l \in [\![k-1]\!]} c_{\boldsymbol{\theta}\ l,l+1}, \min_{m \in [\![k+1,n]\!]} \max_{l \in [\![k,m-1]\!]} c_{\boldsymbol{\theta}\ l,m} \right\}} \ln\left( \frac{1}{4\delta} \right)$$

$$= \frac{1}{16 c_{\boldsymbol{\theta}\ k}^*} \ln\left( \frac{1}{4\delta} \right),$$

where $c_{\boldsymbol{\theta}\ k}^* = \min\left\{ \min_{l \in [\![k-1]\!]} c_{\boldsymbol{\theta}\ l,l+1}, \min_{m \in [\![k+1,n]\!]} \max_{l \in [\![k,m-1]\!]} c_{\boldsymbol{\theta}\ l,m} \right\}$ is the *bottleneck identification rate* of the instance $\xi_{\boldsymbol{\theta}\ k}^{\mathrm{win}}$. To conclude the theorem we need to show that, for every value $c_k^* \in \left(0, \frac{1}{6n^3}\right]$, there exists a choice of parameters $\boldsymbol{\theta}$, satisfying $1 = \theta_1 > \cdots > \theta_n \geq 1/2$, such that $c_{\boldsymbol{\theta}\ k}^* = c_k^*$. First of all observe that the domain of $\boldsymbol{\theta}$ is convex, being a polytope. Furthermore, $c_{\boldsymbol{\theta}\ k}^*$ is a continuous function of $\boldsymbol{\theta}$ being obtained by taking the minimum or the maximum of continuous functions of $\boldsymbol{\theta}$. Clearly, as $\theta_l \to \theta_{l+1}$, $c_{\boldsymbol{\theta}\ k}^* \to 0$. Now, consider the uniform allocation $\boldsymbol{\theta}^{\mathrm{unif}} = (\theta_l^{\mathrm{unif}})_{l \in [\![n]\!]}$:

$$\theta_l^{\mathrm{unif}} = 1 - (l-1)\Delta \quad \text{for every } l \in [\![n]\!], \text{ with } \Delta = \frac{1}{2(n-1)}.$$

Observe that $\theta_1^{\mathrm{unif}} = 1$ and $\theta_n^{\mathrm{unif}} = 1/2$ so that the set of parameters is inside the polytope. Under this choice, we have:

$$\Theta^{\mathrm{unif}} = \frac{3n}{4}.$$

Then:

$$c_{\boldsymbol{\theta}^{\text{unif}}\ l,m} = \frac{4}{3n}\frac{(m-l)^2\Delta^2}{\theta_l+\theta_m} \geq \frac{2}{3n}(m-l)^2\Delta^2 \tag{22}$$
$$= \frac{1}{6n}(m-l)^2\frac{1}{(n-1)^2} \geq \frac{1}{6n^3}(m-l)^2$$
$$\geq \frac{1}{6n^3}$$

for every $l \in [\![n-1]\!], m \in [\![l+1,n]\!]$, where Line (22) follows from the fact that $\theta_l, \theta_m \leq 1$. Hence:

$$c^*_{\boldsymbol{\theta}\ k} \geq \frac{1}{6n^3}.$$

Finally, the proof follows by applying the intermediate value theorem on the function obtained by restricting $c^*_{\boldsymbol{\theta}\ k}$ on the segment which connects $\boldsymbol{\theta}^{\text{unif}}$ and a point $\boldsymbol{\theta}^\epsilon$ where $c^*_{\boldsymbol{\theta}^\epsilon\ k} < \epsilon$ for $\epsilon > 0$, showing that $c^*_{\boldsymbol{\theta}\ k}$ can attain each value in $\left(0, \frac{1}{6n^3}\right]$. Observe that such segment is in the polytope due to its convexity. $\qquad\square$

### E.3. Factorization of the Identification Rate

Here we prove the results stated in Section 4.4.

We start with the lower bound for the identification rate $c_{l,m}$.

**Lemma E.6** (Factorized Lower Bound to the Identification Rate). *For every problem instance $\xi \in \mathcal{E}$, $l, m \in [\![n]\!]$, $l \neq m$, the identification rate for the pair $(i_l^*, i_m^*)$ admits the following lower bound:*

$$c_{l,m} \geq \tilde{c}_{l,m} := \langle H_{\rho\ l,m}, P_\psi \rangle_F,$$

*where $H_{\rho\ l,m}$ and $P_\psi$ are defined in Section 4.4.*

*Proof.* Given a positional censorship mechanism $\prec_{\text{pos}}$ and a corresponding incomplete ranking $\prec_{\text{obs}}$, we denote by $A_{\text{pos}} = (\mathbb{I}[l \prec_{\text{pos}} m])_{l,m \in [\![n]\!]}$ and $A_{\text{obs}} = (\mathbb{I}[i_l^* \prec_{\text{obs}} i_m^*])_{l,m \in [\![n]\!]}$ their respective adjacency matrix, where, in the second matrix, we index items according to their true labels given by $\ell^*$. Observe that $i_l^* \prec_{\text{obs}} i_m^*$ if and only if $\pi(l) \prec_{\text{pos}} \pi(m)$ according to the sampled permutation $\pi$. This means that the two matrices are linked by the following relationship:

$$A_{\text{obs}} = \Sigma_\pi A_{\text{pos}} \Sigma_\pi^\top,$$

where $\Sigma_\pi = (\mathbb{I}[r = \pi(l)])_{l,r \in [\![n]\!]}$ is the permutation matrix associated with the permutation $\pi$. We recall the vectorization identity $\text{vec}(ABC) = (C^\top \otimes A)\text{vec}(B)$, which we can apply to obtain:

$$\text{vec}(A_{\text{obs}}) = (\Sigma_\pi \otimes \Sigma_\pi)\text{vec}(A_{\text{pos}}). \tag{23}$$

Now, let $P_\xi = (\mathbb{P}_\xi[i_l^* \prec_{\text{obs}}^{(1)} i_m^*])_{l,m \in [\![n]\!]}$. By taking the expectation $\mathbb{E}_\xi$ of Equation 23, we obtain a relationship between $P_\psi$ and $P_\xi$:

$$\text{vec}(P_\xi) = K_\rho \text{vec}(P_\psi), \tag{24}$$

where $K_\rho := \mathbb{E}_{\pi\sim\rho}[\Sigma_\pi \otimes \Sigma_\pi]$. Observe that $K_\rho$ is the matrix of *second order marginals*: $K_\rho = (k_{\rho\ (l,m),(r,s)})_{(l,m),(r,s)\in[\![n]\!]^2}$ with $k_{\rho\ (l,m),(r,s)} = \mathbb{P}_{\pi\sim\rho}[\pi(l) = r, \pi(m) = s]$. We are ready to derive an expression for the identification rate matrix $C = (c_{l,m})_{l,m\in[\![n]\!]}$, which depends on $K_\rho$ and $P_\psi$. Observe that through straightforward algebraic manipulation, we can write: $c_{l,m} = f_{i_l^*,i_m^*}(\mu_{i_l^*,i_m^*} - \mu_{i_m^*,i_l^*})^2 = (p_{i_l^*,i_m^*} - p_{i_m^*,i_l^*})^2/(p_{i_l^*,i_m^*} + p_{i_m^*,i_l^*})$. Even though $c_{l,m}$ is not defined for $l = m$, this is not a problem, since the sample complexity is determined only by the values of $C$ above the main diagonal. We can set the values in the diagonal of $C$ as we prefer. Then:

$$C = \frac{(P - P^\top)^{\odot 2}}{P + P^\top},$$

which in turn implies:

$$\text{vec}(C) = \frac{((I_{n^2} - T_{n,n})K_\rho \text{vec}(P_\psi))^{\odot 2}}{(I_{n^2} + T_{n,n})K_\rho \text{vec}(P_\psi)}.$$

To lower bound the identification rate matrix $C$, we apply Lemma F.10 element-wise. By defining the vectors $\boldsymbol{u}_{\rho\,l,m} := K_\rho^\top(\boldsymbol{e}_{l,m} - \boldsymbol{e}_{m,l})$ and $\boldsymbol{v}_{\rho\,l,m} := K_\rho^\top(\boldsymbol{e}_{l,m} + \boldsymbol{e}_{m,l})$, where $\boldsymbol{e}_{l,m}$ is the vectorization of the matrix with a single $1$ in the $(l,m)$-th entry, we can express each entry $c_{l,m}$ in the form required by the lemma:

$$c_{l,m} = \frac{(\boldsymbol{u}_{\rho\,l,m}^\top \operatorname{vec}(P_\psi))^2}{\boldsymbol{v}_{\rho\,l,m}^\top \operatorname{vec}(P_\psi)}.$$

We choose the linearization point $\boldsymbol{x}_0 = \operatorname{vec}(I_{n,<})$. This choice corresponds to the full-ranking observation model. Let $r_{\rho\,l,m}$ be the ratio evaluated at this point:

$$r_{\rho\,l,m} := \frac{\boldsymbol{u}_{\rho\,l,m}^\top \operatorname{vec}(I_{n,<})}{\boldsymbol{v}_{\rho\,l,m}^\top \operatorname{vec}(I_{n,<})}.$$

Applying Lemma F.10 with $\boldsymbol{x} = \operatorname{vec}(P_\psi)$, we obtain the lower bound $c_{l,m} \geq \boldsymbol{g}_{\rho\,l,m}^\top \operatorname{vec}(P_\psi)$, where the gradient vector is $\boldsymbol{g}_{\rho\,l,m} = 2r_{\rho\,l,m}\boldsymbol{u}_{\rho\,l,m} - r_{\rho\,l,m}^2\boldsymbol{v}_{\rho\,l,m}$. Finally, since $P_\psi$ is strictly upper-triangular, being each positional censorship mechanism consistent with the standard order $<$, we have that $\boldsymbol{g}_{\rho\,l,m}^\top \operatorname{vec}(P_\psi) = \langle H_{\rho\,l,m}, P_\psi \rangle_F$, with $H_{\rho\,l,m} := \operatorname{mat}(\boldsymbol{g}_{\rho\,l,m}) \odot I_{n,<}$. Now, because of the structure of $K_\rho$ and the definition of $M_{\rho\,l,m}$ provided in Section 4.4, we have that:

$$U_{\rho\,l,m} := \operatorname{mat}(\boldsymbol{u}_{\rho\,l,m}) = M_{\rho\,l,m} - M_{\rho\,m,l},$$
$$V_{\rho\,l,m} := \operatorname{mat}(\boldsymbol{v}_{\rho\,l,m}) = M_{\rho\,l,m} + M_{\rho\,m,l},$$
$$\gamma_{\rho\,l,m} := \boldsymbol{u}_{\rho\,l,m}^\top \operatorname{vec}(I_{n,<}) = \langle U_{\rho\,l,m}, I_{n,<}\rangle_F = \mathbb{P}_{\pi\sim\rho}\left[\pi(l) < \pi(m)\right] - \mathbb{P}_{\pi\sim\rho}\left[\pi(m) < \pi(l)\right],$$
$$\boldsymbol{v}_{\rho\,l,m}^\top \operatorname{vec}(I_{n,<}) = \langle V_{\rho\,l,m}, I_{n,<}\rangle_F = \mathbb{P}_{\pi\sim\rho}\left[\pi(l) < \pi(m)\right] + \mathbb{P}_{\pi\sim\rho}\left[\pi(m) < \pi(l)\right] = 1.$$

Thus:

$$r_{\rho\,l,m} = \gamma_{\rho\,l,m},$$
$$\begin{aligned}
H_{\rho\,l,m} &= \operatorname{mat}(\boldsymbol{g}_{\rho\,l,m}) \odot I_{n,<} \\
&= 2\gamma_{\rho\,l,m}(\operatorname{triu}(M_{\rho\,l,m}) - \operatorname{triu}(M_{\rho\,m,l})) - \gamma_{\rho\,l,m}^2(\operatorname{triu}(M_{\rho\,l,m}) + \operatorname{triu}(M_{\rho\,m,l})) \\
&= \gamma_{\rho\,l,m}(2 - \gamma_{\rho\,l,m})\operatorname{triu}(M_{\rho\,l,m}) - \gamma_{\rho\,l,m}(2 + \gamma_{\rho\,l,m})\operatorname{triu}(M_{\rho\,m,l}).
\end{aligned}$$

$\square$

The following corollary follows immediately from the fact that the sample complexity upper bound for the PIRATE algorithm reported in Theorem 4.2 is non-increasing in the identification rates $c_{l,m}$.

**Corollary E.7** (Factorized Sample Complexity Upper Bound). *For every confidence parameter $\delta \in (0,1]$ and problem instance $\xi \in \mathcal{E}$:*

$$S_\xi^{(\delta)}(\text{PIRATE}) \leq \frac{128}{\tilde{c}_k^*} \ln\left(\frac{96n}{\tilde{c}_k^*\delta}\right),$$

*with:*

$$\tilde{c}_k^* := \min\left\{\min_{l\in[\![k-1]\!]} \tilde{c}_{l,l+1}, \min_{m\in[\![k+1,n]\!]} \max_{l\in[\![k,m-1]\!]} \tilde{c}_{l,m}\right\},$$
$$\tilde{c}_{l,m} := \max\left\{\langle H_{\rho\,l,m}, P_\psi\rangle_F, 0\right\}, \text{ for } l,m \in [\![n]\!], l \neq m.$$

### E.3.1. COMPUTATIONS FOR EXAMPLE 1

Here we report the computations for the closed form expressions of the quantities involved in Example 1.

The adjacency matrix of the censorship mechanism is:

$$P_{(1,2)} = \begin{pmatrix} 0 & 1 & 0 \\ 0 & 0 & 0 \\ 0 & 0 & 0 \end{pmatrix}.$$

Let us abbreviate $C(\phi) = C(\phi, T, e)$ and compute the closed form expression for $H_{\phi\ 1,2}$ and $H_{\phi\ 2,3}$. Under this choice for the ranking model, we have:

$$\gamma_{\phi\ 1,2} = \gamma_{\phi\ 2,3} = \frac{1-\phi}{1+\phi}.$$

Furthermore:

$$M_{\phi\ 1,2} = C(\phi) \begin{pmatrix} 0 & 1 & \phi \\ \phi & 0 & \phi^2 \\ \phi^2 & \phi^3 & 0 \end{pmatrix},$$

$$M_{\phi\ 2,3} = C(\phi) \begin{pmatrix} 0 & \phi^2 & \phi \\ \phi^3 & 0 & 1 \\ \phi^2 & \phi & 0 \end{pmatrix}.$$

Finally, it holds that:

$$H_{\phi\ 1,2} = N(\phi)\frac{1}{\sqrt{1+\phi^2+\phi^4}} \begin{pmatrix} 0 & 1 & \phi \\ 0 & 0 & \phi^2 \\ 0 & 0 & 0 \end{pmatrix},$$

$$H_{\phi\ 2,3} = N(\phi)\frac{1}{\sqrt{1+\phi^2+\phi^4}} \begin{pmatrix} 0 & \phi^2 & \phi \\ 0 & 0 & 1 \\ 0 & 0 & 0 \end{pmatrix},$$

where $N(\phi) = \|H_{\phi\ 1,2}\|_F = \|H_{\phi\ 2,3}\|_F = \left(\frac{1-\phi}{1+\phi}\right)^2 \sqrt{\frac{1-\phi+\phi^2}{1+\phi+\phi^2}}$.

### E.3.2. CASTING THE FACTORIZED IDENTIFICATION RATE TO THE PLACKETT-LUCE SETTING

The following lemma provides a lower bound for $\langle H_{\rho\ l,m}, P_\psi\rangle_F$, when the ranking model is $PL_{\boldsymbol\theta}$ for a parameter collection $\boldsymbol\theta = (\theta_l)_{l\in[\![n]\!]}$ with $1 \geq \theta_1 > \cdots > \theta_n \geq 1/2$ and the distribution over the positional censorship mechanisms is a Dirac delta over the top-$h$ feedback: $\delta(\prec_{pos}^{top\text{-}h})$.

**Lemma E.8.** *Let $n \in \mathbb{N}_{\geq 20}$ be the number of items. Let $\boldsymbol\theta = (\theta_l)_{l\in[\![n]\!]}$, with $1 \geq \theta_1 > \ldots \theta_n \geq 1/2$ be the parameters of a PL model. Consider the problem instance $\xi_{\boldsymbol\theta\ k}^{top\text{-}h} = \left([\![n]\!], PL_{\boldsymbol\theta}, \delta(\prec_{pos}^{top\text{-}h}), e, k\right)$ where the item labeling is given by the identity permutation, the ranking model is PL and the distribution over the positional censorship mechanisms is a Dirac delta over the top-$h$ feedback with $h \in [\![\lfloor n/20 \rfloor]\!]$. We replace $\boldsymbol\theta$ to $\rho$ and top-$h$ to $\delta(\prec_{pos}^{top\text{-}h})$ in the subscripts to simplify the notation. Then, for every item labels $l, m \in [\![n]\!]$, $l < m$:*

$$\langle H_{\boldsymbol\theta\ l,m}, P_{top\text{-}h}\rangle_F \geq \frac{1}{4}\frac{h}{n}(\theta_l - \theta_m)^2.$$

*Proof.* Towards the end of the proof of Lemma E.6, we show that:

$$\langle H_{\boldsymbol\theta\ l,m}, P_{top\text{-}h}\rangle_F = \boldsymbol{g}_{\boldsymbol\theta\ l,m}^\top \text{vec}(P_{top\text{-}h}).$$

The expression of $\boldsymbol{g}_{\boldsymbol\theta\ l,m}$, which can be found in the proof of Lemma E.6, shows that computing $\langle H_{\boldsymbol\theta\ l,m}, P_{top\text{-}h}\rangle_F$ requires the computation of $r_{\boldsymbol\theta\ l,m} = \gamma_{\boldsymbol\theta\ l,m}$, $\boldsymbol{u}_{\boldsymbol\theta\ l,m}^\top \text{vec}(P_{top\text{-}h})$ and $\boldsymbol{v}_{\boldsymbol\theta\ l,m}^\top \text{vec}(P_{top\text{-}h})$. Because of Equation (24), the definitions of $\boldsymbol{u}_{\boldsymbol\theta\ l,m}$, $\boldsymbol{v}_{\boldsymbol\theta\ l,m}$ in the proof of Lemma E.6, the definition of the $\prec_{pos}^{top\text{-}h}$ censorship mechanism, being $I_{n,<}$ the adjacency matrix associated to the $\prec_{pos}^{full}$ censorship mechanism, we have that for $l, m \in [\![n]\!]$, $l \neq m$:

$$\boldsymbol{u}_{\boldsymbol\theta\ l,m}^\top \text{vec}(P_{top\text{-}h}) = \mathbb{P}_{\xi_{\boldsymbol\theta\ k}^{top\text{-}h}}\left[i_l^* \prec_{obs}^{(1)} i_m^*\right] - \mathbb{P}_{\xi_{\boldsymbol\theta\ k}^{top\text{-}h}}\left[i_m^* \prec_{obs}^{(1)} i_l^*\right]$$
$$= PL_{\boldsymbol\theta}\left[\pi(l) \in [\![h]\!], \pi(l) < \pi(m)\right]$$
$$- PL_{\boldsymbol\theta}\left[\pi(m) \in [\![h]\!], \pi(m) < \pi(l)\right],$$

$$\boldsymbol{v}_{\boldsymbol\theta\ l,m}^\top \text{vec}(P_{top\text{-}h}) = \mathbb{P}_{\xi_{\boldsymbol\theta\ k}^{top\text{-}h}}\left[i_l^* \prec_{obs}^{(1)} i_m^*\right] + \mathbb{P}_{\xi_{\boldsymbol\theta\ k}^{top\text{-}h}}\left[i_m^* \prec_{obs}^{(1)} i_l^*\right]$$
$$= PL_{\boldsymbol\theta}\left[\pi(l) \in [\![h]\!], \pi(l) < \pi(m)\right]$$

$$+ \text{PL}_{\boldsymbol{\theta}} \left[ \pi(m) \in [\![h]\!], \pi(m) < \pi(l) \right].$$

Thus, in virtue of Lemma F.2, we have:

$$\boldsymbol{u}_{\boldsymbol{\theta}\ l,m}^{\top} \text{vec}(P_{\text{top-}h}) \geq \frac{1}{2} \frac{h}{n} (\theta_l - \theta_m),$$

$$\boldsymbol{v}_{\boldsymbol{\theta}\ l,m}^{\top} \text{vec}(P_{\text{top-}h}) = \frac{\theta_l + \theta_m}{\theta_l - \theta_m} \boldsymbol{u}_{\boldsymbol{\theta}\ l,m}^{\top} \text{vec}(P_{\text{top-}h}).$$

Because of the properties of the PL model (Yellott, 1977):

$$r_{\boldsymbol{\theta}\ l,m} = \gamma_{\boldsymbol{\theta}\ l,m} = \text{PL}_{\boldsymbol{\theta}} \left[ \pi(l) < \pi(m) \right] - \text{PL}_{\boldsymbol{\theta}} \left[ \pi(m) < \pi(l) \right]$$

$$= \frac{\theta_l - \theta_m}{\theta_l + \theta_m}.$$

Finally:

$$\boldsymbol{g}_{\boldsymbol{\theta}\ l,m}^{\top} \text{vec}(P_{\text{top-}h}) = 2 \frac{\theta_l - \theta_m}{\theta_l + \theta_m} \boldsymbol{u}_{\boldsymbol{\theta}\ l,m}^{\top} \text{vec}(P_{\text{top-}h}) - \left( \frac{\theta_l - \theta_m}{\theta_l + \theta_m} \right)^2 \frac{\theta_l + \theta_m}{\theta_l - \theta_m} \boldsymbol{u}_{\boldsymbol{\theta}\ l,m}^{\top} \text{vec}(P_{\text{top-}h})$$

$$= \frac{\theta_l - \theta_m}{\theta_l + \theta_m} \boldsymbol{u}_{\boldsymbol{\theta}\ l,m}^{\top} \text{vec}(P_{\text{top-}h}) \geq \frac{1}{4} \frac{h}{n} (\theta_l - \theta_m)^2.$$

$\square$

Now we provide analogous results for the case of winner feedback and full-ranking feedback. The proofs follow the line of that of Lemma E.8, but are way simpler. For this reason, we will be brief.

**Lemma E.9.** *Let $n \in \mathbb{N}_{\geq 2}$ be the number of items. Let $\boldsymbol{\theta} = (\theta_l)_{l \in [\![n]\!]}$, with $1 \geq \theta_1 > \ldots \theta_n \geq 1/2$ be the parameters of a PL model. Consider the problem instance $\xi_{\boldsymbol{\theta},k}^{win} = \left( [\![n]\!], \text{PL}_{\boldsymbol{\theta}}, \delta(\prec_{pos}^{win}), e, k \right)$ where the item labeling is given by the identity permutation, the ranking model is PL and the distribution over the positional censorship mechanisms is a Dirac delta over the winner feedback. We replace $\boldsymbol{\theta}$ to $\rho$ and win to $\delta(\prec_{pos}^{win})$ in the subscripts to simplify the notation. Then, for every item labels $l, m \in [\![n]\!]$, $l < m$:*

$$\langle H_{\boldsymbol{\theta}\ l,m}, P_{win} \rangle_F \geq \frac{1}{2n} (\theta_l - \theta_m)^2.$$

*Proof.* Under these assumptions on the ranking model and the observation model:

$$\boldsymbol{u}_{\boldsymbol{\theta}\ l,m}^{\top} \text{vec}(P_{\text{win}}) = \frac{\theta_l - \theta_m}{\Theta},$$

$$\boldsymbol{v}_{\boldsymbol{\theta}\ l,m}^{\top} \text{vec}(P_{\text{win}}) = \frac{\theta_l + \theta_m}{\Theta}.$$

Finally:

$$\boldsymbol{g}_{\boldsymbol{\theta}\ l,m}^{\top} \text{vec}(P_{\text{win}}) = 2 \frac{\theta_l - \theta_m}{\theta_l + \theta_m} \frac{\theta_l - \theta_m}{\Theta} - \left( \frac{\theta_l - \theta_m}{\theta_l + \theta_m} \right)^2 \frac{\theta_l + \theta_m}{\Theta} = \frac{(\theta_l - \theta_m)^2}{(\theta_l + \theta_m)\Theta}$$

$$\geq \frac{1}{2n} (\theta_l - \theta_m)^2.$$

$\square$

**Lemma E.10.** *Let $n \in \mathbb{N}_{\geq 2}$ be the number of items. Let $\boldsymbol{\theta} = (\theta_l)_{l \in [\![n]\!]}$, with $1 \geq \theta_1 > \ldots \theta_n \geq 1/2$ be the parameters of a PL model. Consider the problem instance $\xi_{\boldsymbol{\theta}\ k}^{ful} = \left( [\![n]\!], \text{PL}_{\boldsymbol{\theta}}, \delta(\prec_{pos}^{full}), e, k \right)$ where the item labeling is given by the identity permutation, the ranking model is PL and the distribution over the positional censorship mechanisms is a Dirac delta over the full-ranking feedback. We replace $\boldsymbol{\theta}$ to $\rho$ and full to $\delta(\prec_{pos}^{full})$ in the subscripts to simplify the notation. Then, for every item labels $l, m \in [\![n]\!]$, $l < m$:*

$$\langle H_{\boldsymbol{\theta}\ l,m}, P_{full} \rangle_F \geq \frac{1}{4} (\theta_l - \theta_m)^2.$$

*Proof.* Under these assumptions on the ranking model and the observation model:

$$\boldsymbol{u}_{\boldsymbol{\theta}}^{\top}{}_{l,m} \operatorname{vec}(P_{\text{full}}) = \frac{\theta_l - \theta_m}{\theta_l + \theta_m},$$

$$\boldsymbol{v}_{\boldsymbol{\theta}}^{\top}{}_{l,m} \operatorname{vec}(P_{\text{full}}) = \frac{\theta_l + \theta_m}{\theta_l + \theta_m} = 1.$$

Finally:

$$\boldsymbol{g}_{\boldsymbol{\theta}}^{\top}{}_{l,m} \operatorname{vec}(P_{\text{full}}) = 2\frac{\theta_l - \theta_m}{\theta_l + \theta_m}\frac{\theta_l - \theta_m}{\theta_l + \theta_m} - \left(\frac{\theta_l - \theta_m}{\theta_l + \theta_m}\right)^2 = \left(\frac{\theta_l - \theta_m}{\theta_l + \theta_m}\right)^2$$

$$\geq \frac{1}{4}(\theta_l - \theta_m)^2.$$

$\square$

### E.4. PExact Lower Bounds in Restricted Settings

Here we show how it is possible to formally derive the PExact sample complexity lower bounds for the settings considered in Table 2. In particular, we will study the case in which the goal is to learn the full ranking under top-$h$ feedback. The other results can be derived analogously. To this end, we rely on the notation introduced in Appendix E.3.2. Furthermore, for given values $\boldsymbol{\Delta} := (\Delta_{l,l+1})_{l \in [\![n-1]\!]}$, $\Delta_{l,l+1} > 0$ for every $l \in [\![n-1]\!]$ and $\sum_{l \in [\![n-1]\!]} \Delta_{l,l+1} \leq 1/2$, we denote by $\mathcal{PL}(n, \boldsymbol{\Delta})$ the set of PL parameters $\boldsymbol{\theta}$ s.t. $1 \geq \theta_{\ell^{*-1}(1)}, \theta_{\ell^{*-1}(l)} - \theta_{\ell^{*-1}(l+1)} = \Delta_{l,l+1}$.[5]

**Theorem E.11** (Sample Complexity Lower Bound for Learning the Full Ranking from Top-$h$ Feedback). *Let $n \in \mathbb{N}_{\geq 2}$ be the number of items. Then, for every choice of $\boldsymbol{\Delta} = (\Delta_{l,l+1})_{l \in [\![n]\!]}$ with $\Delta_{l,l+1} > 0$, $\sum_{l \in [\![n-1]\!]} \Delta_{l,l+1} \leq 1/2$, $\delta$-correct learner Alg, number of top items $k \in [\![n]\!]$, top-$h$ observation model with $h \leq n/2$, and any confidence parameter $\delta \in (0, 1/2)$, we have that:*

$$\sup_{\boldsymbol{\theta} \in \mathcal{PL}(n,\boldsymbol{\Delta})} S_{\xi_{\boldsymbol{\theta}\ k}^{top-h}}^{(\delta)}(\text{Alg}) \geq \frac{1}{32} \frac{n}{h} \frac{1}{\min_{l \in [\![n-1]\!]} \Delta_{l,l+1}^2} \ln\left(\frac{1}{4\delta}\right).$$

*Proof.* We exploit Lemma E.5. Let $\boldsymbol{\theta} \in \mathcal{PL}(n, \boldsymbol{\Delta})$ with $1 \geq \theta_1 > \cdots > \theta_n \geq 1/2$. Let $\boldsymbol{\theta}^{(l)} = (\theta_{l'}^{(l)})_{l' \in [\![n]\!]}$ be the same instance, but with parameters $\theta_l$ and $\theta_{l+1}$ swapped (for a fixed $l \in [\![n-1]\!]$). Observe that $\boldsymbol{\theta}^{(l)} \in \mathcal{PL}(n, \boldsymbol{\Delta})$. Let $\mathcal{A}$ be the event that a $\delta$-correct learner returns the identity permutation as the learned ranking. This event has probability $1 - \delta$ under instance $\boldsymbol{\theta}$ and probability $\delta$ under instance $\boldsymbol{\theta}^{(l)}$, as swapping the parameters changes the true ranking. By Lemma F.4, we can lower bound the right side of the inequality of Lemma E.5 with $\ln(1/(4\delta))$. To conclude the proof, we must upper bound the KL divergence between the observations generated by $\boldsymbol{\theta}$ and $\boldsymbol{\theta}^{(l)}$ under top-$h$ feedback. Let $\mathbb{P}_{\boldsymbol{\theta}}^{\text{top-}h}$ be the distribution over the tuples $(I_1, \ldots, I_h)$ with $I_r := \pi^{-1}(r)$, $r \in [\![h]\!]$ and $\pi \sim \text{PL}_{\boldsymbol{\theta}}$. $\mathbb{P}_{\boldsymbol{\theta}^{(l)}}^{\text{top-}h}$ is defined analogously. Using the chain rule of the KL divergence, (Popescu et al., 2016, Theorem 1.4), and the properties of the PL model, we get:

$$D_{\text{KL}}\left(\mathbb{P}_{\boldsymbol{\theta}}^{\text{top-}h} \| \mathbb{P}_{\boldsymbol{\theta}^{(l)}}^{\text{top-}h}\right) \leq \sum_{r=1}^{h} \mathbb{E}_{I_1,\ldots,I_{r-1} \sim \mathbb{P}_{\boldsymbol{\theta}}^{\text{top-}h}} \left[ \sum_{i \in [\![n]\!] \setminus \{I_1,\ldots,I_{r-1}\}} \frac{\mathbb{P}_{\boldsymbol{\theta}}^{\text{top-}h}[I_r = i \mid I_1, \ldots, I_{r-1}]^2}{\mathbb{P}_{\boldsymbol{\theta}^{(l)}}^{\text{top-}h}[I_r = i \mid I_1, \ldots, I_{r-1}]} - 1 \right]$$

$$= \sum_{r=1}^{h} \mathbb{E}_{I_1,\ldots,I_{r-1} \sim \mathbb{P}_{\boldsymbol{\theta}}^{\text{top-}h}} \left[ \frac{\Theta_{r-1}^{(l)}}{\Theta_{r-1}^2} \sum_{i \in [\![n]\!] \setminus \{I_1,\ldots,I_{r-1}\}} \frac{\theta_i^2}{\theta_i^{(l)}} - 1 \right],$$

where $\Theta_{r-1} := \sum_{j \in [\![n]\!] \setminus \{I_1,\ldots,I_{r-1}\}} \theta_j$ and $\Theta_{r-1}^{(l)}$ is defined analogously from the parameters $\boldsymbol{\theta}^{(l)}$. By checking all cases induced by the predicates $l \in \{I_1, \ldots, I_{r-1}\}, l+1 \in \{I_1, \ldots, I_{r-1}\}$ it is possible to show that the maximum of the quantity inside the expectation is attained when $\{l, l+1\} \cap \{I_1, \ldots, I_{r-1}\} = \{\}$, where by straightforward algebraic manipulation:

$$\frac{\Theta_{r-1}^{(l)}}{\Theta_{r-1}^2} \sum_{i \in [\![n]\!] \setminus \{I_1,\ldots,I_{r-1}\}} \frac{\theta_i^2}{\theta_i^{(l)}} - 1 = \frac{1}{\theta_{r-1}} \left( \sum_{i \in [\![n]\!] \setminus \{I_1,\ldots,I_{r-1},l,l+1\}} \frac{\theta_i^2}{\theta_i} + \frac{\theta_l^2}{\theta_l^{(l)}} + \frac{\theta_{l+1}^2}{\theta_{l+1}^{(l)}} \right) - 1$$

___

[5]As in the definition of the instance in Section 3, $\ell^*$ sorts the parameters in descending order.

$$= \frac{(\theta_l - \theta_{l+1})^2(\theta_l + \theta_{l+1})}{\Theta_{r-1}\theta_l\theta_{l+1}} \leq \frac{32}{n}\Delta^2_{l,l+1},$$

where the last line holds because the parameters are bounded ($\theta_i \in [1/2, 1]$) and by the assumption $h \leq n/2$, which implies $\Theta_{r-1} \geq (n-h)/2 \geq n/4$. By substituting such an upper bound inside the expectation, we obtain:

$$D_{\mathrm{KL}}\left(\mathbb{P}^{\text{top-}h}_{\boldsymbol{\theta}}\|\mathbb{P}^{\text{top-}h}_{\boldsymbol{\theta}^{(l)}}\right) \leq 32\frac{h}{n}\Delta^2_{l,l+1}.$$

Rearranging the inequality of Lemma E.5 completes the proof. □

## F. Auxiliary Results

This appendix presents the proofs of the auxiliary results utilized throughout our manuscript.

### F.1. Ranking Models

The next result provides a characterization of the full support property for a ranking model $\rho \in \Delta(S_n)$ satisfying Assumption 3.1.

**Theorem F.1** (Full Support Characterization under Complete Consensus of a Ranking Model). *Let $\rho \in \Delta(S_n)$ be a ranking model satisfying Assumption 3.1. Then the reverse permutation satisfies $\omega = \arg\min_{\pi \in S_n} \rho(\pi)$. This implies that:*

$$\rho(\pi) > 0 \text{ for every } \pi \in S_n \text{ iff } \rho(\omega) > 0.$$

*Proof.* We proceed by induction on the number of correctly ranked pairs $(l, m)$ by a ranking $\pi \in S_n$, i.e. $C_\pi = |\{l \in [\![n-1]\!], m \in [\![l+1,n]\!] \text{ s.t } \pi(l) < \pi(m)\}|$, showing that, in every case $\rho(\pi) \geq \rho(\omega)$, with equality if and only if $\pi = \omega$.

*Base case ($C_\pi = 0$):* In this case we know that if $l < m$ then $\pi(l) > \pi(m)$, which implies directly that $\pi = \omega$ ($n$ has rank 1, $n-1$ has rank 2, and so on).

*Inductive step ($C_\pi > 0$):* Because of what we remarked in the base case, it must be $\pi \neq \omega$, which implies $\pi^{-1} \neq \omega^{-1} = \omega$. Then, there must be $r \in [\![n-1]\!]$ such that $\pi^{-1}(r) < \pi^{-1}(r+1)$ (otherwise $\pi^{-1} = \omega$). Let $l = \pi^{-1}(r)$, $m = \pi^{-1}(r+1)$, and $\pi' = \pi \circ \tau_{l,m}$. Now $\pi'(l) = \pi(m) = r+1 > r = \pi(l) = \pi'(m)$, but $l = \pi^{-1}(r) < \pi^{-1}(r+1) = m$, so that $l$ and $m$ are now in the wrong order in $\pi'$. If $q \in [\![n]\!] \setminus \{l, m\}$, then either:

$$\pi(q) < r = \pi(l) = \pi'(m) < \pi'(l) = \pi(m), \text{ or}$$

$$\pi(q) > r + 1 = \pi(m) = \pi'(l) > \pi'(m) = \pi(l),$$

so that the relative ordering of the pairs $(q, l)$ and $(q, m)$ remains unchanged. Thus $C_{\pi'} = C_\pi - 1$. Finally, because of the strict complete consensus of $\rho$:

$$\rho(\pi) > \rho(\pi \circ \tau_{l,m}) = \rho(\pi'),$$

concluding the proof. □

The following result is of particular interest in the study of the sample complexity when the observations are generated according to the PL ranking model with top-$h$ feedback.

**Lemma F.2.** *Let $\boldsymbol{\theta} = (\theta_l)_{l=1}^n$, with $1 \geq \theta_1 > \cdots > \theta_n \geq 1/2$ be the parameters of a PL model. Then, for every $h \in [\![\lfloor n/20 \rfloor]\!]$, and item labels $l, m \in [\![n]\!]$, $l < m$, it holds that:*

$$\mathrm{PL}_{\boldsymbol{\theta}}\left[\pi(l) \in [\![h]\!], \pi(l) < \pi(m)\right] - \mathrm{PL}_{\boldsymbol{\theta}}\left[\pi(m) \in [\![h]\!], \pi(m) < \pi(l)\right] \geq \frac{1}{2}\frac{h}{n}(\theta_l - \theta_m),$$

$$\mathrm{PL}_{\boldsymbol{\theta}}\left[\pi(l) \in [\![h]\!], \pi(l) < \pi(m)\right] + \mathrm{PL}_{\boldsymbol{\theta}}\left[\pi(m) \in [\![h]\!], \pi(m) < \pi(l)\right]$$

$$= \frac{\theta_l + \theta_m}{\theta_l - \theta_m}\left(\mathrm{PL}_{\boldsymbol{\theta}}\left[\pi(l) \in [\![h]\!], \pi(l) < \pi(m)\right] - \mathrm{PL}_{\boldsymbol{\theta}}\left[\pi(m) \in [\![h]\!], \pi(m) < \pi(l)\right]\right).$$

*Proof.* Given a subset of the item labels: $S \subseteq [\![n]\!]$ with $|S| = r$, we denote by $\mathcal{C}_S^{(r)} = \{S = \pi^{-1}([\![r]\!])\}$ the event in which the top-$r$ items in the ranking correspond to the set $S$. Observe that, due to a well-known property of the PL model (Yellott, 1977):

$$\text{PL}_{\boldsymbol{\theta}}\left[\pi(l) = r + 1 \mid \mathcal{C}_S^{(r)}\right] = \frac{\theta_l}{\Theta - \sum_{m \in S} \theta_s} \text{ for every } l \in [\![n]\!], r \in [\![n-1]\!], S \subseteq [\![n]\!] \setminus \{l\}, |S| = r.$$

Let us exploit this property to decompose the expressions that we want to bound:

$$\begin{aligned}
\text{PL}_{\boldsymbol{\theta}}\left[\pi(l) \in [\![h]\!], \pi(l) < \pi(m)\right] &= \sum_{r \in [\![h]\!]} \text{PL}_{\boldsymbol{\theta}}\left[\pi(l) = r, r < \pi(m)\right] \\
&= \sum_{r \in [\![h]\!]} \sum_{\substack{S \subseteq [\![n]\!] \setminus \{l,m\} \\ |S| = r-1}} \text{PL}_{\boldsymbol{\theta}}\left[\pi(l) = r \mid \mathcal{C}_S^{(r-1)}\right] \text{PL}_{\boldsymbol{\theta}}\left[\mathcal{C}_S^{(r-1)}\right] \\
&= \sum_{r \in [\![h]\!]} \sum_{\substack{S \subseteq [\![n]\!] \setminus \{l,m\} \\ |S| = r-1}} \frac{\theta_l}{\Theta - \sum_{q \in S} \theta_q} \text{PL}_{\boldsymbol{\theta}}\left[\mathcal{C}_S^{(r-1)}\right]. \quad (25)
\end{aligned}$$

Then:

$$\text{PL}_{\boldsymbol{\theta}}\left[\pi(l) \in [\![h]\!], \pi(l) < \pi(m)\right] - \text{PL}_{\boldsymbol{\theta}}\left[\pi(m) \in [\![h]\!], \pi(m) < \pi(l)\right] = \sum_{r \in [\![h]\!]} \sum_{\substack{S \subseteq [\![n]\!] \setminus \{l,m\} \\ |S| = r-1}} \frac{\theta_l - \theta_m}{\Theta - \sum_{q \in S} \theta_q} \text{PL}_{\boldsymbol{\theta}}\left[\mathcal{C}_S^{(r-1)}\right],$$

$$\text{PL}_{\boldsymbol{\theta}}\left[\pi(l) \in [\![h]\!], \pi(l) < \pi(m)\right] + \text{PL}_{\boldsymbol{\theta}}\left[\pi(m) \in [\![h]\!], \pi(m) < \pi(l)\right] = \sum_{r \in [\![h]\!]} \sum_{\substack{S \subseteq [\![n]\!] \setminus \{l,m\} \\ |S| = r-1}} \frac{\theta_l + \theta_m}{\Theta - \sum_{q \in S} \theta_q} \text{PL}_{\boldsymbol{\theta}}\left[\mathcal{C}_S^{(r-1)}\right].$$

The second equality follows. We now just have to prove the lower bound to the difference of probabilities. Observe that, since $1 \geq \theta_1 > \cdots > \theta_n \geq 1/2$, we have:

$$\Theta - \sum_{q \in S} \theta_s \in \left[\frac{n - h + 1}{2}, n\right].$$

To get the desired result, we need to lower bound the probability of the event $\mathcal{A}_{l,m}^{(r)} = \{l, m \notin \pi^{-1}([\![r]\!])\}$, since:

$$\text{PL}_{\boldsymbol{\theta}}\left[\mathcal{A}_{l,m}^{(r)}\right] = \sum_{\substack{S \subseteq [\![n]\!] \setminus \{l,m\} \\ |S| = r-1}} \text{PL}_{\boldsymbol{\theta}}\left[\mathcal{C}_S^{(r-1)}\right].$$

Let us consider the complementary event: $\overline{\mathcal{A}_{l,m}^{(r)}} = \{l \in \pi^{-1}([\![r]\!]) \text{ or } m \in \pi^{-1}([\![r]\!])\}$. Exploiting the assumption $h \leq n/20$ and a decomposition completely analogous to the one we did above (Equation (25)), we get:

$$\text{PL}_{\boldsymbol{\theta}}\left[l \in \pi^{-1}([\![r]\!])\right] \leq \sum_{s \in [\![r]\!]} \frac{2\theta_l}{n - h + 1} \sum_{\substack{S \subseteq [\![n]\!] \setminus \{l\} \\ |S| = s-1}} \text{PL}_{\boldsymbol{\theta}}\left[\mathcal{C}_S^{(s-1)}\right] \leq \frac{2r}{n - h + 1} \cdot 1$$

$$\leq \frac{2h}{n - h + 1} \leq \frac{2h}{n - h} \leq \frac{2}{19},$$

being $h \leq n/20$. The same bound holds for $\text{PL}_{\boldsymbol{\theta}}\left[m \in \pi^{-1}([\![r]\!])\right]$, hence, thanks to a union bound, we get $\text{PL}_{\boldsymbol{\theta}}\left[\overline{\mathcal{A}_{l,m}^{(r)}}\right] \leq \frac{4}{19}$. Thus:

$$\text{PL}_{\boldsymbol{\theta}}\left[\mathcal{A}_{l,m}^{(r)}\right] = 1 - \text{PL}_{\boldsymbol{\theta}}\left[\overline{\mathcal{A}_{l,m}^{(r)}}\right] \geq \frac{15}{19} \geq \frac{1}{2}.$$

This lower bound, together with the upper bound to $\Theta - \sum_{q \in S} \theta_s$ and the fact that $\theta_l > \theta_m$, allows to derive the lower bound for the difference of the probabilities. $\square$

### F.2. Kullback-Leibler Divergence

The following result provides a distribution-dependent refinement of the well-known Pinsker inequality for Bernoulli distributions.

**Lemma F.3** (Refined Pinsker Inequality for Bernoulli Distributions). *Let $p, q \in (0, 1)$ such that $q < p$. Then:*

$$D_{\mathrm{KL}}(\mathrm{Be}(q)\|\mathrm{Be}(p)) \geq \frac{(p-q)^2}{2p}.$$

*Proof.* We fix $p \in (0, 1)$ and consider the divergence as a function of $q$. Let $h(q) = D_{\mathrm{KL}}(\mathrm{Be}(q)\|\mathrm{Be}(p))$. Explicitly, for $q \in (0, 1)$, this is given by:

$$h(q) = q \ln \frac{q}{p} + (1 - q) \ln \frac{1-q}{1-p}.$$

We apply Taylor's Theorem to expand $h(q)$ around the point $p$. The expansion of $h(q)$ centered at $p$ with the Lagrange remainder is:

$$h(q) = h(p) + h'(p)(q - p) + \frac{1}{2}h''(\xi)(q - p)^2,$$

where $\xi \in (q, p)$. First, we calculate the derivatives of $h(q)$ with respect to $q$:

$$h'(q) = \ln \frac{q}{1-q} - \ln \frac{p}{1-p},$$
$$h''(q) = \frac{1}{q} + \frac{1}{1-q} = \frac{1}{q(1-q)}.$$

Evaluating at $p$, we have $h(p) = D_{\mathrm{KL}}(\mathrm{Be}(p)\|\mathrm{Be}(p)) = 0$. Similarly, $h'(p) = 0$. Thus, the Taylor expansion simplifies to:

$$D_{\mathrm{KL}}(\mathrm{Be}(q)\|\mathrm{Be}(p)) = \frac{1}{2}\frac{1}{\xi(1-\xi)}(q - p)^2.$$

Since $\xi \in (q, p)$, we have $0 < \xi < p$. Consequently, $1 - \xi < 1$, which implies $\xi(1 - \xi) < \xi < p$. Therefore:

$$\frac{1}{\xi(1-\xi)} > \frac{1}{p}.$$

Substituting this inequality back into the expansion yields:

$$D_{\mathrm{KL}}(\mathrm{Be}(q)\|\mathrm{Be}(p)) = \frac{(p-q)^2}{2\xi(1-\xi)} > \frac{(p-q)^2}{2p}.$$

$\square$

A slight variant of the following result is mentioned in (Kaufmann et al., 2016).

**Lemma F.4** (Lower Bound to the Kullback-Leibler Divergence of Separated Bernoulli). *Let $\delta \in (0, 1/2)$. Let $p, q \in (0, 1)$ such that $\min(p, q) < \delta$ and $\max(p, q) > 1 - \delta$. Then:*

$$D_{\mathrm{KL}}(\mathrm{Be}(q)\|\mathrm{Be}(p)) \geq \ln\left(\frac{1}{4\delta}\right).$$

*Proof.* Without loss of generality, we assume $q < \delta$ and $p > 1 - \delta$. The converse case, where $p < \delta$ and $q > 1 - \delta$, follows identically by considering the complementary probabilities. Because $D_{\mathrm{KL}}(\mathrm{Be}(q)\|\mathrm{Be}(p)) = D_{\mathrm{KL}}(\mathrm{Be}(1 - q)\|\mathrm{Be}(1 - p))$, letting $\tilde{q} = 1 - q$ and $\tilde{p} = 1 - p$ yields $\tilde{q} < \delta$ and $\tilde{p} > 1 - \delta$, allowing us to proceed with the primary assumption. We rely on the monotonicity of the Kullback-Leibler divergence. For a fixed $q$, the function $x \mapsto D_{\mathrm{KL}}(\mathrm{Be}(q)\|\mathrm{Be}(x))$ is strictly increasing on the interval $(q, 1)$. Since $p > 1 - \delta$ and $q < \delta < 1 - \delta$, we have:

$$D_{\mathrm{KL}}(\mathrm{Be}(q)\|\mathrm{Be}(p)) \geq D_{\mathrm{KL}}(\mathrm{Be}(q)\|\mathrm{Be}(1 - \delta)).$$

Let $g(q, \delta) = D_{\mathrm{KL}}(\mathrm{Be}(q) \| \mathrm{Be}(1 - \delta))$. Expanding the definition:

$$g(q, \delta) = \ln \frac{1}{\delta} + [q \ln q + (1 - q) \ln(1 - q)] - q \ln(1 - \delta) - q \ln \frac{1}{\delta}.$$

We wish to show that $g(q, \delta) \geq \ln(1/4\delta) = \ln(1/\delta) - \ln 4$. This is equivalent to showing:

$$\ln 4 \geq -(q \ln q + (1 - q) \ln(1 - q)) + q \ln(1 - \delta) + q \ln \frac{1}{\delta}.$$

Let $\phi(q, \delta)$ denote the right-hand side. For $q < \delta < 1/2$, $\phi(q, \delta)$ is strictly increasing in $q$. Thus, it suffices to prove the bound for the limit $\phi(\delta)$:

$$\phi(\delta) = -2\delta \ln \delta - (1 - 2\delta) \ln(1 - \delta).$$

We analyze $\phi(\delta)$ on the interval $(0, 1/2)$ using Taylor's Theorem. First, we compute the first two derivatives:

$$\phi'(\delta) = 2 \ln \left( \frac{1 - \delta}{\delta} \right) - 1 - \frac{\delta}{1 - \delta},$$

$$\phi''(\delta) = -\frac{2}{\delta(1 - \delta)} - \frac{1}{(1 - \delta)^2}.$$

For $\delta \in (0, 1/2)$, we observe that $\delta(1 - \delta) > 0$ and $(1 - \delta)^2 > 0$. Therefore, $\phi''(\delta) < 0$ for every $\delta \in (0, 1/2)$, meaning $\phi$ is strictly concave.

We apply Taylor's Theorem centered at $\delta_0 = 1/3$. For any $\delta \in (0, 1/2)$, there exists $\xi$ between $\delta$ and $1/3$ such that:

$$\phi(\delta) = \phi(1/3) + \phi'(1/3) \left( \delta - \frac{1}{3} \right) + \frac{1}{2} \phi''(\xi) \left( \delta - \frac{1}{3} \right)^2.$$

Since $\phi''(\xi) < 0$, we can drop the quadratic term to obtain a linear upper bound:

$$\phi(\delta) \leq \phi(1/3) + \phi'(1/3) \left( \delta - \frac{1}{3} \right).$$

We evaluate the function and its slope at $1/3$:

$$\phi(1/3) = \ln 3 - \frac{1}{3} \ln 2 < 1,$$

$$\phi'(1/3) = 2 \ln 2 - 1 - \frac{1/3}{2/3} = \ln 4 - 3/2 < 0.$$

Thus the linear upper bound to $\phi$ reaches its supremum for $\delta \to 0$, implying:

$$\phi(\delta) \leq 1 + \frac{3/2 - \ln(4)}{3}.$$

Hence, $\ln 4 \geq \phi(\delta)$ is implied by $\frac{4}{3} \ln(4) \geq 3/2$ which is a true inequality, concluding our proof. $\qquad \square$

### F.3. Graph Theory

The next result addresses a bottleneck optimization problem on directed acyclic graphs (DAGs). Formally, this is a specific instance of the *minimum bottleneck spanning arborescence* (MBSA) problem (Camerini, 1978). While the general MBSA problem requires complex algorithms, the acyclic structure of our graph allows for a direct closed-form solution.

**Lemma F.5** (Minimum Bottleneck Spanning Arborescence). *Let $n \in \mathbb{N}_{\geq 2}$, $G = (\llbracket n \rrbracket, E)$ with $E = \{(l, m) \in \llbracket n \rrbracket^2$ s.t. $l < m\}$ be a complete DAG. Let $S_{l,m} > 0$ denote the cost associated with edge $(l, m) \in E$. The minimum bottleneck cost $S^*$ required to select a subgraph $G' = (\llbracket n \rrbracket, E')$ where $E' \subseteq E$, such that $1$ is a root (i.e., $1$ can reach all other nodes in $G'$) is given by:*

$$S^* := \min_{E' \subseteq E, \, 1 \text{ is root}} \max_{(l,m) \in E'} S_{l,m} = \max_{m \in \llbracket 2, n \rrbracket} \min_{l \in \llbracket m-1 \rrbracket} S_{l,m}.$$

*Proof.* We denote the target value by $\hat{S}^* = \max_{m \in [\![2,n]\!]} \min_{l \in [\![m-1]\!]} S_{l,m}$. We prove the equality $S^* = \hat{S}^*$ by establishing both an upper and a lower bound.

*Upper Bound ($S^* \leq \hat{S}^*$):* We construct a specific set of edges $E_{\text{greedy}}$ by selecting, for each node $m \in [\![2,n]\!]$, an incoming edge that minimizes the cost. Formally, for each $m$, we choose a parent index $l^*(m) < m$ such that $S_{l^*(m),m} = \min_{l \in [\![m-1]\!]} S_{l,m}$. The resulting set of edges is $E_{\text{greedy}} = \{(l^*(m), m) \text{ s.t. } m \in [\![2,n]\!]\}$.

Since every node $m > 1$ has exactly one incoming edge and $l^*(m) < m$, traversing the path from any node $m$ backwards to its parent results in a strictly decreasing sequence of indices. As the index is lower bounded by 1, all such paths must eventually terminate at node 1. Thus, the subgraph $G' = ([\![n]\!], E_{\text{greedy}})$ forms a valid spanning arborescence rooted at 1. The bottleneck cost of this subgraph is:

$$\max_{(l,m) \in E_{\text{greedy}}} S_{l,m} = \max_{m \in [\![2,n]\!]} S_{l^*(m),m} = \max_{m \in [\![2,n]\!]} \min_{l \in [\![m-1]\!]} S_{l,m} = \hat{S}^*.$$

Since $S^*$ is the minimum cost over all valid subgraphs, it follows that $S^* \leq \hat{S}^*$.

*Lower Bound ($S^* \geq \hat{S}^*$):* Let $G' = ([\![n]\!], E')$ be any subgraph where node 1 is a root. By the definition of a rooted graph, for every node $m \in [\![2,n]\!]$ there must exist a path from 1 to $m$, which implies that $m$ must have at least one incoming edge in $E'$. Let $(l(m), m) \in E'$ denote one such incoming edge for node $m$.

The cost of this specific edge is naturally lower-bounded by the minimum possible incoming cost for $m$:

$$S_{l(m),m} \geq \min_{l \in [\![m-1]\!]} S_{l,m}.$$

The bottleneck cost of the subgraph $G'$ is the maximum cost over all selected edges. Since $E'$ must contain at least one incoming edge for every $m \in [\![2,n]\!]$, we have:

$$\max_{(l,m) \in E'} S_{l,m} \geq \max_{m \in [\![2,n]\!]} S_{l(m),m} \geq \max_{m \in [\![2,n]\!]} \min_{l \in [\![m-1]\!]} S_{l,m} = \hat{S}^*.$$

As this inequality holds for any valid edge set $E'$, we conclude that $S^* \geq \hat{S}^*$.

Combining both bounds, we obtain $S^* = \hat{S}^*$. $\qquad\square$

The results which follow give a characterization of the existence of certain paths in a DAG in terms of the out-degree of its nodes in the graph obtained through transitive closure.

**Lemma F.6.** *Consider the directed acyclic graph (DAG) $G = ([\![n]\!], E)$ where $(l, m) \in E$ only if $l < m$ for every $l, m \in [\![n]\!]$. Then vertex 1 is a root (i.e., there is a path from 1 to every node $l \in [\![2,n]\!]$) if and only if the out-degree of 1 in the transitive closure of $G$, denoted by $G^+ = ([\![n]\!], \prec)$, is $d_{out}^+(1) = n - 1$.*

*Proof.* By definition, $u \prec v$ if and only if there is a path from $u$ to $v$ in $G$.

*If:* Assume 1 is a root. By definition, there exists a path from 1 to every node $v \in [\![n]\!] \setminus \{1\}$. Therefore, in the transitive closure $G^+$, there is an edge $(1, v)$ for every $v \in [\![n]\!] \setminus \{1\}$. Consequently, the out-degree of 1 in $G^+$ is exactly $|[\![n]\!]| = n - 1$, since there cannot be a path from 1 to itself as this would form a cycle.

*Only if:* Assume $d_{out}^+(1) = n - 1$ in $G^+$. Since the total number of vertices is $n$, the set of out-neighbors of 1 in $G^+$ must be exactly $[\![n]\!] \setminus \{1\}$, as we remarked before that there cannot be a path from 1 to itself. This implies that for every $v \in [\![n]\!] \setminus \{1\}$, we have $1 \prec v$. By the definition of transitive closure, $1 \prec v$ implies the existence of a path from 1 to $v$ in the original graph $G$. Thus, 1 can reach every node in $[\![2,n]\!]$, making it a root. $\qquad\square$

**Lemma F.7.** *Consider the directed acyclic graph (DAG) $G = ([\![n]\!], E)$ where $(l, m) \in E$ only if $l < m$ for every $l, m \in [\![n]\!]$. Let $k \in [\![n]\!]$. Denote by $G^+ = ([\![n]\!], \prec)$ the transitive closure of $G$. Then, if $k \prec l$ for every $l \in [\![k+1, n]\!]$, it holds that:*

$$(l, l+1) \in E \text{ for every } l \in [\![k-1]\!] \text{ iff } d_{out}^+(l) = d_{out}^+(l+1) + 1 \text{ for every } l \in [\![k-1]\!],$$

*where $d_{out}^+$ is the out-degree of the nodes in $G^+$.*

*Proof.* Let $R(u) = \{v \in [\![n]\!] \text{ s.t. } u \prec v\}$ denote the set of nodes reachable from $u$ in $G$ (which corresponds to the out-neighbors of $u$ in $G^+$). By assumption: $R(k) = [\![k+1, n]\!]$. It follows from Lemma F.6 applied to the graph $([\![k, n]\!], E \cap [\![k, n]\!]^2)$ that $d_{\text{out}}^+(k) = n - k$.

*Only if:* Assume $(l, l+1) \in E$ for every $l \in [\![k-1]\!]$. We proceed by backward induction. Consider $l = k - 1$. Since $(k - 1, k) \in E$, we have $R(k) \subset R(k-1)$. Specifically, $R(k-1) = \{k\} \cup R(k)$ because any path starting at $k - 1$ moves to a node $v > k$. Thus, $d_{\text{out}}^+(k-1) = 1 + d_{\text{out}}^+(k)$. By iterating this logic for $l = k - 2, \ldots, 1$, we find that $R(l) = \{l+1\} \cup R(l+1)$, and therefore $d_{\text{out}}^+(l) = d_{\text{out}}^+(l+1) + 1$ for every $l \in [\![k-1]\!]$.

*If:* Assume $d_{\text{out}}^+(l) = d_{\text{out}}^+(l+1) + 1$ for every $l \in [\![k-1]\!]$. We again proceed by backward induction starting from $l = k - 1$. We know $R(k) = [\![k+1, n]\!]$ and $d_{\text{out}}^+(k) = n - k$. The hypothesis gives $d_{\text{out}}^+(k-1) = d_{\text{out}}^+(k) + 1 = n - k + 1$. The maximum possible reachable set for $k - 1$ is $[\![k, n]\!]$, which has size $n - k + 1$. Since the size of $R(k-1)$ matches this maximum, it must be that $R(k-1) = [\![k, n]\!]$. This implies $k - 1 \prec k$. By definition of the transitive closure, there is a path from $k - 1$ to $k$. Since edges exist only for $u < v$, a path from $k - 1$ to $k$ cannot involve any intermediate node $w$ (as there is no integer $w$ such that $k - 1 < w < k$). Therefore, the direct edge $(k - 1, k)$ must exist in $E$. Repeating this argument for $k - 2, \ldots, 1$ proves that $(l, l+1) \in E$ for every $l \in [\![k-1]\!]$. $\qquad\square$

### F.4. General Inequalities

The following result is adapted from (Kaufmann et al., 2016, Lemma 22). It provides an inequality relating linear and logarithmic growth, which is useful in sample complexity bounds.

**Lemma F.8.** *Let $a, b > 0$ such that $ab > 1$. Then, for any $T \in \mathbb{R}$,*

$$T \geq 2a \ln(ab) \text{ implies } T \geq a \ln(bT).$$

*Proof.* Consider the function $f(t) = t - a \ln(bt)$ defined for $t > 0$. We calculate its first derivative:

$$f'(t) = 1 - \frac{a}{t}.$$

The function $f(t)$ achieves its minimum at $t = a$ and is strictly increasing for $t > a$.

Let $T_0 = 2a \ln(ab)$. We first verify that $f(T_0) \geq 0$. Substituting $T_0$ into $f(t)$:

$$\begin{aligned}
f(T_0) &= 2a \ln(ab) - a \ln(2ab \ln(ab)) \\
&= a\left[2\ln(ab) - \ln(2ab) - \ln(\ln(ab))\right] \\
&= a\left[\ln((ab)^2) - \ln(2ab \ln(ab))\right] \\
&= a \ln\left(\frac{(ab)^2}{2ab \ln(ab)}\right) \\
&= a \ln\left(\frac{ab}{2\ln(ab)}\right).
\end{aligned}$$

Consider the auxiliary function $g(u) = u - 2\ln u$ for $u > 1$. Its derivative is $g'(u) = 1 - 2/u$, which is positive for $u > 2$. The minimum occurs at $u = 2$, where $g(2) = 2 - 2\ln 2 > 0$. Thus, $u \geq 2\ln u$ for every $u > 0$. Applying this with $u = ab$ (since $ab > 1$), we have $\frac{ab}{2\ln(ab)} \geq 1$. Consequently, the logarithm is non-negative, and $f(T_0) \geq 0$.

Now we consider two cases for the monotonicity:

1. If $ab \geq \sqrt{e}$, then $\ln(ab) \geq 1/2$, which implies $T_0 = 2a \ln(ab) \geq a$. Since $T \geq T_0 \geq a$ and $f$ is increasing on $[a, \infty)$, we have $f(T) \geq f(T_0) \geq 0$.
2. If $1 < ab < \sqrt{e}$, then the minimum of $f$ occurs at $t = a$. The value at the minimum is $f(a) = a - a \ln(ab) = a(1 - \ln(ab))$. Since $ab < \sqrt{e} < e$, we have $\ln(ab) < 1$, so $f(a) > 0$. Thus, $f(t) > 0$ for every $t > 0$, and the inequality holds trivially.

In both cases, $T \geq a \ln(bT)$. $\qquad\square$

The following result establishes a linear lower bound for a scalar function of interest.

**Lemma F.9.** *Let $x \in [0, 1]$. Then:*

$$1 - \exp(-x) \geq \frac{1}{2}x.$$

*Consequently, for any constant $c \in [0, 1]$, we have:*

$$1 - \exp\left(-\frac{1}{8}c\right) \geq \frac{1}{16}c.$$

*Proof.* We apply Taylor's Theorem with the Lagrange remainder to the function $g(x) = \exp(-x)$ around the point 0. For any $x > 0$, there exists some $\xi \in (0, x)$ such that:

$$\exp(-x) = 1 - x + \frac{x^2}{2}\exp(-\xi).$$

Substituting this expansion into the expression of interest:

$$1 - \exp(-x) = 1 - \left(1 - x + \frac{x^2}{2}\exp(-\xi)\right) = x - \frac{x^2}{2}\exp(-\xi).$$

To prove the lemma, we must show that $x - \frac{x^2}{2}\exp(-\xi) \geq \frac{1}{2}x$. Rearranging the terms, this inequality is equivalent to:

$$\frac{1}{2}x \geq \frac{x^2}{2}\exp(-\xi) \text{ iff } 1 \geq x\exp(-\xi).$$

Since $x \in [0, 1]$ and $\xi > 0$ (which implies $\exp(-\xi) < 1$), the product $x\exp(-\xi)$ is strictly less than 1. Thus, the inequality holds.

The second part follows directly by setting $x = c/8$. Since $c \in [0, 1]$, we have $x \in [0, 1/8] \subset [0, 1]$, so the bound applies. $\qquad\square$

The following lemma establishes a first-order lower bound for a quadratic-over-linear function. This result relies on the convexity of the function over the domain where the denominator is positive.

**Lemma F.10.** *Let $u, v \in \mathbb{R}^n$ and let $\mathcal{D} = \{x \in \mathbb{R}^n \text{ s.t. } v^T x > 0\}$. Consider the function $f : \mathcal{D} \to \mathbb{R}$ defined by:*

$$f(x) = \frac{(u^T x)^2}{v^T x}.$$

*For any $x, x_0 \in \mathcal{D}$, the following inequality holds:*

$$f(x) \geq \nabla f(x_0)^T x,$$

*where the gradient is given by:*

$$\nabla f(x_0) = 2\frac{u^T x_0}{v^T x_0}u - \left(\frac{u^T x_0}{v^T x_0}\right)^2 v.$$

*Proof.* We first derive the expression for the gradient $\nabla f(x)$. Let $N(x) = (u^T x)^2$ and $D(x) = v^T x$. The gradients of the numerator and denominator are $\nabla N(x) = 2(u^T x)u$ and $\nabla D(x) = v$, respectively. Applying the quotient rule, we obtain:

$$\nabla f(x) = \frac{D(x)\nabla N(x) - N(x)\nabla D(x)}{D(x)^2} = \frac{2(u^T x)(v^T x)u - (u^T x)^2 v}{(v^T x)^2}.$$

Evaluating this at $x_0$ yields the expression provided in the statement.

To prove the inequality, we define the difference $\Delta = f(x) - \nabla f(x_0)^T x$ and show that $\Delta \geq 0$. Substituting the gradient into the expression for $\Delta$:

$$\Delta = \frac{(u^T x)^2}{v^T x} - \left(2\frac{u^T x_0}{v^T x_0}u - \frac{(u^T x_0)^2}{(v^T x_0)^2}v\right)^T x.$$

For compactness, let us define the scalars $a = \boldsymbol{u}^T \boldsymbol{x}$, $b = \boldsymbol{v}^T \boldsymbol{x}$, $a_0 = \boldsymbol{u}^T \boldsymbol{x}_0$, and $b_0 = \boldsymbol{v}^T \boldsymbol{x}_0$. Note that $b, b_0 > 0$ by the definition of $\mathcal{D}$. The expression becomes:

$$\Delta = \frac{a^2}{b} - \left( 2 \frac{a_0}{b_0} a - \frac{a_0^2}{b_0^2} b \right).$$

We bring terms to a common denominator $bb_0^2$:

$$\Delta = \frac{a^2 b_0^2 - 2aa_0 bb_0 + a_0^2 b^2}{bb_0^2} = \frac{(ab_0 - a_0 b)^2}{bb_0^2}.$$

Since the numerator is a perfect square and the denominator is strictly positive on $\mathcal{D}$, we conclude that $\Delta \geq 0$. $\qquad\square$

## G. Additional Experimental Results

In this appendix, we present additional experimental results which compare PIRATE with the baseline algorithms reported in Section 6 over real-world datasets. In particular, in Figure 3a we consider the *Movielens* dataset (Harper & Konstan, 2015). The observations are generated as follows: first, we selected the 25 movies with the highest number of ratings. Each user is treated as a sample. An independent portion of the dataset is used to estimate an underlying ranking from the average ratings.[6] The preferences observed by the algorithms are generated from the ratings of a given user: movie $A$ is preferred to movie $B$ according to user $U$ if and only if $U$ rated $A$ strictly higher than $B$. In this case, PIRATE converges to the underlying estimated ranking at a speed which is comparable with the baselines. In Figure 3b, we consider the *SUSHI preference* dataset (Kamishima, 2003). An independent portion of the dataset is used to compute the underlying ranking from full-rankings thanks to Borda count. The algorithms are trained by observing a MNAR portion of the comparisons in the full-ranking, extracted according to the following positional censoring: the first item in the latent ranking precedes the second which precedes the third and the third last item precedes the second last which precedes the last. In this case, the PIRATE algorithm slightly outperforms the baselines. The difference in performance is not pronounced due to the scarcity of data, a common problem of full-ranking datasets. It is important to stress that the speed of convergence is not the only factor to be taken into account. Indeed, the real advantage of the PIRATE algorithm lies in the mild assumptions on which the approach relies, which allow robust learning in real-world scenarios, even when most of the customary assumptions of the literature are violated.

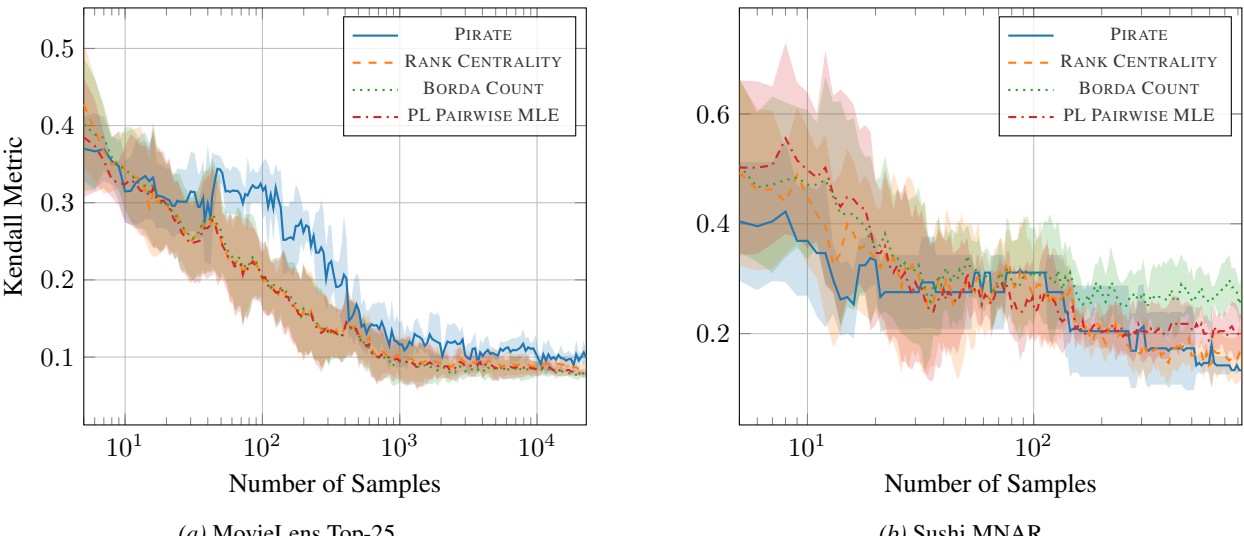

*(a)* MovieLens Top-25.    *(b)* Sushi MNAR.

*Figure 3.* Comparison of PIRATE against the baselines on real-world datasets. Shaded regions represent the 90% confidence intervals over 5 runs.

---

[6]Observe that this estimation, which is analogous to Borda count, is affected by MNAR bias, but it constitutes the only reasonable way of estimating a ranking if we cannot resort to our framework.

