# OpenReview forum: "Learning to Rank from Incomplete Rankings"
_ICML.cc/2026/Conference — ICML 2026 regular_

### Official Review · Reviewer_PRDS · 2026-03-02

**Soundness:** 2
**Presentation:** 3
**Significance:** 3
**Originality:** 3
**Overall Recommendation:** 4
**Confidence:** 3

**Summary:**

This paper considers the problem of recovering an underlying ranking of items from partial feedback. This is a well studied topic and the main contribution of this paper is to consider a more general setting than previous work. Specifically, the ranking model can be any distribution over permutations respecting two properties (strict complete consensus and full-support), and the feedback provided to the algorithm is called ''positional censoring'' and it is introduced in this paper. This feedback works by sampling a strict partial order over the positions {1, …, $n$}, and then returning the comparisons between the items occupying these positions in the sampled ranking. This type of feedback can be used to implement standard observation models like top-k or winner.

The authors consider the problem of exactly recovering the underlying ranking (the ranking having highest probability, which exists and is unique by the assumption of strict complete consensus) and provide a new algorithm that works in this more general model they introduced. The algorithm provided achieves a nearly optimal sample complexity. Specifically, the authors provide upper and lower bounds for this task that match up to a $O(\log(n))$ factor. They also show that in the special case of Plackett-Luce, the algorithm has better performances and recovers previously known guarantees.

**Compliance With Llm Reviewing Policy:**

Affirmed.

**Final Justification:**

I found this paper clear and enjoyable. I do not see the lack of experiments as a significant shortcoming since theory papers are explicitly welcomed in the CfP.

The rebuttal addressed most of my concerns. The only reason why I did not increase my score to "Accept" is because the proof of the lower bound for Plackett-Luce models as stated in the paper (Remark 5.1) requires a major fix, which I believe is a significant drawback, especially in theory papers. I still maintained my positive assessment of "Weak Accept" because the fix sketched by the authors in the rebuttal looks reasonable to me; and moreover, I acknowledge that the lower bound for Plackett-Luce models is a secondary point of this paper.

**Key Questions For Authors:**

In relation to W2, can you better describe how lower bounds for PAC settings can be used to derive lower bounds for your setting?

**Limitations:**

yes

**Strengths And Weaknesses:**

**Strengths:**

S1. The algorithm proposed is quite clean and intuitive. This is quite surprising considering the level of generality to which it operates and that it achieves an almost tight bound.

S2. I found very interesting that the results hold for any distribution of non-empty strict partial orders. This seems to heavily rely on the two assumptions, but I still found this surprising.

S3. The paper is well-written (although the notation is sometimes a bit heavy).


**Weaknesses:**

W1. It is not entirely clear what kind of new interesting scenarios can be modeled using these generalized tools. I would have appreciated more discussion on the motivation for this generalization.

W2. I have some concerns regarding Remark 5.1. Specifically, the lower bounds of Section 5 are obtained by using lower bounds derived from $(\epsilon, \delta)$-PAC settings, and by choosing $\epsilon$ smaller than the minimum distance between two PL-parameters. However, the lower bounds for these settings usually proceed by first fixing $\epsilon$ and then producing a family of instances that requires a certain number of samples to be learned. This means that the distance between two PL-parameters is defined only after choosing $\epsilon$ and might be smaller than $\epsilon$ itself.

As a concrete example, in Theorem 12 of [Saha & Gopalan, 2019a] (used in Table 2), the instances produced have multiple items with the same parameter value, and therefore the minimum distance between items is zero.

**Minor typos:**
- line 236 is a bit misleading, since it suggests that RUMs respect the full support assumption, which is in general not true
- Algorithm 1, line 3, $w_{j,i}$ is not defined when computing $N_{i,j}$
- Algorithm 1, line 10, mention that the sorting is in increasing order
- line 119, $\lambda$ is not defined


Overall I hold a favorable opinion of this work, but W2 should be properly addressed in the rebuttal.

---

> ### Author Rebuttal · Authors · 2026-03-30
>
> We thank the Reviewer for the time spent reviewing our work and for having appreciated it. Below, our answers to the Reviewer's questions and comments.
>
> > W1. It is not entirely clear what kind of new interesting scenarios can be modeled using these generalized tools. [...]
>
> Positional censoring arises naturally in many interesting scenarios, e.g. in ratings from users. Indeed, a user is likely to rate items that they liked or disliked the most. This induces preference feedback which suffer from MNAR bias that can be modeled through positional censoring. This setting is considered in Figure 1.b of the experimental campaign (please refer to the rebuttal to Reviewer `37Ev`, when answering to `W3`). The experiment shows that, even under the Plackett-Luce ranking model, the MNAR bias due to positional censoring is sufficient to prevent the baselines from learning the true ranking of the items. Conversely, PIRATE converges to the true ranking. This constitutes a strong motivation for our work: we will add the discussion to the camera-ready version.
>
> > W2. I have some concerns regarding Remark 5.1. Specifically, the lower bounds of Section 5 are obtained by using lower bounds derived from $(\epsilon, \delta)$-PAC settings [...]
>
> > In relation to W2, can you better describe how lower bounds for PAC settings can be used to derive lower bounds for your setting?
>
> We agree with the Reviewer that comparing PExact upper bounds with PAC lower bounds is informal. The intuition behind the choice of $\epsilon$ is that, by choosing an $\epsilon$ which is greater or equal to the minimum parameter gap, we allow the PAC learner to return a ranking which can differ from the true one, making the learning objective easier w.r.t. the PExact setting.
>
> Luckily, the machinery developed in the paper allows us to rigorously prove the results. We consider one case below; the others are analogous.
>
> **Statement.** Consider a Plackett-Luce ranking model of parameters $1 \geq a_1 > \dots > a_n \geq 1/2$. Let $h \leq n/2$. For any confidence parameter $\delta \in (0, 1/2)$, under top-$h$ feedback, the sample complexity of learning the true ranking of any $\delta$-correct learner is lower bounded by:
>
> $$\frac{n}{32h \min_l \Delta_{l,l+1}^2} \ln(1/(4\delta))$$
>
> where $\Delta_{l,l+1} = a_l - a_{l+1}$.
>
> **Proof sketch.** We are going to exploit Lemma E.5 (adapted to the offline setting from (Kaufmann et al., 2016)). Let $a = (a_1, \dots, a_n)$ with $1 \geq a_1 > \dots > a_n \geq 1/2$. Let $a^{(l)}$ be the same instance, but with parameters $a_l$ and $a_{l+1}$ swapped (for $l$ up to $n-1$). Fix such an $l$. Let $\mathcal{A}$ be the event that a $\delta$-correct learner returns the identity permutation. This event has probability $1-\delta$ under instance $a$ and probability $\delta$ under instance $a^{(l)}$. We can lower bound the right side of the inequality of Lemma E.5 with $\ln(1/(4\delta))$ (Lemma F.8). To conclude the proof we simply need to upper bound the KL divergence between the observation generated by $a$ and $a^{(l)}$ under top-$h$ feedback and rearrange the inequality. The required upper bound follows from the chain rule of the KL divergence and the properties of the PL model. Let $i_ r$ be the $r$-th ranked item and $S_ r = (i_ 1, \dots, i_ r)$ for $r$ up to $h$. Let $E$ denote the expectation and $P$ denote the probability measure. Then:
>
> $$D(S_ h || S_ h') = \sum_ {r=1}^h E_ S \left[ D(i_ r \mid S_ {r-1} = S || i_ r' \mid S'_ {r-1} = S) \right]$$
>
> $$\leq \sum_ {r=1}^h E_ S \left[ \sum_ {i \notin S} \frac{P(i_ r = i \mid S_ {r-1} = S)^2}{P(i_ r' = i \mid S'_ {r-1} = S)} - 1 \right] = \sum_ {r=1}^h E_ S \left[ \frac{A_ S^{(l)}}{A_ S^2} \sum_ {i \notin S} \frac{a_ i^2}{a_ i^{(l)}} - 1 \right]$$
>
> where $A_ S = \sum_ {j \notin S} a_ j$ and $A_ S^{(l)} = \sum_ {j \notin S} a_ j^{(l)}$, and the second line follows from (Popescu et al., 2016).
>
> By checking all cases and carrying out the algebraic steps, it is easy to show that the term inside the expectation reaches its maximum when $\\{ l, l+1 \\} \cap S = \\{ \\}$. In this scenario (i.e., $\\{ l, l+1 \\} \cap S = \\{ \\}$), the remaining parameter mass is identical: $A_S^{(l)} = A_S$. This simplifies the expression inside the expectation to:
>
> $$\frac{1}{A_S} \sum_{i \notin S} \frac{a_i^2}{a_i^{(l)}} - 1 = \frac{(a_l - a_{l+1})^2}{A_S} \frac{a_l + a_{l+1}}{a_l a_{l+1}}$$
>
> Since the parameters are bounded such that $a_i \in [1/2, 1]$, we can safely bound the fraction $\frac{a_l + a_{l+1}}{a_l a_{l+1}} \leq 8$. Furthermore, the remaining mass satisfies $A_S \geq (n-h)/2$. Assuming $h \leq n/2$, we get $A_S \geq n/4$. Thus, the KL divergence is bounded by:
>
> $$D(S_h || S_h') \leq 32 \frac{h}{n} \Delta_{l,l+1}^2$$
>
> As the inequality holds for any $l$ up to $n-1$, it also holds for the minimum ranging over $l$. Rearranging the inequality of Lemma E.5 produces the required result, concluding the proof.
>
> > Minor typos.
>
> We thank the Reviewer for the feedback; we will update the manuscript.

---

> > ### Author Rebuttal · Reviewer_PRDS · 2026-04-02
> >
> > I thank the authors for the response. It seems that indeed Remark 5.1 requires fixing. I maintain my current positive score because the new proof sketched by the authors seems reasonable.

---

### Official Review · Reviewer_viGf · 2026-03-07

**Soundness:** 2
**Presentation:** 3
**Significance:** 3
**Originality:** 3
**Overall Recommendation:** 5
**Confidence:** 4

**Summary:**

**Summary**

The paper introduces an algorithm for aggregating partial rankings in order to learn a ranking distribution. Previous work typically assumes a fixed model, such as the Mallows model (MM), and focuses on learning the central ranking from observations such as pairwise comparisons, top-k feedback, or samples of full permutations. Other works study alternative models such as the Plackett–Luce model (PL). Some classic papers assume the existence of a fixed ground-truth order and that pairwise comparisons are observed with noise (e.g., work by Mark Braverman and Elchanan Mossel; see the third bullet in the related work section of this review).

The paper claims to extend prior work in two main directions:

1- The proposed algorithm applies to any probability distribution over rankings, including MM and PL.

2- The observations do not need to follow a predefined structure; instead, they can be any arbitrary partially ordered set (poset).

I find the algorithm interesting, particularly because it allows observations to be arbitrary posets. However, I was curious how the algorithm achieves such generality. For example, when learning the center of a distribution under the Mallows model, the sample complexity typically depends on the dispersion parameter $\beta$.

After reading the proof of correctness, it appears that the analysis assumes each pairwise comparison is drawn from a binomial distribution. This suggests that the framework may be closer to the model studied by Mark Braverman and Elchanan Mossel rather than being applicable to arbitrary ranking distributions such as the Mallows model. I may be misunderstanding part of the analysis, and I would appreciate clarification from the authors. Please see the key questions section of this review.

Overall, I find the contribution interesting even if the results ultimately apply to a setting similar to the model of Mark Braverman and Elchanan Mossel. The paper presents solid theoretical work.

**Related Work**

Sections 2.3 and Appendix A list several observation models that have been studied previously. However, the following related lines of work appear to be missing:

- There is a substantial body of work on recovering the center of the Mallows model from pairwise comparisons. Examples include:

“Mallows Ranking Models: Maximum Likelihood Estimate and Regeneration” by Wenpin Tang (ICML 2019)

“Probabilistic Preference Learning with the Mallows Rank Model” by Valeria Vitelli et al. (JMLR 2018)

“Effective Sampling and Learning for Mallows Models with Pairwise-Preference Data” by Tyler Lu and Craig Boutilier (JMLR 2014)

- Some works treat the Mallows model as a choice model, where the observations correspond to selections from offered assortments—i.e., the top element of a sampled permutation restricted to a given subset. Examples include:

“Generalized Top-k Mallows Model for Ranked Choices” by Shahrzad Haddadan and Sara Ahmadian (ICML 2025)

“On a Mallows-Type Model for (Ranked) Choices” by Yifan Feng and Yuxuan Tang (NeurIPS 2022)

- There are also classic works that assume a ground-truth ordering with noisy pairwise comparisons sampled from it, such as:

“Noisy Sorting Without Resampling” by Mark Braverman and Elchanan Mossel (SODA 2018)

“Simple, Robust and Optimal Ranking from Pairwise Comparisons” by Nihar B. Shah and Martin J. Wainwright (JMLR 2018)

**Compliance With Llm Reviewing Policy:**

Affirmed.

**Final Justification:**

I think the paper is a solid theoretical paper tackling an important problem of learning a central ranking from observations. The paper studies the problem in a general setting: they don't assume any particular model or form of observation. Instead they show that if a model satisfies Complete Consensus and the observations are arbitrary posets that can have different structures, then the algorithm works.

In my initial assessment I was not sure if some parts of the results hold true, but after reading the authors response to my questions, and their response to other reviewers questions I am convinced that all the claims are valid.

In summary the paper is a solid theoretical paper and studies an important problem

**Key Questions For Authors:**

I have several questions regarding the theorems on the complexity and correctness of the algorithm and the assumptions used in the analysis.

**Complete Consensus**

On page 3, the paper defines the notion of complete consensus for ranking models. Could the authors provide an example of a model that satisfies complete consensus but is not unimodal? For instance, would a mixture of two Mallows models with different centers satisfy the complete consensus property?

In Assumption 3.1, the ranking model is assumed to satisfy complete consensus, whereas earlier the paper assumes the model is unimodal. Could the authors clarify the distinction between these two assumptions?

**Algorithm Correctness and Generality of Results**

Theorem 4.1 establishes the correctness of Algorithm 1. A key step occurs in line 5, where an additive error bound is obtained for estimating the direction of an edge. From the proof provided in the appendix, it appears that the analysis assumes that for any pair of items
$i,j$,  the number of times $i$ appears above  $j$ follows a binomial distribution.
I am not sure  whether this assumption holds for the models mentioned in the paper, such as the Mallows model. In MM, $i>j$ is not independent of the relative ordering of other pairs. Therefore, it is unclear whether the correctness guarantee extends to the Mallows model.

It seems that the analysis may instead correspond more closely to the framework of Mark Braverman and Elchanan Mossel (see the third bullet point  in the related work section). I would appreciate clarification on this point.

**Minor Question**

In Table 1, pairwise comparisons are not listed among the observation models. Is this intentional, or is it a typo? In other words, is the algorithm applicable to pairwise comparisons?

**Limitations:**

Yes

**Strengths And Weaknesses:**

strengths

- providing solid mathematical analysis for the propose algorithm
- the framework that observations may not follow a fixed structure is realistic and interesting

weakness

- the results may not be applicable to MM and PL as the authors claim.
- some important parts of the literature have not been discussed in related work

---

> ### Author Rebuttal · Authors · 2026-03-30
>
> We thank the Reviewer for the time spent reviewing our work and for having appreciated it. Below, our answers to the Reviewer's questions and comments.
>
> > When learning the center of a distribution under the Mallows model, the sample complexity typically depends on the dispersion parameter $\beta$.
>
> This is the case also in our work. Indeed, when the underlying ranking model is Mallows, the identification rates depend on the parameter $\beta$ ($\phi$ in the paper). In Appendix E.3.1, where we carry out the computations for Example 1, we show the closed-form expression of the factorized identifications rates as a function of $\phi$. Figure 1 depicts the evolution of the factorized identification rates as a function of $\phi$.
>
> > After reading the proof of correctness, it appears that the analysis assumes each pairwise comparison is drawn from a binomial distribution. [...]
>
> > The results may not be applicable to MM and PL as the authors claim.
>
> > It seems that the analysis may instead correspond more closely to the framework of Mark Braverman and Elchanan Mossel [...]
>
> > Theorem 4.1 establishes the correctness of Algorithm 1. A key step occurs in line 5, where an additive error bound is obtained for estimating the direction of an edge. From the proof provided in the appendix, it appears that the analysis assumes that for any pair of items $i, j$, the number of times $i$ appears above $j$ follows a binomial distribution. I am not sure whether this assumption holds for the models mentioned in the paper, such as the Mallows model. [...]
>
> Our theoretical results indeed hold for any ranking model satisfying Assumptions 1 and 2 (Complete Consensus and Full Support).
> For what regards the concern of the Reviewer on the proof of $\delta$-correctness, the Reviewer is likely referring to the fact that, in the proof of Lemma E.1, we formally show that the number of times we observe item $i$ preceding item $j$, conditioned on the number of observed comparisons between the two, follows a Binomial distribution. It is always true that the indicator variable of the event $i$ precedes $j$ in a given sample follows a Bernoulli distribution. These random variables can be correlated if we consider different pairs $(i, j)$ for the same sample, but are i.i.d. if we consider the same pair for different samples, inducing the Binomial distribution when we take the sum.
>
> Our observation model is different than that considered in "Noisy Sorting Without Resampling" by Braverman and Mossel who study an active setting where the learner observes a pairwise comparison. In our case instead, the learning problem is offline and the learner observes a strict partial order among items as a feedback. We will integrate the discussion on this work to our Related Works section.
>
> > Sections 2.3 and Appendix A list several observation models that have been studied previously. However, the following related lines of work appear to be missing [...]
>
> > Some important parts of the literature have not been discussed in related work.
>
> We thank the Reviewer for the references. We will add the additional works mentioned by the Reviewer to the Related Works section.
>
> > On page 3, the paper defines the notion of complete consensus for ranking models. Could the authors provide an example of a model that satisfies complete consensus but is not unimodal? [...]
>
> > In Assumption 3.1, the ranking model is assumed to satisfy complete consensus, whereas earlier the paper assumes the model is unimodal. Could the authors clarify the distinction between these two assumptions?
>
> A ranking model satisfying Complete Consensus is Strongly Unimodal (see, e.g., Critchlow et al., 1991), thus it is not possible to construct a ranking model which satisfies Complete Consensus but is not unimodal. Furthermore, the notion of Complete Consensus is defined in terms of the consensus ordering of the items, which corresponds to the modal ranking.
>
> > [Minor] In Table 1, pairwise comparisons are not listed among the observation models. Is this intentional? [...] is the algorithm applicable to pairwise comparisons?
>
> MCAR pairwise comparisons are implementable through a distribution over positional censorship mechanisms which assigns the same mass to all the pairs of positions $(l, m)$ with $l < m$. The guarantees provided in our work hold for the general case where a positional censorship mechanism is sampled from a distribution, including MCAR pairwise comparisons. In the table, however, we report just the observation models which can be implemented through a single censorship mechanism.

---

> > ### Author Rebuttal · Reviewer_viGf · 2026-04-03
> >
> > My questions about the following are resolved:
> > - discrepancy parameter $\beta$ and its relation to sample complexity
> > - table 1 and pairwise comparisons.
> > - Question about the usage of binomial distribution in the proof and whether it is assumed that the direction of all pairs $i$ and $j$ are independent in also resolved.
> >
> > I still have a few questions about the assumptions:
> >
> > 1- Baverman et al's work is not an active learning setting, and I don't mean that your work is similar or dissimilar to this paper from this perspective.  I think that noise model is similar. In this paper, each edge is samples from a true ranking but the direction may be flipped with a certain probability. Is your model applicable to this *noise* model?
> >
> > 2- Can you provide a concrete example that is Complete Consensus but is not unimodal.

---

> > > ### Author Response · Authors · 2026-04-03
> > >
> > > We thank the Reviewer for the acknowledgment. We are happy we clarified several points. Below, our answers to the remaining concerns.
> > >
> > > 1. In "Noisy Sorting Without Resampling" by Braverman and Mossel, the outcome of $\binom{n}{2}$ noisy comparisons among items is sampled independently once. The learner has access to the outcome of each of these queries (which is not resampled) and needs to aggregate them into a ranking. With the _active_ adjective, we wanted to refer to the fact that, despite the queries being sampled all at once, the learner does not need to process all of them. Indeed, Theorem 5 proves the existence of an algorithm with sampling complexity $O(n \log n)$ while the number of queries is $\Omega(n^2)$. Regarding the noise model, it is **different** w.r.t. the one considered in our work, as our focus is learning from incomplete rankings where the missing preferences are Missing Not At Random. Moreover, in our work, the preferences among different pairs in the same incomplete ranking are not independent (in general) one from the other. In "Noisy Sorting Without Resampling" instead, pairwise preferences are independent. The second (and most important) difference, is the fact that, in our setting, the learner cannot control the preferences that are missing and needs to handle the fact that they are Missing Not At Random. In "Noisy Sorting Without Resampling" instead, the learner can potentially access each of the pairwise queries. Indeed, there is no missingness in such a setting, making it unrelated to ours. For these reasons, our model is **not applicable** to the noise model of Braverman and Mossel, and the noise model of Braverman and Mossel is not applicable to ours. We will add this detailed discussion to the paper.
> > >
> > > 2. Unfortunately, this is not possible. Indeed, as remarked in the rebuttal, **Complete Consensus implies Unimodality** (Critchlow et al., 1991), hence there is no ranking model which satisfies Complete Consensus but is not unimodal.
> > >
> > > ---
> > >
> > > Critchlow, D. E., Fligner, M. A., and Verducci, J. S. Probability models on rankings. Journal of Mathematical
> > > Psychology, 1991

---

### Official Review · Reviewer_HC1W · 2026-03-13

**Soundness:** 4
**Presentation:** 3
**Significance:** 3
**Originality:** 3
**Overall Recommendation:** 4
**Confidence:** 4

**Summary:**

This paper studies ordered top-k identification from incomplete rankings. It introduces an observation model called positional censoring, which unifies several standard feedback types, and proposes an algorithm based on pairwise comparisons. The paper provides sample complexity upper bounds, and matching lower bounds up to log factors. They also show that their algorithm has a near-optimal sample complexity in some special cases.

**Compliance With Llm Reviewing Policy:**

Affirmed.

**Key Questions For Authors:**

1. Does the algorithm presented match the concrete efficiency of existing algorithms for more restrictive feedback settings?

**Limitations:**

Yes

**Strengths And Weaknesses:**

The main strength is the generality of the framework. The positional censoring model appears to strictly generalize several classical observation settings. The paper also gives correctness guarantees, and almost tight sample complexity bounds.

The main weakness is the lack of experiments. Even a small synthetic study would help illustrate the behavior of the algorithm and the practical meaning of the hardness terms. In addition, the presentation could be improved. For example, it would be nice to have a short lemma or proof sketch to help understand the intuitions rather than deferring everything technical to the appendix. The assumptions of strict complete consensus and full support are also somewhat strong.

I think this is a good theory paper with a clean modeling contribution and meaningful technical results. My main concern is that the paper is almost entirely theoretical, with little evidence about empirical behavior. Still, the analysis appears substantial, so despite
the limitations mentioned above, I lean toward a borderline accept.

---

> ### Author Rebuttal · Authors · 2026-03-30
>
> We thank the Reviewer for the time spent reviewing our work and for having appreciated our contribution and the generality of our setting. Below, our answers to the Reviewer's questions and comments.
>
> > The main weakness is the lack of experiments. Even a small synthetic study would help illustrate the behavior of the algorithm and the practical meaning of the hardness terms.
>
> > My main concern is that the paper is almost entirely theoretical, with little evidence about empirical behavior.
>
> We agree with the Reviewer on the importance of empirical validation.
> To avoid redundancy in our response, we kindly ask the Reviewer to refer to the results of our experimental campaign, which are detailed in our response to Reviewer `37Ev` under Weakness `W3`.
>
> > In addition, the presentation could be improved. For example, it would be nice to have a short lemma or proof sketch to help understand the intuitions rather than deferring everything technical to the appendix.
>
> We thank the Reviewer for the feedback. We will update the manuscript, adding proof sketches to the sections of the lower bound (Theorem 4.3) and the factorization of the identification rate (Corollary 4.4), which are those that lack the most of the intuitions behind the technical results, exploiting the additional page in the final version.
>
> > The assumptions of strict complete consensus and full support are also somewhat strong.
>
> In the Preliminary section of the paper, we show that almost the entirety of the ranking models studied in the literature satisfy both the assumptions of Complete Consensus and Full Support, including the ubiquitous Plackett-Luce and Mallows ranking models. Furthermore, in Theorems D.1 and D.2 we prove non-learnability results outside of these reasonable assumptions.
>
> > Does the algorithm presented match the concrete efficiency of existing algorithms for more restrictive feedback settings?
>
> In Figure 1.a of the experimental campaign (see the rebuttal to Reviewer `37Ev` while addressing weakness `W3`), we show that the baselines slightly outperform PIRATE on the restricted feedback setting: Plackett-Luce ranking model with MCAR comparisons. Anyway, the performance is still comparable. This is reasonable for an algorithm with theoretical guaranteees on a more general setting.
> For what regards the theoretical results, in Table 2 we show that the algorithm attains, up to logarithmic dependencies, the minimax sample complexity rate under the Plackett-Luce ranking model with well-established types of feedback (winner, top-$h$, full-ranking).

---

> > ### Author Rebuttal · Reviewer_HC1W · 2026-04-04
> >
> > Thank you for your detailed response. I have no further questions or concerns.

---

### Official Review · Reviewer_37Ev · 2026-03-18

**Soundness:** 3
**Presentation:** 2
**Significance:** 2
**Originality:** 3
**Overall Recommendation:** 3
**Confidence:** 3

**Summary:**

This paper investigates the task of learning the top k items from an incomplete ranking. It highlights the rigorous assumptions of existing methods regarding the ranking model and the censoring mechanism. Then, it proposes a novel, general framework for learning incomplete rankings. It introduces a new preference-based feedback model to generalize existing methods and provides a rigorous generalization of existing frameworks. Based on this learning framework, the PIRATE algorithm is proposed, and its sample complexity is shown to be comparable to that of existing algorithms.

**Compliance With Llm Reviewing Policy:**

Affirmed.

**Final Justification:**

I am not an expert in theoretical topics, so my assessment and final justification are based on my perspective as a researcher working on the applications (e.g., recommendations, advertising) of the topics studied in this paper.

I agree that this paper provides some rigorous theoretical insights into the task of learning to rank, under certain relaxed assumptions. However, I also feel that its value beyond theoretical insights may be limited for the community applying this task, as the gains from these relaxed assumptions in practice are likely to be modest. Furthermore, I do think this submission is very weak in terms of experimental insights (perhaps this is permissible in theoretical topics, and I will listen to the AC's judgment). Even though the authors provided some additional experiments in their rebuttals, I find it difficult to see any consensus advantage.

Therefore, although I am in the minority, from my perspective, this submission does indeed have significant shortcomings and does not meet the acceptable threshold.

**Key Questions For Authors:**

* Q1: Could you discuss the impact on the proposed learning framework or algorithm if the positional censoring-based MNAR mechanism is violated? Are there any measures that can mitigate this impact?
* Q2: Can you provide the necessary experimental verification to demonstrate the effectiveness and generality of the proposed learning framework or algorithm?

**Limitations:**

* L1: The applicability of positional censoring-based MNAR mechanisms lacks sufficient discussion.
* L2: The current version is very weak in terms of experimental verification.

**Strengths And Weaknesses:**

**Strengths:**

* S1: This paper explores a potentially valuable task scenario: accurately learning the top k item rankings from incomplete rankings, such as for recommender systems and training large language models.
* S2: This paper proposes a new general learning framework for this task scenario, which introduces a positional censoring-based feedback model to consider the MNAR mechanism for missing data instead of the MCAR mechanism, in order to be closer to the real-world scenario, and demonstrates compatibility with existing methods.
* S3: Based on the proposed learning framework, a PIRATE algorithm is presented as a concrete example of the execution strategy. It also provides theoretical insights into the algorithm, including upper and lower bounds on sample complexity and the decomposition of the identification rate.

**Weaknesses:**

* W1: The clarity of the paper's presentation can be further enhanced. Some important suggestions include: 1) providing a diagram of the problem to help readers better understand it; 2) due to the large number and complexity of symbols introduced, it is necessary to provide a comprehensive table to summarize them; 3) discussing existing methods, proposed solutions, and the differences between them in relation to ranking and feedback models may be more beneficial for readability.
* W2: The discussion on the rationale behind the methodology is insufficient. A key issue is that the generality and limitations of the positional censoring-based MNAR mechanism have not been clearly discussed. For example, in recommender systems, location bias is only one part of the overall bias effect, and many other factors contribute to various potential biases and data gaps. Another key issue is that the PIRATE algorithm's complexity and generality relative to existing methods have not been clearly demonstrated.
* W3: The current version of this paper is very weak in terms of its experimental section. Although the main contribution lies in the theoretical development, the theory's validity still needs to be verified. Furthermore, demonstrating the theory's practicality and applicability is essential, especially through validation on real-world datasets.

---

> ### Author Rebuttal · Authors · 2026-03-30
>
> We thank the Reviewer for the time spent reviewing our work. Below, our answers to the Reviewer's questions and concerns.
>
> > W1: The clarity of the paper's presentation can be further enhanced. [...]
>
> We thank the Reviewer for the feedback.
> We are going to adapt the manuscript in order to improve its clarity by implementing all the suggestions.
>
> > W2: The discussion on the rationale behind the methodology is insufficient. [...]
>
> The comparison of PIRATE against existing methods is addressed in the answer to `W3` by presenting our experimental campaign.
> To prove the applicability of the setting to practical scenarios, we conducted experiments on the behavior of the PIRATE algorithm on real-world data. These are extensively discused when we address `W3`.
> We agree with the Reviewer that positional filtering is just one of the sources of bias in real-world applications. For example, the way in which the items are presented to the user, e.g. on a web page, induces another bias on the missingness of preferences. Anyway, our method constitutes a step towards the development of robust techniques that can learn despite MNAR observations in real-world datasets.
>
> > W3: The current version of this paper is very weak in terms of its experimental section. [...]
>
> > Q1: Could you discuss the impact on the proposed learning framework or algorithm if the positional censoring-based MNAR mechanism is violated? [...]
>
> > Q2: Can you provide the necessary experimental verification to demonstrate the effectiveness and generality of the proposed learning framework or algorithm?
>
> > L1, L2: [...]
>
> Figure 1 at the following link provides the result of our experimental campaign: [https://drive.proton.me/urls/6YBYJNC12W#oeKWNAl28ISN](https://drive.proton.me/urls/6YBYJNC12W#oeKWNAl28ISN).
> We compared the PIRATE algorithm against well-established offline Learn-to-Rank baselines: the RankCentrality algorithm (Negahban et al., 2017), the ML estimation of Plackett-Luce parameters from pairwise comparisons (PL Pairwise MLE) (Khetan & Oh, 2016), and BordaCount (Ranking: Compare, don’t score, Ammar and Shah, 2011).
> The algorithms are compared in terms of Kendall Tau distance of the predicted ranking against an underlying ranking which is either the true one (for synthetic data) or has been obtained through estimation (for real-world data).
> Figure 1.a and 1.b show the result of simulations on synthetic data. In both cases, a latent ranking is generated from the Plackett-Luce ranking model. In Figure 1.a the learners observe a pairwise comparison of a pair of items chosen uniformly at random (MCAR). Surprisingly, PIRATE obtains a comparable performance w.r.t. baselines specifically designed for this restricted setting.
> In Figure 1.b, we consider instead a very natural positional censorship mechanism: the learners observe that the first item in the latent ranking is preferred to the second which is preferred to the third and that the third last item is preferred to the second last which is preferred to the last. This positional filter arises normally in practical application as a user is more prone to rate the items which they very like or that they very dislike. Even under the Plackett-Luce ranking model, the MNAR bias is sufficient to cause the baselines to converge to the wrong ranking. This shows that the well-established methods in the literature are very sensitive to MNAR bias, providing a strong motivation for our work. Indeed, the PIRATE algorithm is the only one capable of learning the true underlying learning.
> Figure 1.c and 1.d regard experimental validation on real-world datasets.
> In particular, Figure 1.c shows a comparison of the algorithm on the MovieLens-25M dataset.
> We consider only the 25 movies with the highest number of ratings. An independent portion of the dataset is used to estimate an underlying ranking according to the average rating. Observe that this estimation, which is analogous to BordaCount, is affected by MNAR bias, but it constitutes the only reasonable way of estimating a ranking if we cannot resort to our framework. The effect of the MNAR bias is mitigated by the vast amount of independent users that provided ratings. User ratings naturally induce a strict partial order over the movies, allowing to generate preferences. Despite the choice of the underlying ranking, which favors the baselines as it is aligned with BordaCount, PIRATE shows only sligthly slower convergence.
> The experiment depicted in Figure 1.d instead tries to solve the problem of MNAR bias in the estimation of the underlying ranking by relying on a dataset of full-rankings: the Sushi Preference Data. Here the learners observe MNAR comparisons obtained from the full-ranking using the same positional censoring of the synthetic instance considered in Figure 1.b. In this case, the PIRATE algorithm slightly outperforms the baselines. The difference in performance is not net due to the scarcity of data, a common problem of full-ranking datasets.

---

> > ### Author Rebuttal · Reviewer_37Ev · 2026-04-04
> >
> > I appreciate the authors' responses to my previous concerns.
> >
> > Although some experimental results involving both synthetic and real-world data have been presented, my concerns about the weaknesses in the experimental section persist. These concerns primarily fall into three areas:
> > * The experimental setup lacks clarity, and the chosen baselines do not appear sufficiently representative. Are recent methods rare in this research field?
> > * Based on current results, PIRATE appears to have a disadvantage compared to the baseline in some cases, especially with small sample sizes. This reinforces my concerns regarding the practical applicability of the proposed framework beyond its theoretical underpinnings.
> > * Strengthening the experimental section would necessitate substantial revisions to the current version of the manuscript. Such revisions could potentially introduce new issues regarding presentation and clarity.
> >
> > Consequently, I maintain that my initial assessment remains valid.

---

> > > ### Author Response · Authors · 2026-04-05
> > >
> > > We thank the Reviewer for the follow-up.
> > >
> > > > The experimental setup lacks clarity.
> > >
> > > Given the rebuttal space limit (5000 characters), we focused on reporting the key components of the setup: the baselines, the evaluation metric, and the synthetic data generation process. We will make our best efforts to present them more clearly in the revised manuscript, including implementation details and a more detailed discussion of the results.
> > >
> > > > The chosen baselines do not appear sufficiently representative. Are recent methods rare in this research field?
> > >
> > > Our work is, to the best of our knowledge, **the first to introduce the positional censoring** feedback model. As a consequence, there are **no existing baselines** designed for this setting. For this reason, we compare against well-established methods from offline learning-to-rank under MCAR (Missing Completely At Random) pairwise comparisons, a setting that is a strict special case of our positional censoring. In particular:
> > >
> > > - RankCentrality (Negahban et al., 2017) and the Maximum Likelihood estimator (Khetan & Oh, 2016) are designed for the Plackett–Luce model, one of the most widely adopted ranking models in the literature. Both methods are minimax optimal in the MCAR setting.
> > > - BordaCount (Ammar and Shah, 2011) is a non-parametric baseline that does not rely on a specific parametric ranking model.
> > >
> > > When the MCAR assumptions are violated, the baselines converge to an incorrect ranking (Figure 1.b), whereas PIRATE still converges to the correct ranking. When the MCAR assumptions hold, PIRATE achieves performance in line with these established methods (Figure 1.a). In this sense, the baselines are **representative** because they **demonstrate** that the **MCAR assumption** we relax through positional censoring **is not merely technical**; it is in fact necessary for existing methods to succeed.
> > >
> > > > Based on current results, PIRATE appears to have a disadvantage compared to the baseline in some cases, especially with small sample sizes. This reinforces my concerns regarding the practical applicability of the proposed framework beyond its theoretical underpinnings.
> > >
> > > We believe that the described behavior of PIRATE does not compromise its practicality. Indeed:
> > >
> > > - PIRATE is the **only algorithm** that converges to the **correct ranking in all four scenarios** when using the complete dataset (the rightmost point in the plots).
> > > - PIRATE displays **comparable** Kendall tau loss in the three scenarios in which the baselines also converge, with the exception of two cases in which, when limited to a moderate-sample regime, it exhibits a larger Kendall tau loss than the baselines.
> > >
> > > Given that PIRATE operates under **significantly weaker assumptions** than the baselines and succeeds in converging in all scenarios, a larger error in the moderate-sample regime is a **practically acceptable** phenomenon.
> > >
> > > > Strengthening the experimental section would necessitate substantial revisions to the current version of the manuscript. Such revisions could potentially introduce new issues regarding presentation and clarity.
> > >
> > > We believe that since (i) the main contribution of the paper is theoretical and (ii) the experiments we provided during the rebuttal already demonstrate that PIRATE performs well (as explained in the previous answers), there is no need to strengthen the experimental section. This would avoid any potential clarity or presentation issues.

---

### Decision · Program_Chairs · 2026-04-30

**Decision:**

Accept (regular)

**Comment:**

This is a solid theoretical work where the authors provide an algorithm to learn the underlying ranking under relaxed conditions for both the ranking model and the feedback type - in particular, MCAR is a pretty strong assumption in practice and theoretical works towards MNAR are important. I went through all the reviews and rebuttals carefully - while real-world experimental results would have definitely strengthened the paper, the main contribution of this work is theoretical and well motivated.

I agree with the authors that experiments in such instances are only for validation of the theory. Further, on common datasets, the standard assumptions hold and therefore it is also challenging to find appropriate datasets where the assumptions might not be true. Nevertheless, the authors have provided reasonable empirical results for a theoretical paper. I request the authors to add the synthetic studies and the real-world experiments to the paper.

However, I strongly encourage the authors to provide a detailed discussion on limitations of the positional censoring-based MNAR mechanism and also point towards more practical models for capturing other biases (in the camera-ready).